# The Non-Linear Representation Dilemma: Is Causal Abstraction Enough for Mechanistic Interpretability?

**Denis Sutter,**[ETH] **Julian Minder,**[EPFL] **Thomas Hofmann,**[ETH] **Tiago Pimentel**[ETH]
[ETH]ETH Zürich [EPFL]EPFL
densutter@ethz.ch, julian.minder@epfl.ch, {thomas.hofmann, tiago.pimentel}@inf.ethz.ch
densutter/non-linear-representation-dilemma

## Abstract

The concept of **causal abstraction** got recently popularised to demystify the opaque decision-making processes of machine learning models; in short, a neural network can be **abstracted** as a higher-level algorithm if there exists a function which allows us to map between them. Notably, most interpretability papers implement these maps as linear functions, motivated by the **linear representation hypothesis**: the idea that features are encoded linearly in a model's representations. However, this linearity constraint is not required by the definition of causal abstraction. In this work, we critically examine the concept of causal abstraction by considering arbitrarily powerful alignment maps. In particular, we prove that under reasonable assumptions, *any* neural network can be mapped to *any* algorithm, rendering this unrestricted notion of causal abstraction trivial and uninformative. We complement these theoretical findings with empirical evidence, demonstrating that it is possible to perfectly map models to algorithms even when these models are incapable of solving the actual task; e.g., on an experiment using randomly initialised language models, our alignment maps reach 100% interchange-intervention accuracy on the indirect object identification task. This raises the **non-linear representation dilemma**: if we lift the linearity constraint imposed to alignment maps in causal abstraction analyses, we are left with no principled way to balance the inherent trade-off between these maps' complexity and accuracy. Together, these results suggest an answer to our title's question: causal abstraction is *not* enough for mechanistic interpretability, as it becomes vacuous without assumptions about how models encode information. Studying the connection between this information-encoding assumption and causal abstraction should lead to exciting future work.

## 1 Introduction

The increasing popularity of machine learning (ML) models has led to a surge in their deployment across various industries. However, the lack of interpretability in these models raises significant concerns, particularly in high-stakes applications where understanding the decision-making process is crucial (Goodman and Flaxman, 2017; Tonekaboni et al., 2019; Zhu et al., 2020; Gao and Guan, 2023). Unsurprisingly, this opacity has motivated a multitude of research on mechanistic (or causal) interpretability, which tries to analyse and understand the hidden algorithms that underlie these models (Olah et al., 2020; Elhage et al., 2021; Mueller et al., 2024; Ferrando et al., 2024; Sharkey et al., 2025).

A promising approach to address this challenge is **causal abstraction** (Beckers and Halpern, 2019; Geiger et al., 2024a), which tries to map the behaviour of a model to a higher-level (and conceptually simpler) algorithm which solves the task. At the core of this concept is the idea that if an intervention is found to change a model's behaviour in a way that aligns with a specific algorithm, then that algorithm can be considered implemented by the model. Recent research, however, has raised considerable issues with this approach (e.g., Makelov et al., 2024; Mueller, 2024; Sun et al., 2025). Among those, Méloux et al. (2025) notes that a model's causal abstraction is not necessarily unique, showing that many algorithms can be aligned to the same neural network. Additionally, most work on causal

39th Conference on Neural Information Processing Systems (NeurIPS 2025).

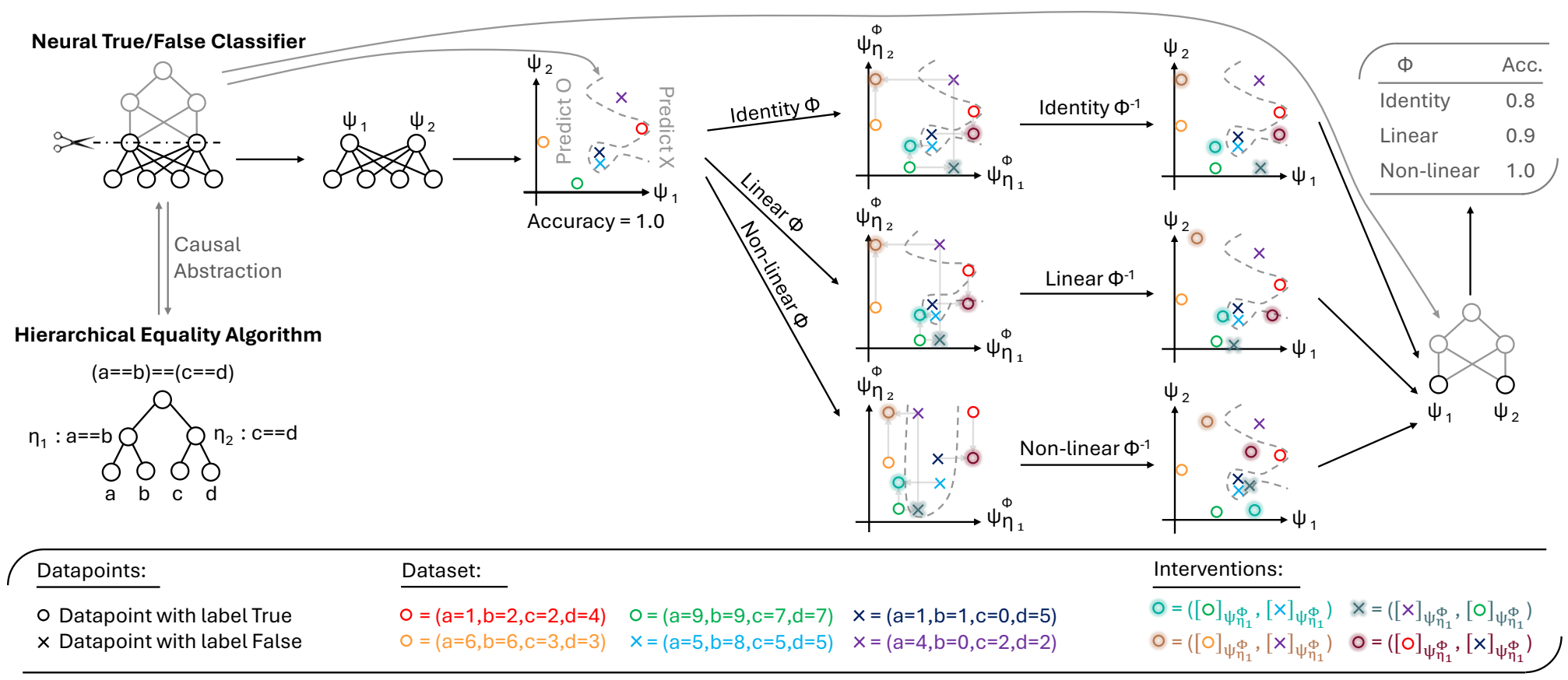

Figure 1: A visualisation of what happens when analysing causal abstractions with increasingly complex alignment maps $\phi$. The more complex $\phi$ is, the higher the intervention accuracy—and, consequently, the stronger the algorithm–DNN alignment. In Theorem 1, we show that given arbitrarily complex alignment maps, we can always find a perfect alignment (under reasonable assumptions).

abstraction (Wu et al., 2023; Geiger et al., 2024b; Minder et al., 2025; Sun et al., 2025) implicitly assumes information is linearly encoded in models' representations, relying on the **linear representation hypothesis** (Alain and Bengio, 2016; Bolukbasi et al., 2016). Linearity, however, is not required by the definition of causal abstraction (Beckers and Halpern, 2019) and increasing evidence suggests that not all representations may be linearly encoded (White et al., 2021; Olah and Jermyn, 2024; Mueller, 2024; Csordás et al., 2024; Engels et al., 2025a,b; Kantamneni and Tegmark, 2025).

In this paper, we first prove that, once we drop the linearity constraint, *any* model can be perfectly mapped to *any* algorithm under relatively weak assumptions—e.g., hidden activation's input-injectivity and output-surjectivity, which we will define formally. This renders causal abstraction vacuous when used without constraints. If we restrict alignment maps to only consider, e.g., linear functions, this problem does not arise though. It follows that causal abstraction implicitly relies on strong assumptions about how features are encoded in deep neural networks (DNNs), and becomes trivial without such assumptions. This puts us at an impasse: we may want to rely on stronger notions of causal abstraction which may leverage non-linearly encoded information, but this may make our analyses vacuous; we call this the **non-linear representation dilemma** (schematised in Fig. 1).

To empirically validate our theoretical results, we reproduce the original distributed alignment search (DAS) experiments (Geiger et al., 2024b), but while leveraging more complex alignment maps. We find that key empirical patterns they observed—such as the first layer being easier to map to the tested algorithms in a hierarchical equality task—vanish when we use more powerful maps. Additionally, we find that we can achieve over 80% interchange intervention accuracy (IIA) using non-linear alignment maps in randomly initialised models. Extending our experiments to language models from the Pythia suite (Biderman et al., 2023), we show that near-perfect maps can be found for randomly initialised models in the indirect object identification (IOI) task (Wang et al., 2023); notably, as training progresses, the complexity of the alignment maps needed to achieve perfect IIA in this task decreases. Overall, our results show that causal abstraction, while promising in theory, suffers from a fundamental limitation: without *a priori* constraints on the used alignment maps, it becomes vacuous as a method for understanding neural networks.

## 2 Background

In this section, we formally define algorithms (§2.1) and deep neural networks (§2.2). These will then be used to define a causal abstraction (§3). First, we formalise a task as a function $\mathtt{T} : \mathcal{X} \to \mathcal{Y}$, where $\mathbf{x} \in \mathcal{X}$ represents a set of input features and $\mathbf{y} \in \mathcal{Y}$ denotes the corresponding output.

### 2.1 Algorithms

Given a task $\mathtt{T}$, we may hypothesise different ways it can be solved. We term each such hypothesis an algorithm[1] $\mathtt{A}$, which we represent as a deterministic causal model—a directed acyclic graph that

[1] "Algorithm" here need not match a formal definition as, e.g., the considered functions may be uncomputable.

implements a function $f_{\mathtt{A}} : \mathcal{X} \to \mathcal{Y}$.[2] These causal models have a set of nodes $\boldsymbol{\eta}_{\mathtt{all}}$ which can be decomposed into three disjoint sets: (i) input nodes $\boldsymbol{\eta}_{\mathbf{x}}$ representing elements in $\mathbf{x}$, (ii) output nodes $\boldsymbol{\eta}_{\mathbf{y}}$ representing elements in $\mathbf{y}$, and (iii) inner nodes $\boldsymbol{\eta}_{\mathtt{inner}}$ representing intermediate variables used in the computation of $f_{\mathtt{A}}$. As we focus on acyclic causal models, the edges in this graph induce a partial ordering on nodes $\boldsymbol{\eta}_{\mathbf{x}} \prec \boldsymbol{\eta}_{\mathtt{inner}} \prec \boldsymbol{\eta}_{\mathbf{y}}$. Let $v_\eta$ denote the value held by node $\eta$, and let $\mathbf{v}_{\boldsymbol{\eta}}$ denote values taken by the set of nodes $\boldsymbol{\eta}$. The set of incoming edges to a node $\eta$ represent a direct causal relationship between that node and its parents $\mathtt{par}_{\mathtt{A}}(\eta)$, denoted: $v_\eta = f^{\eta}_{\mathtt{A}}(\mathbf{v}_{\mathtt{par}_{\mathtt{A}}(\eta)})$. We can compute algorithm $f_{\mathtt{A}}$ by iteratively solving the value of its nodes while respecting their partial ordering:

$$\mathbf{v}_{\boldsymbol{\eta}_{\mathbf{x}}} \overset{\text{def}}{=} \mathbf{x}, \qquad \underset{\eta \in \boldsymbol{\eta}_{\mathtt{inner}} \cup \boldsymbol{\eta}_{\mathbf{y}}}{\forall} v_\eta = f^{\eta}_{\mathtt{A}}(\mathbf{v}_{\mathtt{par}_{\mathtt{A}}(\eta)}), \qquad f_{\mathtt{A}}(\mathbf{x}) \overset{\text{def}}{=} \mathbf{v}_{\boldsymbol{\eta}_{\mathbf{y}}} \tag{1}$$

where we define $\mathbf{v}_{\boldsymbol{\eta}_{\mathbf{x}}}$ to be the input value $\mathbf{x}$, and take the value of the output nodes $\mathbf{v}_{\boldsymbol{\eta}_{\mathbf{y}}}$ as our algorithm's output. Importantly, for an algorithm $\mathtt{A}$ to represent a task $\mathtt{T}$, its output under "normal" operation must be $f_{\mathtt{A}}(\mathbf{x}) = \mathtt{T}(\mathbf{x})$. For an example of a task and related algorithms, see App. I.1.

These causal models, however, allow us to go beyond "normal" operations and investigate the behaviour of our algorithm under counterfactual settings. We can, for instance, investigate what its behaviour would be if we enforce a node $\eta'$'s value to be a constant $v_{\eta'} = c$, which we write as:

$$\mathbf{v}_{\boldsymbol{\eta}_{\mathbf{x}}} = \mathbf{x}, \quad v_{\eta'} = c, \quad \underset{\eta \in (\boldsymbol{\eta}_{\mathtt{inner}} \cup \boldsymbol{\eta}_{\mathbf{y}}) \setminus \{\eta'\}}{\forall} v_\eta = f^{\eta}_{\mathtt{A}}(\mathbf{v}_{\mathtt{par}_{\mathtt{A}}(\eta)}), \quad f_{\mathtt{A}}(\mathbf{x}, (v_{\eta'} \leftarrow c)) = \mathbf{v}_{\boldsymbol{\eta}_{\mathbf{y}}} \tag{2}$$

Now, let $f^{:\eta}_{\mathtt{A}}(\mathbf{x})$ represent a function which runs our algorithm with input $\mathbf{x}$ until it reaches node $\eta$, outputting its value $v_\eta$. We can use such interventions to investigate the behaviour of algorithm $\mathtt{A}$ under input $\mathbf{x}$, when node $\eta'$ is forced to assume the value it would have under $\mathbf{x}'$ as: $f_{\mathtt{A}}(\mathbf{x}, (v_{\eta'} \leftarrow f^{:\eta'}_{\mathtt{A}}(\mathbf{x}')))$. Now, let $\mathbf{I}_{\mathtt{A}}$ be a multi-node intervention, e.g., $\mathbf{I}_{\mathtt{A}} = (\mathbf{v}_{\boldsymbol{\eta}'} \leftarrow \mathbf{c}')$ where $\boldsymbol{\eta}' = [\eta', \eta'']$ and $\mathbf{c}' = [f^{:\eta'}_{\mathtt{A}}(\mathbf{x}'), c]$. We can observe how our model operates under those interventions by running $f_{\mathtt{A}}(\mathbf{x}, \mathbf{I}_{\mathtt{A}})$. See App. B for a pseudo-code implementation.

### 2.2 Deep Neural Networks

Deep neural networks (DNNs) are the driving force behind recent advances in ML and can be defined as a sequence of functions $f^{\ell}_{\mathtt{N}} : \mathcal{H}_{\boldsymbol{\psi}_\ell} \to \mathcal{H}_{\boldsymbol{\psi}_{\ell+1}}$, where $\boldsymbol{\psi}_\ell$ denotes the set of neurons in layer $\ell$ and $\mathcal{H}_{\boldsymbol{\psi}_\ell}$ is the corresponding vector space. A DNN $\mathtt{N}$ with $L$ layers can be specified as follows:

$$\mathbf{h}_{\boldsymbol{\psi}_1} = f^{0}_{\mathtt{N}}(\mathbf{x}), \qquad \mathbf{h}_{\boldsymbol{\psi}_{\ell+1}} = f^{\ell}_{\mathtt{N}}(\mathbf{h}_{\boldsymbol{\psi}_\ell}), \qquad p_{\mathtt{N}}(\mathbf{y} \mid \mathbf{x}) = f^{L}_{\mathtt{N}}(\mathbf{h}_{\boldsymbol{\psi}_L}) \tag{3}$$

where $\mathbf{h}_{\boldsymbol{\psi}_\ell}$ denotes the vector of activations for neurons $\boldsymbol{\psi}_\ell$. We focus on DNNs with real-valued neurons and probabilistic outputs, so that $\mathcal{H}_{\boldsymbol{\psi}_0} = \mathcal{X}$, $\mathcal{H}_{\boldsymbol{\psi}_\ell} = \mathbb{R}^{|\boldsymbol{\psi}_\ell|}$ and $\mathcal{H}_{\boldsymbol{\psi}_{L+1}} = \Delta^{|\mathcal{Y}|-1}$. We define $f^{\boldsymbol{\psi}'}_{\mathtt{N}}(\mathbf{x}')$ as the function that computes the activations of the subset of neurons $\boldsymbol{\psi}'$ when the network is evaluated on input $\mathbf{x}'$. In particular, $f^{\boldsymbol{\psi}_\ell}_{\mathtt{N}}(\mathbf{x})$ returns the activations at layer $\ell$ for input $\mathbf{x}$. Thus, the standard computation of the DNN corresponds to evaluating $p_{\mathtt{N}}(\mathbf{y} \mid \mathbf{x}) = f^{\boldsymbol{\psi}_{L+1}}_{\mathtt{N}}(\mathbf{x})$. This formulation allows us to instantiate common architectures, such as multi-layer perceptrons (MLPs) or transformers, by specifying the form of each $f^{\ell}_{\mathtt{N}}$ and the structure of the neuron sets $\boldsymbol{\psi}_\ell$. The parameters of these models are typically optimised to minimise the cross-entropy loss.

Notably, similarly to the algorithms above, a DNN's architecture induces a partial ordering on its neurons, respecting the order in which they are computed $\boldsymbol{\psi}_0 \prec \boldsymbol{\psi}_\ell \prec \boldsymbol{\psi}_L$. We can thus analogously define a **DNN intervention** as follows: given a set of neurons $\boldsymbol{\psi}$ in the network and a corresponding set of values $\mathbf{c}_{\boldsymbol{\psi}}$, we denote the intervention by $f^{\boldsymbol{\psi}_{L+1}}_{\mathtt{N}}\big(\mathbf{x}, (\mathbf{h}_{\boldsymbol{\psi}} \leftarrow \mathbf{c}_{\boldsymbol{\psi}})\big)$. This notation means that, during the forward computation of the DNN, the activations of the neurons in $\boldsymbol{\psi}$ are fixed to the specified values $\mathbf{c}_{\boldsymbol{\psi}}$, while the rest of the network operates as usual.

## 3 Causal Abstraction

To define causal abstraction, we will base ourselves on the definitions in Beckers and Halpern (2019) and Geiger et al. (2024a). Let an **abstraction map** be defined as $\tau : \boldsymbol{\mathcal{H}} \to \boldsymbol{\mathcal{N}}$, where $\boldsymbol{\mathcal{H}}$ and $\boldsymbol{\mathcal{N}}$ are, respectively, the Cartesian products of the hidden state-spaces in a neural network $\mathtt{N}$ (i.e., $\boldsymbol{\psi}_{\mathtt{int}} \overset{\text{def}}{=} \boldsymbol{\psi}_{1:}$), and the node value-spaces in an algorithm $\mathtt{A}$ (i.e., $\boldsymbol{\eta}_{\mathtt{int}} \overset{\text{def}}{=} \boldsymbol{\eta}_{\mathtt{inner}} \cup \boldsymbol{\eta}_{\mathbf{y}}$), both excluding the inputs $\boldsymbol{\psi}_0$ and $\boldsymbol{\eta}_{\mathbf{x}}$. In words, an abstraction map translates the inner states of a neural network into an algorithms' inner states. Now, consider the DNN intervention $\mathbf{I}_{\mathtt{N}} = \mathbf{h}_{\boldsymbol{\psi}} \leftarrow \mathbf{c}_{\boldsymbol{\psi}}$ and the algorithm

[2] Geiger et al. (2024a) also considers cyclic deterministic causal models and Beckers and Halpern (2019) considers cyclic and stochastic causal models. We leave the expansion of our work to such models for future work.

intervention $\mathbf{I}_{\mathtt{A}} = \mathbf{v}_{\eta} \leftarrow \mathbf{c}_{\eta}$. Further, let $\mathcal{H}_{\mathbf{h}_{\psi}=\mathbf{c}_{\psi}}$ be the set of states in a DNN for which $\mathbf{h}_{\psi} = \mathbf{c}_{\psi}$ holds, and equivalently for $\mathcal{N}_{\mathbf{v}_{\eta}=\mathbf{c}_{\eta}}$. Under abstraction map $\tau$, we can define an intervention map as:

$$\omega_{\tau}(\mathbf{h}_{\psi} \leftarrow \mathbf{c}_{\psi}) = \begin{cases} \mathbf{v}_{\eta} \leftarrow \mathbf{c}_{\eta} & \texttt{if } \mathcal{N}_{\mathbf{v}_{\eta}=\mathbf{c}_{\eta}} = \{\tau(\mathbf{h}) \mid \mathbf{h} \in \mathcal{H}_{\mathbf{h}_{\psi}=\mathbf{c}_{\psi}}\} \\ \texttt{undefined} & \texttt{else} \end{cases} \tag{4}$$

Intuitively, $\omega_{\tau}$ maps a DNN intervention $\mathbf{I}_{\mathtt{N}}$ to an algorithmic one $\mathbf{I}_{\mathtt{A}}$ if the sets of states they induce on $\mathtt{N}$ and $\mathtt{A}$, respectively, are the same. Further, let $\mathcal{I}_{\mathtt{A}}$ be a set of interventions $\mathbf{I}_{\mathtt{A}}$ which can be performed on algorithm $\mathtt{A}$. We can use $\omega_{\tau}$ to derive a set of equivalent DNN interventions as:

$$\mathcal{I}_{\mathtt{N}} = \{\mathbf{h}_{\psi} \leftarrow \mathbf{c}_{\psi} \mid \mathbf{v}_{\eta} \leftarrow \mathbf{c}_{\eta} \in \mathcal{I}_{\mathtt{A}}, \omega_{\tau}(\mathbf{h}_{\psi} \leftarrow \mathbf{c}_{\psi}) = \mathbf{v}_{\eta} \leftarrow \mathbf{c}_{\eta}\} \tag{5}$$

Given these definitions, we now put forward a first notion of causal abstraction.

**Definition 1** (**from Beckers and Halpern, 2019**). *An algorithm $\mathtt{A}$ is a* ***$\tau$-abstraction*** *of a neural network $\mathtt{N}$ iff: $\tau$ is surjective; $\mathcal{I}_{\mathtt{A}} = \omega_{\tau}(\mathcal{I}_{\mathtt{N}})$;[3] and there exists a surjective $\tau_{\eta_{\mathbf{x}}}$ such that:*

$$\forall_{\substack{\mathbf{x}\in\mathcal{X} \\ \mathbf{I}_{\mathtt{N}}\in\mathcal{I}_{\mathtt{N}}}} : \tau(f_{\mathtt{N}}^{\psi_{\mathtt{int}}}(\mathbf{x}, \mathbf{I}_{\mathtt{N}})) = f_{\mathtt{A}}^{:\eta_{\mathtt{int}}}(\tau_{\eta_{\mathbf{x}}}(\mathbf{x}), \mathbf{I}_{\mathtt{A}}) \quad \texttt{where } \mathbf{I}_{\mathtt{A}} = \omega_{\tau}(\mathbf{I}_{\mathtt{N}}) \tag{6}$$

In words, the first condition in this definition enforces that all states in an algorithm are needed to abstract the DNN, while the second and third enforce that interventions in the algorithm have the same effect as interventions in the DNN. We further say that $\mathtt{A}$ is a **strong $\tau$-abstraction** of $\mathtt{N}$ if it is a $\tau$-abstraction and $\mathcal{I}_{\mathtt{A}}$ is maximal, meaning that any intervention is allowed on algorithm $\mathtt{A}$. However, while strong $\tau$-abstractions give us a notion of equivalence between algorithms and DNNs, the maps $\tau$ may be highly entangled and provide little intuition about the DNN's behaviour. To ensure algorithmic information is disentangled in the DNN, we say $\tau$ is a **constructive abstraction map** if there exists a partition of $\mathtt{N}$'s neurons $\{\psi_{\eta} \mid \eta \in \eta_{\mathtt{int}}\} \cup \{\psi_{\perp}\}$—where $\psi_{\eta}$ are non-empty—and there exist maps $\tau_{\eta}$ such that $\tau$ is equivalent to the block-wise application of $\tau_{\eta}(\mathbf{h}_{\psi_{\eta}})$. In other words, constructive abstraction maps compute the value $v_{\eta}$ of each node $\eta \in \eta_{\mathtt{int}}$ in $\mathtt{A}$ using non-overlapping sets of neurons $\psi_{\eta}$ from $\mathtt{N}$, with set $\psi_{\perp}$ being left unused. We now define a second notion of causal abstraction.

**Definition 2** (**from Beckers and Halpern, 2019**). *An algorithm $\mathtt{A}$ is a* ***constructive abstraction*** *of a neural network $\mathtt{N}$ iff there exists an $\tau$: for which $\mathtt{A}$ is a strong $\tau$-abstraction of $\mathtt{N}$; and $\tau$ is constructive.*

As we will deal with algorithm–DNN pairs which share the same input and output spaces, we will impose an additional constraint on $\tau$—one that is not present in Beckers and Halpern's (2019) definition. Namely, we restrict: $\psi_{\eta_{\mathbf{x}}}$ to be the neurons in layer zero and $\tau_{\eta_{\mathbf{x}}}$ to be the identity; and $\psi_{\eta_{\mathbf{y}}}$ to be the neurons in layer $L+1$ and $\tau_{\eta_{\mathbf{y}}}$ to be the $\operatorname{argmax}$ operation.[4]

### 3.1 Information Encoding in Neural Networks

The definition of constructive abstraction above maps non-overlapping sets of neurons in $\mathtt{N}$, i.e., $\psi_{\eta}$, to nodes in $\mathtt{A}$, i.e., $\eta$. However, much research in ML interpretability highlights that concept information is not always neuron-aligned and that neurons are often polysemantic (Olah et al., 2017, 2020; Arora et al., 2018; Elhage et al., 2022). In fact, there is a large debate about how information is encoded in DNNs. We highlight what we see as the three most prominent hypotheses here.

**Definition 3.** *The* ***privileged bases hypothesis*** *(Elhage et al., 2023) argues that neurons form privileged bases to encode information in a neural network.*

Most evidence in favour of this hypothesis comes from indirect evidence: i.e., the presence of neuron-aligned outlier features or activations in DNNs (Kovaleva et al., 2021; Elhage et al., 2023; He et al., 2024; Sun et al., 2024). Going back to 2015, Karpathy (2015) already showed that a single neuron in a language model could carry meaningful information. Importantly, this hypothesis is consistent with the notion of constructive abstraction above, as it argues each node $\eta$'s information should be encoded in separate, non-overlapping sets of neurons. Several researchers, however, question the special status of neurons assumed by this hypothesis, assuming instead that information is encoded in linear subspaces of the representation space, of which neurons are only a special case.

**Definition 4.** *The* ***linear representation hypothesis*** *(Alain and Bengio, 2016) argues that information is encoded in linear subspaces of a neural network.*

[3] We overload function $\omega_{\tau}$ here, with $\omega_{\tau}(\mathcal{I}_{\mathtt{N}})$ simply applying $\omega_{\tau}$ elementwise to the interventions in set $\mathcal{I}_{\mathtt{N}}$.
[4] We note that this implies algorithm $\mathtt{A}$ and network $\mathtt{N}$ must have the same outputs on the input set $\mathcal{X}$.

A large literature has developed, backed by the linear representation hypothesis, including: concept erasure methods (Ravfogel et al., 2020, 2022), probing methodologies (Elazar et al., 2021; Ravfogel et al., 2021; Lasri et al., 2022), and work on disentangling activations (Yun et al., 2021; Elhage et al., 2022; Huben et al., 2024; Templeton et al., 2024). Some, however, still question this idea that all information must be encoded linearly in DNNs: as neural networks implement non-linear functions, there is no a priori reason for why information should be linearly encoded in them (Conneau et al., 2018; Hewitt and Liang, 2019; Pimentel et al., 2020b,a, 2022). Further, recent research presents strong evidence that some concepts are indeed non-linearly encoded in DNNs (White et al., 2021; Pimentel et al., 2022; Olah and Jermyn, 2024; Csordás et al., 2024; Engels et al., 2025a,b; Kantamneni and Tegmark, 2025).

**Definition 5.** *The **non-linear representation hypothesis** (Pimentel et al., 2020b) argues that information may be encoded in arbitrary non-linear subspaces of a neural network.*

### 3.2 Distributed Causal Abstractions

Following the discussion above, the definition of constructive abstraction may be too strict, as it assumes $\tau$ must decompose across neurons—and thus that node information is encoded in non-overlapping neurons. With this in mind, Geiger et al. (2024a,b) proposed the notion of distributed interventions: they expose the subspaces where node information is encoded in a DNN $\mathtt{N}$ by applying a bijective function to its hidden states; this function's output is then itself a constructive abstraction of algorithm $\mathtt{A}$. Here, we make this notion a bit more formal.

We define $\tau$ as a **distributed abstraction map** if the following two conditions hold. First, there exists a bijective function $\phi$—termed here an **alignment map**—that maps the inner neurons $\boldsymbol{\psi}_{\mathtt{int}}$ of $\mathtt{N}$ block-wise to an equal-sized set of **latent variables** $\boldsymbol{\psi}^{\phi}_{\mathtt{int}}$, in a manner that respects the partial ordering of computations in the network. Specifically, for each layer $\ell$, there exists a bijection $\phi_\ell : \mathbb{R}^{|\boldsymbol{\psi}_\ell|} \to \mathbb{R}^{|\boldsymbol{\psi}_\ell|}$ on its neurons such that $\phi$ is defined as the concatenation of these layer-wise bijections. Similarly to the neurons' activation $\mathbf{h}_{\boldsymbol{\psi}}$, we will denote latent variables as $\mathbf{h}_{\boldsymbol{\psi}^\phi} = \phi(\mathbf{h}_{\boldsymbol{\psi}})$. Second, there exists a partition $\{\boldsymbol{\psi}^\phi_\eta \mid \eta \in \boldsymbol{\eta}_{\mathtt{int}}\} \cup \{\boldsymbol{\psi}^\phi_\perp\}$ of the resulting latent variables $\boldsymbol{\psi}^\phi_{\mathtt{int}}$—where $\boldsymbol{\psi}^\phi_\eta$ are non-empty—and a set of maps $\tau_\eta$ such that $\tau$ is equivalent to the block-wise application of $\tau_\eta(\mathbf{h}_{\boldsymbol{\psi}^\phi_\eta})$. In words, a distributed abstraction map computes the value $v_\eta$ of each node $\eta \in \boldsymbol{\eta}_{\mathtt{int}}$ in $\mathtt{A}$ using non-overlapping partitions of latent variables $\boldsymbol{\psi}^\phi_\eta$ from $\mathtt{N}$, with partition $\boldsymbol{\psi}^\phi_\perp$ remaining unused.

Given an alignment map $\phi$, we can perform distributed interventions: $\mathbf{h}_{\boldsymbol{\psi}^\phi_\eta} \leftarrow \mathbf{c}_{\boldsymbol{\psi}^\phi_\eta}$. These interventions are performed by first mapping the hidden state $\mathbf{h}_{\boldsymbol{\psi}}$ to the latent variables $\mathbf{h}_{\boldsymbol{\psi}^\phi} = \phi(\mathbf{h}_{\boldsymbol{\psi}})$, intervening on a subset $\boldsymbol{\psi}^\phi_\eta$ by replacing $\mathbf{h}_{\boldsymbol{\psi}^\phi_\eta}$ with desired values $\mathbf{c}_{\boldsymbol{\psi}^\phi_\eta}$, and then mapping these intervened latent variables back to the original neuron base via $\mathbf{h}'_{\boldsymbol{\psi}} = \phi^{-1}(\mathbf{h}'_{\boldsymbol{\psi}^\phi})$. Thus, interventions are applied in the latent space defined by $\phi$, generalising privileged-bases interventions to arbitrary (possibly non-linear) subspaces. We are now in a position to define distributed abstractions.

**Definition 6.** *An algorithm $\mathtt{A}$ is a **distributed abstraction** of a neural network $\mathtt{N}$ iff there exists an $\tau$: for which $\mathtt{A}$ is a strong $\boldsymbol{\tau}$-abstraction of $\mathtt{N}$; and $\tau$ is a distributed abstraction map.*

Finally, we note that the set of all possible interventions $\boldsymbol{\mathcal{I}}_{\mathtt{A}}$ and $\boldsymbol{\mathcal{I}}_{\mathtt{N}}$ may be hard to analyse in practice. Geiger et al. (2024b) thus restrict their analyses to what we term here **input-restricted interventions**: the set of interventions which are themselves producible by a set of other input-restricted interventions. In other words, we restrict interventions $\mathbf{h}_{\boldsymbol{\psi}'} \leftarrow \mathbf{c}_{\boldsymbol{\psi}'}$ (where $\boldsymbol{\psi}' \subseteq \boldsymbol{\psi}_{\mathtt{int}}$ or $\boldsymbol{\psi}' \subseteq \boldsymbol{\psi}^\phi_{\mathtt{int}}$) and $\mathbf{v}_{\boldsymbol{\eta}'} \leftarrow \mathbf{c}_{\boldsymbol{\eta}'}$ to $\mathbf{c}_{\boldsymbol{\psi}'}$ and $\mathbf{c}_{\boldsymbol{\eta}'}$ which are a product of other input-restricted interventions, e.g., $\mathbf{c}_{\boldsymbol{\psi}'} = f^{\boldsymbol{\psi}'}_{\mathtt{N}}(\mathbf{x}')$ or $\mathbf{c}_{\boldsymbol{\eta}'} = f^{:\boldsymbol{\eta}'}_{\mathtt{A}}(\mathbf{x}', (\mathbf{v}_{\boldsymbol{\eta}''} \leftarrow f^{:\boldsymbol{\eta}'}_{\mathtt{A}}(\mathbf{x}'')))$. This leads to the definition of **input-restricted $\boldsymbol{\tau}$-abstraction**: a weakened notion of strong $\tau$-abstraction, where intervention sets are restricted to input-restricted interventions. Finally, we define an analogous version of distributed abstraction, which is input-restricted; this is the notion typically used in practice by machine learning practitioners.

**Definition 7 (inspired by Geiger et al., 2024b).** *An algorithm $\mathtt{A}$ is an **input-restricted distributed abstraction** of a neural network $\mathtt{N}$ iff there exists an $\tau$: for which $\mathtt{A}$ is an input-restricted $\boldsymbol{\tau}$-abstraction of $\mathtt{N}$; and $\tau$ is a distributed abstraction map.*

A visual representation of how these causal abstraction definitions are related is given in App. D as Fig. 6. Finally, we further introduce **input-restricted $\boldsymbol{\mathcal{V}}$-abstractions**: input-restricted distributed abstractions for which we restrict alignment maps $\phi$ to be in a specific variational family $\mathcal{V}$. The case of linear alignment maps, will be particular important here—as it relates to the linear representation hypothesis—and we will thus explicitly label it as **input-restricted linear abstraction**.

### 3.3 Finding Distributed Abstractions

How do we evaluate if an algorithm is an input-restricted distributed abstraction of a DNN? Geiger et al. (2024b) proposes an efficient method to answer this, called distributed alignment search (DAS). Before applying DAS, one must assume a partitioning $\{\psi_\eta^\phi \mid \eta \in \boldsymbol{\eta}_{\mathtt{int}}\} \cup \{\psi_\perp^\phi\}$ which remains fixed during the method's application; we term $|\psi_\eta^\phi|$ the **intervention size**. The principle behind DAS is then to leverage the constraint on $\tau_{\boldsymbol{\eta}_\mathbf{y}}$, which is fixed as: $\mathbf{v}_{\boldsymbol{\eta}_\mathbf{y}} = \mathrm{argmax}_{\mathbf{y}\in\mathcal{Y}}\, p_{\mathtt{N}}(\mathbf{y} \mid \mathbf{x})$ since $p_{\mathtt{N}}(\mathbf{y} \mid \mathbf{x}) = \mathbf{h}_{\psi_{L+1}}$. Given this constraint, we can initialise a parametrised function $\phi$, which we train to predict this equality under possible interventions; this is done via gradient descent, minimising the cross-entropy between the DNN and the algorithm. Specifically, we first select a set of nodes to be intervened $\overline{\boldsymbol{\eta}} \in \mathcal{P}(\boldsymbol{\eta}_{\mathtt{inner}})$, where $\mathcal{P}$ is a function that takes the powerset of a set, along with corresponding counterfactual inputs $\mathbf{x}_\eta \in \mathcal{X}$ for each $\eta \in \overline{\boldsymbol{\eta}}$ and a base input $\mathbf{x}_\emptyset \in \mathcal{X}$. We then define the following two interventions:

$$\mathbf{I}_{\mathtt{N}} = \left[\mathbf{h}_{\psi_\eta^\phi}\right]_{\eta\in\overline{\boldsymbol{\eta}}} \leftarrow \left[f_{\mathtt{N}}^{\psi_\eta^\phi}(\mathbf{x}_\eta)\right]_{\eta\in\overline{\boldsymbol{\eta}}} \quad \text{and} \quad \mathbf{I}_{\mathtt{A}} = \left[\mathbf{v}_\eta\right]_{\eta\in\overline{\boldsymbol{\eta}}} \leftarrow \left[f_{\mathtt{A}}^{:\eta}(\mathbf{x}_\eta)\right]_{\eta\in\overline{\boldsymbol{\eta}}} \tag{7}$$

Finally, we run our algorithm under base input $\mathbf{x}_\emptyset$ and intervention $\mathbf{I}_{\mathtt{A}}$ to get a ground truth output: $\mathbf{y} = f_{\mathtt{A}}(\mathbf{x}_\emptyset, \mathbf{I}_{\mathtt{A}})$. Repeating this process $N$ times, we build a dataset $\mathcal{D} = \{(\mathbf{x}_\emptyset^{(n)}, \mathbf{I}_{\mathtt{N}}^{(n)}, \mathbf{y}^{(n)})\}_{n=1}^N$ on which we can train the alignment map $\phi$ such that the DNN matches the algorithm:

$$\mathcal{L} = -\sum_{(\mathbf{x}_\emptyset^{(n)}, \mathbf{I}_{\mathtt{N}}^{(n)}, \mathbf{y}^{(n)})\in\mathcal{D}} \log p_{\mathtt{N}}^\phi(\mathbf{y}^{(n)} \mid \mathbf{x}_\emptyset^{(n)}, \mathbf{I}_{\mathtt{N}}^{(n)}), \quad \texttt{where} \quad p_{\mathtt{N}}^\phi(\mathbf{y} \mid \mathbf{x}_\emptyset, \mathbf{I}_{\mathtt{N}}) = f_{\mathtt{N}}(\mathbf{x}_\emptyset, \mathbf{I}_{\mathtt{N}}) \tag{8}$$

Notably, DAS mostly ignores how function $\tau$ is constructed, relying solely on the assumed definition of $\tau_{\boldsymbol{\eta}_\mathbf{y}}$. Finding a low-loss alignment map $\phi$ is then assumed as sufficient evidence that `A` is an input-restricted distributed abstraction of `N`.

## 4 Unbounded Abstractions are Vacuous

In this section, we provide our main theorem: that under reasonable assumptions, *any* algorithm `A` can be shown to be an input-restricted distributed abstraction of *any* DNN `N`, making this notion of causal abstraction vacuous. To show that, we need a few assumptions (for their formal definition, see App. F). Our first assumption (Assump. 1) is that we have a **countable input-space** $\mathcal{X}$. While this may not hold in general, it holds for common applications such as language modelling (where the input-space is the countably infinite set of finite strings) or computer vision (where the input-space is a countable union of pixels, which can assume a finite set of values). The second assumption (Assump. 2) is that DNNs are **input-injective in all layers**: i.e., $f_{\mathtt{N}}^{\psi_\ell}$ is injective for all layers. This guarantees that no information about a DNN's input $\mathbf{x}$ is lost when computing the hidden states $\mathbf{h}_{\psi_\ell}$. This assumption is also present in prior work (e.g., Pimentel et al., 2020b) and we show in App. G—assuming real-valued weights and activations—that this is almost surely true for transformers at initialisation.[5] Due to floating point precision and neural collapse (Papyan et al., 2020), it is likely not to hold fully in practice; however, it still seems to be well-approximated in many empirical settings (Morris et al., 2023, and App. H). The third assumption (Assump. 3) is **strict output-surjectivity in all layers**. This assumption guarantees that in each layer there is at least one choice of $\mathbf{h}_{\psi_\ell}$ that will produce the desired output. Notably, this assumption may not hold in theory, due to issues like the softmax-bottleneck (Yang et al., 2018). In practice, however, even with large vocabulary sizes, it seems that almost all outputs can still be produced by language models (Grivas et al., 2022) which is sufficient for these DNNs to be abstracted by many algorithms. Our fourth assumption (Assump. 4) is that the **algorithm `A` and DNN `N` have matchable partial-orderings**, meaning that there is a partitioning of neurons in `N` which would match the partial-ordering of nodes in `A`; this is likely to be the case for most reasonable algorithms given the size of state-of-the-art deep neural networks. Finally, our last assumption (Assump. 5) is that the **DNN `N` solves the given task** `T`. We believe this assumption to be reasonable, as it would be impractical in practice to evaluate a neural network that does not perform the task correctly.[6] Given these assumptions, we can now present our main theorem.

**Theorem 1.** *Given any algorithm `A` and any neural network `N` such that Assumps. 1 to 5 hold, we can show that `A` is an input-restricted distributed abstraction of `N`.*

*Proof.* We refer to App. F for the proof. □

[5] Also see Nikolaou et al. (2025), who show almost sure injectivity holds for transformers throughout training.

[6] If the model does not solve the task, perfect IIA is impossible since non-intervened inputs yield incorrect outputs. Thus, assuming the model solves the task is necessary. In practice, however, even when the DNN is imperfect, an alignment map could produce correct outputs for all intervened inputs, achieving near-perfect IIA scores.

## 5 Experimental Setup

Building on the previous section's proof that alignment maps between DNNs and algorithms always exist, we now demonstrate their practical learnability and how increasingly complex alignment maps reveal various causal abstractions for different tasks, even on DNNs that do not solve them.

**Alignment Maps.** To assess how complexity impacts causal-abstraction analyses, we explore three ways to parameterise $\phi$. First, we will consider the simplest **identity maps**: $\phi^{\texttt{id}}(\mathbf{h}) = \mathbf{h}$. This is the least expressive $\phi$ we consider, and if we find that $\texttt{A}$ abstracts $\texttt{N}$ under this map, we can say that $\texttt{A}$ is a constructive abstraction of $\texttt{N}$; further, this map implicitly assumes the privileged bases hypothesis. For $\phi^{\texttt{id}}$, we greedily search for the optimal partition $\{\psi^{\phi}_{\eta} \mid \eta \in \boldsymbol{\eta}_{\texttt{int}}\}$ (instead of keeping it fixed) by iteratively adding neurons to them. For all $\boldsymbol{\eta}_{\texttt{inner}}$ simultaneously, one neuron is added at a time for each $\psi^{\phi}_{\eta}$, up to a maximum allowed intervention size; these neurons are chosen to minimise the loss in eq. (8). Second, we will consider **linear maps**: $\phi^{\texttt{lin}}(\mathbf{h}) = \mathbf{W}_{\texttt{orth}}\mathbf{h}$, where $\mathbf{W}_{\texttt{orth}} \in \mathbb{R}^{d_\ell \times d_\ell}$ is an orthogonal matrix. This is the type of alignment map originally considered by Geiger et al. (2024b),[7] and implicitly assumes the linear representation hypothesis, evaluating input-restricted linear abstractions. Finally, we consider **non-linear maps**: $\phi^{\texttt{nonlin}}(\mathbf{h}) = \texttt{revnet}[L_{\text{rn}}, d_{\text{rn}}](\mathbf{h})$, where $\texttt{revnet}[L_{\text{rn}}, d_{\text{rn}}]$ is a reversible residual network (RevNet; Gomez et al., 2017) with $L_{\text{rn}}$ layers and hidden size $d_{\text{rn}}$. We can modulate the complexity of this final map by increasing $L_{\text{rn}}$ and $d_{\text{rn}}$, assuming the non-linear representation hypothesis. We note that all three maps are bijective and easily invertible.

**Evaluation Metric.** We evaluate the effectiveness of an alignment map $\phi$ using the **interchange intervention accuracy** (IIA) metric proposed by Geiger et al. (2024b). For a held out test set $\mathcal{D}_{\texttt{test}}$ with the same structure as the training set $\mathcal{D}$ defined in §3.3, we compute the accuracy of our model (i.e., $\text{argmax}_{\mathbf{y}' \in \mathcal{Y}}\, p^{\phi}_{\texttt{N}}(\mathbf{y}' \mid \mathbf{x}^{(n)}_{\emptyset}, \mathbf{I}^{(n)}_{\texttt{N}})$) when predicting the intervened $\mathbf{y}^{(n)} = f_{\texttt{A}}(\mathbf{x}^{(n)}_{\emptyset}, \mathbf{I}^{(n)}_{\texttt{A}})$. We compare this to the DNN's accuracy on the test set $\mathcal{D}_{\texttt{test}}$ without interventions.

### 5.1 Tasks, Algorithms, and DNNs.

**Hierarchical equality task (Geiger et al. (2024b)).** We will showcase our results primarily on this task. Let $\mathbf{x} = \mathbf{x}_1 \circ \mathbf{x}_2 \circ \mathbf{x}_3 \circ \mathbf{x}_4$ be a 16-dimensional vector, and $\mathbf{x}_1$ to $\mathbf{x}_4$ each be 4-dimensional vectors, where $\circ$ represents vector concatenation. Further, let $\mathcal{X} = [-.5, .5]^{16}$. This task consists of evaluating: $\mathbf{y} = (\mathbf{x}_1 == \mathbf{x}_2) == (\mathbf{x}_3 == \mathbf{x}_4)$. As our DNN $\texttt{N}$, we investigate a 3-layer multi-layer perceptron (MLP) with hidden size 16, trained to perform this task; we describe this DNN, and its training procedure in more detail in App. I.1. Finally, we explore three algorithms for this task. The `both equality relations` algorithm first computes the two equalities ($v_{\eta_1} = (\mathbf{x}_1 == \mathbf{x}_2)$ and $v_{\eta_2} = (\mathbf{x}_3 == \mathbf{x}_4)$) separately; it then determines whether they are equivalent as a second step. The `left equality relation` algorithm first computes the left equality ($v_{\eta_1} = (\mathbf{x}_1 == \mathbf{x}_2)$), and then determines in a single step if this is equivalent to $(\mathbf{x}_3 == \mathbf{x}_4)$. Finally, the `identity of first argument` algorithm assumes we copy the first input to a node ($v_{\eta_1} = \mathbf{x}_1$) and then compute the output directly. These three algorithms are more rigorously defined in App. I.1.

**Indirect object identification (IOI) task.** In a second set of experiments, we explore this task, inspired by Wang et al. (2023) and using the dataset of Muhia (2022). This task is more realistic and relies on larger (language) models. Here, inputs $\mathbf{x} \in \mathcal{X}$ are strings where two people are first introduced, and later one of them assumes the role of subject (`S`), giving or saying something to the other, the indirect object (`IO`). The task is then to predict the first token of the `IO`, with the output set $\mathcal{Y}$ containing the first token of each person's name. E.g., $\mathbf{x} =$ "`Friends Juana and Kristi found a mango at the bar. Kristi gave it to`" and $\mathbf{y} =$ "`Juana`". As our DNN, we use models from the Pythia suite (Biderman et al., 2023) across different sizes (from 31M to 410M parameters) and training stages. We evaluate the `ABAB-ABBA` algorithm where, given two names A and B, an inner node $v_{\eta_1}$ captures if the sentence structure is ABAB (e.g., "A and B ... A gave to B") or ABBA (e.g., "A and B ... B gave to A"), and the algorithm outputs prediction B if $v_{\eta_1}$ is ABAB and A otherwise. This algorithm is more rigorously defined in App. I.2.

## 6 Experiments and Results

We now proceed with our empirical study, applying alignment maps of varying complexity on both "toy" and real neural networks to evaluate their effects on the causal abstraction method DAS.

[7] We note that, while Geiger et al. (2024b) describe the used $\phi$ as a rotation, their pyvene (Wu et al., 2024a) implementation uses orthogonal matrices. This, however, makes no difference in the power of the alignment map.

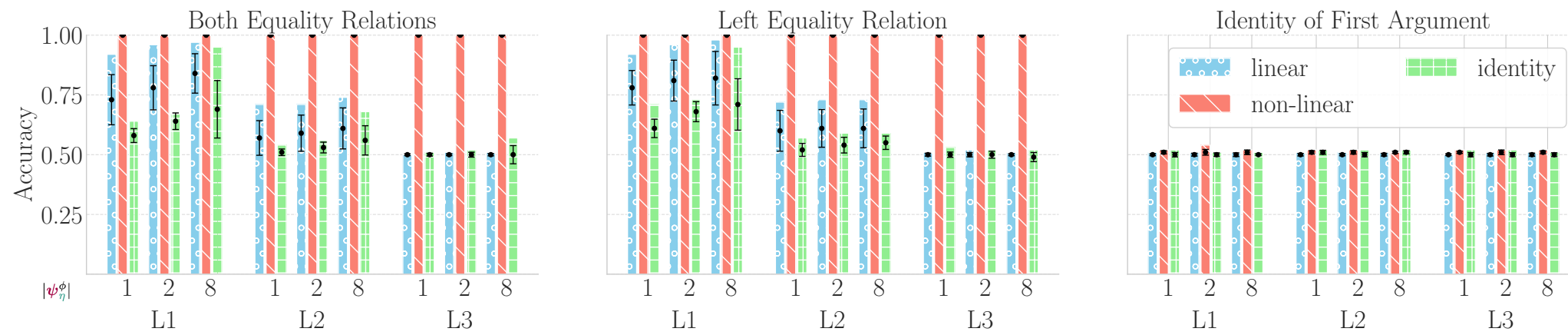

Figure 2: IIA in the hierarchical equality task for causal abstractions trained with different alignment maps $\phi$. The figure shows results for all three analysed algorithms for this task. The bars represent the max IIA across 10 runs with different random seeds. The black lines represent mean IIA with 95% confidence intervals. The $|\psi_\eta^\phi|$ denotes the intervention size per node. Without interventions, all DNNs reach almost perfect accuracy (>0.99). The used $\phi^{\texttt{nonlin}}$ uses $L_{\text{rn}} = 10$ and $d_{\text{rn}} = 16$.

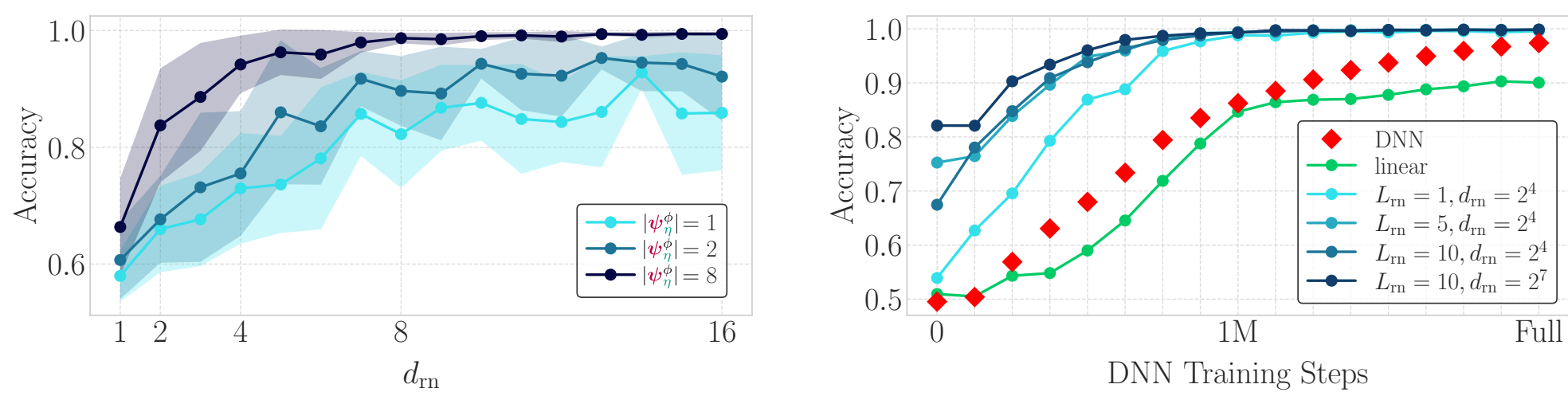

Figure 3: IIA of alignment between the `both equality relations` algorithm and an MLP, with interventions at layer 1. Left: Mean IIA over 5 seeds using $\phi^{\texttt{nonlin}}$ ($L_{\text{rn}} = 1$) on the trained DNN. Performance improves with larger hidden dimension $d_{\text{rn}}$ and intervention size $|\psi_\eta^\phi|$. Right: Maximum IIA across 5 seeds using $\phi^{\texttt{lin}}$ and $\phi^{\texttt{nonlin}}$ with $|\psi_\eta^\phi| = 8$. Complex alignment maps achieve high IIA even with randomly initialised DNNs, while simpler maps gradually improve as training progresses.

**Hierarchical equality task, main results.**[8] Fig. 2 presents IIA results across different alignment maps $\phi$ for all three algorithms. As expected, the identity map $\phi^{\texttt{id}}$ generally results in the worst performance. Using linear alignments ($\phi^{\texttt{lin}}$), we observe patterns consistent with Geiger et al. (2024b): IIA for `both equality relations` and `left equality relation` decreases substantially in the third layer, indicating information becomes difficult to manipulate using linear transformations at deeper layers. With the non-linear alignment ($\phi^{\texttt{nonlin}}$), this layer-dependent degradation vanishes, yielding near-optimal IIA across all layers. Consequently, while assuming linear representations seems to enable us to identify the location of certain variables in our DNN, many of these insights fail to generalise when more powerful non-linear alignment maps are employed. The `identity of first argument` algorithm's IIA consistently hovers around 50% for $\phi^{\texttt{id}}$, $\phi^{\texttt{lin}}$ and $\phi^{\texttt{nonlin}}$. Additional experiments (App. H) suggest this is caused by insufficient capacity of the used `revnet` model, as the identity of $\mathbf{x}_1$ seems to be encoded in the model's hidden states.

**Hierarchical equality task, exploring $\phi^{\texttt{nonlin}}$'s complexity.** Fig. 3 (left) illustrates how varying the hidden size $d_{\text{rn}}$ and intervention size $|\psi_\eta^\phi|$ affects IIA with the `both equality relations` algorithm on layer 1 of our MLP. Fig. 3 (right) shows IIA evolution as alignment complexity increases throughout the MLP's training (evaluated on its layer 1). Remarkably, even with randomly initialised DNNs, we achieve over 80% IIA using the most complex alignment map. As training progresses, simpler alignment maps gradually attain higher IIA values. Additional results in App. I.1.3 extend these findings to other MLP layers, intervention sizes, and algorithms, consistently revealing similar patterns that reinforce our conclusion about the impact of alignment map complexity on IIA dynamics.

**Indirect object identification task, main results.** Fig. 4 (left) presents the results of trying to find causal abstractions between the `ABAB-ABBA` algorithm and Pythia language models, exploring how model size affects alignment capabilities. Notably, despite only the larger models (160M and 410M parameters) successfully learning the IOI task, we can align the algorithm to models of all sizes—including the 31M and 70M parameter models that fail to learn the task. Further, and somewhat surprisingly, this alignment is perfect even for randomly initialised models across all sizes; smaller fully trained models (31M, 70M), though, show slightly reduced alignment accuracy. This reduction

[8] As an additional task similar to hierarchical equality, we also explore the **distributive law task** in App. I.3.

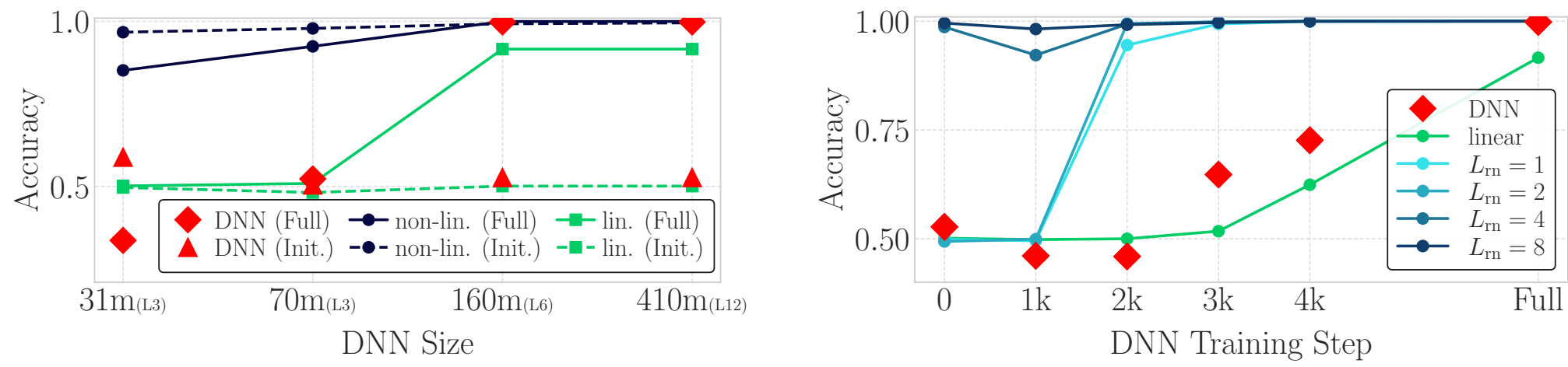

Figure 4: IIA of alignment between ABAB-ABBA algorithm and Pythia language models. Left: IIA across model sizes at initialisation (Init.) or after full training (Full), with intervention at the middle layer. Right: IIA with increasingly complex alignment maps during *Pythia-410m*'s training. Results show complex alignment maps yield near-perfect IIA. All $\phi^{\texttt{nonlin}}$ use $d_{\text{rn}} = 64$.

may stem from these smaller models saturating late in training (Godey et al., 2024), becoming highly anisotropic and making it harder for $\phi^{\texttt{nonlin}}$ to access the information needed to match the algorithm.

**Indirect object identification task, exploring $\phi^{\texttt{nonlin}}$'s complexity.** Fig. 4 (right) illustrates the interplay between model training progression and algorithmic alignment for the Pythia with 410M parameters. Notably, while this model begins to acquire task proficiency only around training step 3000 (as indicated by model accuracy), employing an 8-layer $\phi^{\texttt{nonlin}}$ as alignment map yields near-perfect IIA across all training steps, including for randomly initialised models. This pattern partially extends to a 4-layers $\phi^{\texttt{nonlin}}$ configuration; however, there is a noticeable dip in IIA at step 1000 for this configuration, which may be due to the model over-fitting to unigram statistics (Chang and Bergen, 2022; Belrose et al., 2024) at this point—thereby making context (and hidden states) be mostly ignored when producing model outputs. Interestingly, as training advances, even less complex alignment maps (1- and 2-layer $\phi^{\texttt{nonlin}}$) eventually attain perfect alignment. In contrast, linear maps only approximate perfect alignment in the fully trained model, following a similar trend to the DNN's performance.

## 7 Discussion

Our results show that when we lift the assumption of linear representations, sufficiently complex alignment maps can achieve near-perfect alignment across all models—regardless of their ability to solve the underlying task. This provides compelling evidence for the non-linear representation dilemma, suggesting causal alignment may be possible even when the model lacks task capability. We now discuss our results in the context of prior literature, with additional related work in App. C.

**Causal Abstraction is not Enough.** Causal abstraction (Geiger et al., 2024a) has gained traction as a theoretical framework for mechanistic interpretability, promising to overcome probing limitations by analysing DNN behaviour through interventions: if you intervene on a DNN's representations and its behaviour changes in a predictable way, you have identified how the DNN "truly" encodes that feature (Elazar et al., 2021; Ravfogel et al., 2021; Lasri et al., 2022). Recent critiques of causal abstraction (e.g., Mueller, 2024) highlight practical shortcomings, including the non-uniqueness of identified algorithms (Méloux et al., 2025) and the risk of "interpretability illusions" (Makelov et al., 2024). Despite counterarguments to some of these critiques (Wu et al., 2024b; Jørgensen et al., 2025), concerns have emerged that methods based on causal abstraction may introduce new information rather than accurately reflect the behaviour of the DNN (Wu et al., 2023; Sun et al., 2025); as an example, causal abstraction methods applied to random models sometimes yield above-chance performance (Geiger et al., 2024b; Arora et al., 2024). By examining the implications of assuming arbitrary complex ways in which features may be encoded in a DNN, we show that nearly any neural network can be aligned to any algorithm. Together, our results thus suggest that the shift in interpretability research to causal abstractions does not, by itself, resolve the core challenge of understanding how representations are encoded. Additionally, we note that early causal abstraction methods (Geiger et al., 2021) implicitly rely on the privileged bases hypothesis, while recent advancements (Geiger et al., 2024b) rely on the linear representation hypothesis instead.

**Balancing the Accuracy vs. Complexity of $\phi$.** Diagnostic probing was a previously popular method for interpretability research (Alain and Bengio, 2016), where a **probe** was applied to the hidden representations of a DNN and trained to predict a specific variable. Notably, the architecture chosen for this probe implicitly reflected assumptions about representation encoding, and the absence of a universally accepted model for representation encoding precluded a theoretically founded

choice of probe architecture (Belinkov, 2022). The debate regarding the trade-off between probing complexity and accuracy (Hewitt and Liang, 2019; Pimentel et al., 2020b,a; Voita and Titov, 2020) underscores the risk of complex probes merely memorising variable-specific relations, instead of revealing which information the DNN "truly" encodes and uses. In this paper, we revive this debate by showing a clear analogue in causal abstraction methodologies: the effect of $\phi$'s complexity on IIA. Unfortunately, this debate was never solved by the probing literature, and solutions ranged from: controlling for the probe's memorisation capacity (Hewitt and Liang, 2019),[9] explicitly measuring a probe's complexity accuracy trade-off (Pimentel et al., 2020a), training minimum description length probes (Voita and Titov, 2020), or leveraging unsupervised probes (Burns et al., 2023).[10]

**The Role of Generalisation.** We now highlight that Theorem 1 provides an existence proof for a perfect abstraction map (thus guaranteeing perfect IIA) between a DNN and an algorithm. This existence proof, however, leverages complex interactions between the intervened hidden states and the DNN's structure, requiring perfect information about both and thus representing a form of extreme overfitting. Crucially, this theorem offers no guarantees regarding the learnability of the alignment map $\phi$ from limited data or its generalisation to unseen inputs. This gap between theoretical existence and practical learnability becomes evident in practise. For instance, in an additional experiment on the IOI task (in App. I.2.3), we show that when training and test sets contain disjoint sets of names, the learned alignment map fails to generalise, resulting in low IIA on the test set. This suggests that generalisation should play a crucial role in causal abstraction analysis, as the ability to learn abstraction maps that transfer beyond training data seems fundamental to interpreting a model, distinguishing a genuine understanding about its inner workings from mere training pattern memorisation.

**Investigating Representation Encoding in DNNs.** How neural networks encode variables/concepts is a long-standing question in interpretability, with three main hypotheses standing out: the privileged bases, linear representation, and non-linear representation hypotheses (see §3.1). One way to try to distinguish between these hypotheses is with causal abstraction analyses, but what can we learn about these hypotheses if our methods themselves rely on them as assumptions? One solution could be to compare results using $\phi$ with different architectures. Our Fig. 4 (right), for instance, shows that while $\phi^{\texttt{nonlin}}$ achieves consistently near-perfect results throughout model training, $\phi^{\texttt{lin}}$ accompanies the actual DNN's performance more closely. Intuitively, we may thus be inclined to support the linear representation hypothesis here. We (the authors), however, cannot make this intuition formal to justify why we believe this is the case. Furthermore, $\phi^{\texttt{lin}}$ still manages to sometimes achieve IIA higher than the DNN's accuracy, implying it may also "learn the task". We expect future work will propose novel methodologies to analyse information encoding and try to answer these questions.

## 8 Conclusion

This paper critically examines causal abstraction in machine learning, when no assumptions are imposed on how representations are encoded. We show that, under mild conditions, any algorithm can be perfectly aligned with any DNN, leading to the non-linear representation dilemma. Empirical validation through experiments on the hierarchical equality and the indirect object identification tasks corroborate our theoretical insights, demonstrating near perfect IIA even in randomly initialised DNNs. So, *what should you do if you want to perform a causal analysis of your DNN?* We believe that it must be decided on a case-by-case basis. If you have reason to believe the linear representation hypothesis holds for the features you wish to extract, constraining $\phi$ to linear functions may be advised. If you do not, however, you may face the non-linear representation dilemma, and be forced to investigate some kind of trade-off between $\phi$'s accuracy and complexity.

**Limitations.** Our proof that any algorithm can be aligned with any DNN (Theorem 1) relies on a form of overfitting. Yet, our experiments show that the learned alignment maps $\phi$ generalise to unseen test data; studying the factors behind this generalisation would be valuable. Further, our theorem relies on two strong assumptions: input-injectivity (Assump. 2) and strict output-surjectivity (Assump. 3) in all layers. While we justify both, there are settings—related to, e.g., the softmax bottleneck—where they may fail; studying these failure modes could clarify our assumptions' limitations.

[9]Notably, this method was previously applied to causal abstraction analysis by Arora et al. (2024).

[10]The complexity—accuracy trade-off in probing arises mainly in supervised settings, where more complex probes can extract richer features from model representations. Unsupervised probing avoids this, lacking the supervision that enables such "gerrymandered" mappings.

## Contributions

Denis Sutter led the project, implemented the base version of the DAS code, conducted the MLP experiments and derived the base proof of Theorem 1 as well as the proof of Theorem 2. Julian Minder implemented and ran the language model experiments, produced all plots, and helped refine the proof of Theorem 2. Thomas Hofmann provided guidance throughout the project. Tiago Pimentel supervised the project, giving initial intuitions for the proofs in both Theorem 1 and Theorem 2, refining the proof of Theorem 1, and defining the main notation in the paper, integrating feedback from Denis and Julian. All authors wrote the paper together.

## Acknowledgments

This work was mostly done in the Data Analytics Lab at ETH Zürich. We would like to thank Pietro Lesci, Julius Cheng, Marius Mosbach, Chris Potts, and Atticus Geiger for their thoughtful feedback. We would also like to thank Frederik Hytting Jørgensen for bringing to our attention a mistake in our original Definition 1 and for his feedback on our manuscript. We thank Zhengxuan Wu and Kevin Du for early discussions related to the ideas presented here. We are grateful to the Data Analytics Lab at ETH for providing access to their computing cluster. Julian Minder is supported by the ML Alignment Theory Scholars (MATS) program. Denis Sutter gratefully acknowledges the financial support of his parents, Renate and Wendelin Sutter, throughout his graduate studies, during which this work was carried out, as well as the technical support of Urban Moser and Leo Schefer.

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

## A Reproducibility

We provide the code to reproduce our experiments in `https://github.com/densutter/non-linear-representation-dilemma`. Refer to the `README.md` for instructions.

## B Pseudo-code for Running an Intervention on an Algorithm

In Fig. 5, we present pseudo-code that demonstrates how algorithms execute under our intervention framework.

```
def f(A):
    def f_A(x, I_A=None):
        v_{η_x} = x
        for η in A.topological_sort(η_inner ∪ η_y):
            if I_A and η in I_A:
                v_η = I_A[η]
            else:
                v_η = f_A^η(v_{par_A(η)})
        return v_{η_y}
    return f_A
```

Figure 5: Pseudo-code implementation of an algorithm with interventions, where interventions $\mathbf{I}_{\mathtt{A}}$ are specified as a Python dictionary mapping nodes to their intervened values.

## C Additional Related Work

The concept of causal abstraction was ported to deep neural networks by Geiger et al. (2024a), providing a generalised framework for understanding how neural networks can be abstracted to higher-level algorithms. Early work by Geiger et al. (2021) explored direct interventions on neuron-aligned activations, laying the groundwork for more sophisticated approaches. Building on this, Geiger et al. (2024b) introduced distributed alignment search (DAS), which uses an alignment map to align distributed representations in neural networks with causal graphs. Several improvements to DAS have been proposed: Sun et al. (2025) developed HyperDAS, which automates the search for node information using hypernetworks, while Wu et al. (2023) introduced Boundless DAS, which automatically determines intervention size through gradient descent, scaling to larger models.

However, recent work has raised important critiques of causal alignment methods. Méloux et al. (2025) demonstrated that multiple algorithms can be causally aligned with the same neural network, and conversely, a single algorithm can align with different network subspaces. Mueller (2024) identified fundamental limitations in counterfactual theories, showing they may miss certain causes and that causal dependencies in neural networks are not necessarily transitive. Makelov et al. (2024) showed that subspace interventions such as those used in DAS can lead to "interpretability illusions"—cases where manipulating a subspace changes the behaviour of the model through activating parallel pathways, rather than directly controlling the target feature. In their response, Wu et al. (2024b) argued these illusions may be artefacts of specific evaluation approaches rather than fundamental flaws, and that they depend on the definition of causality being used, a point also made by Jørgensen et al. (2025).

Recent work has also raised significant challenges to the linear representation hypothesis. White et al. (2021) demonstrated that syntactic structure in language models is encoded non-linearly, showing that kernelised structural probes outperform linear ones while maintaining parameter count. Similarly, Csordás et al. (2024) found that recurrent neural networks use fundamentally non-linear representations for sequence tasks. Engels et al. (2025a) provided concrete examples of non-linear feature representations in language models, such as days of the week being encoded on a circular manifold. While Golechha and Dao (2024) argued that some language modelling behaviours may be represented linearly due to next-token prediction and LayerNorm folding, Mueller et al. (2024) advocated for exploring non-linear mediators to uncover more sophisticated abstractions. Additional evidence comes from Kantamneni and Tegmark (2025), who found that language models represent numbers on a helical manifold. Olah and Jermyn (2024) offered an important clarification: the linear representation hypothesis is not about dimensionality but rather about features behaving mathematically linearly through addition and scaling, allowing for multidimensional features with constrained geometry. This represents a relaxation of the strongest form of the hypothesis.

# D Schematic of the Relation Between Notions of Causal Abstraction

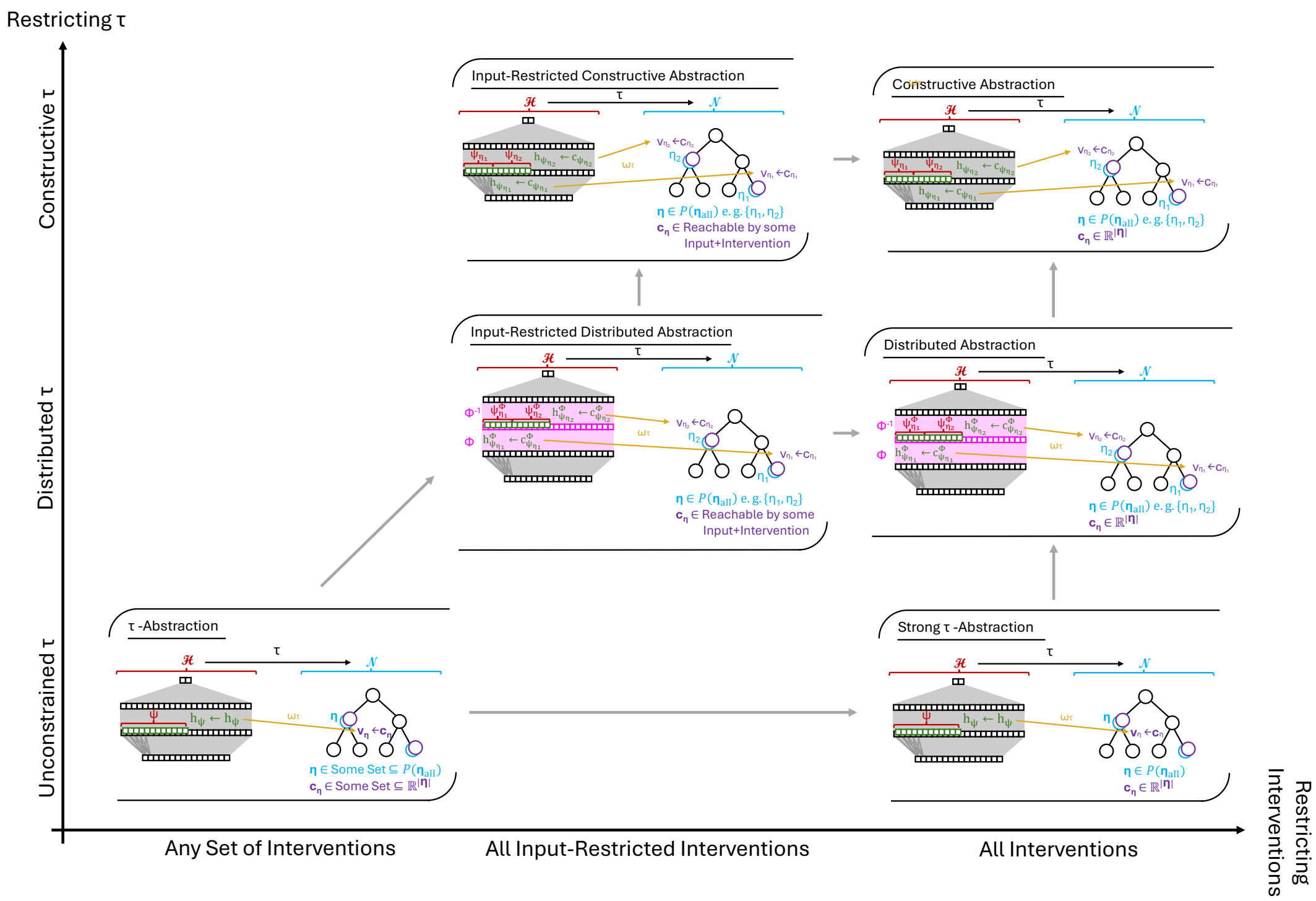

Figure 6: A schematic of the definitions of causal abstraction in §3. The axes represent an increase in how restricted the notion of causal abstraction is based on: $y$-axis, constraints placed on $\tau$; and $x$-axis, constraints placed on the set of allowed interventions. Grey arrows symbolise a superset→subset relationship: if an A-N pair fulfils the conditions in the subset, it also fulfils them in the superset.

# E DNN Definitions

## E.1 MLP

A multi-layer perceptron (MLP) consists of a sequence of linear transformations interleaved with non-linear activation functions.

**Submodule 1.** *We can define a **multi-layer perceptron** (`mlp`) by choosing:*

$$\mathbf{h}_{\psi_1} = f^0_{\mathtt{N}}(\mathbf{x}) = \mathbf{W}_0\,\mathbf{x}, \qquad \mathbf{h}_{\psi_{\ell+1}} = f^\ell_{\mathtt{N}}(\mathbf{h}_{\psi_\ell}) = \mathbf{W}_\ell\,(\sigma(\mathbf{h}_{\psi_\ell})) + \mathbf{b}_\ell \tag{9}$$

*where $\mathbf{W}_0 \in \mathbb{R}^{|\psi_1|\times|\mathbf{x}|}$, $\mathbf{W}_\ell \in \mathbb{R}^{|\psi_{\ell+1}|\times|\psi_\ell|}$, and $\mathbf{b}_\ell \in \mathbb{R}^{|\psi_{\ell+1}|}$ are trainable parameters, and $\sigma$ is a non-linearity like ReLU. For this model, $\mathcal{H}_\ell = \mathbb{R}^{|\psi_\ell|}$ for $0 < \ell < L$ and $\mathcal{H}_L = \mathbb{R}^{|\mathbf{y}|}$.*

In this work, we focus on MLPs used for classification tasks, whose final layer includes a softmax transformation.

**DNN 1.** *A **classification multi-layer perceptron** (MLP) is defined like Submodule 1 but with a softmax on the last layer:*

$$p_{\mathtt{N}}(\mathbf{y} \mid \mathbf{x}) = f^L_{\mathtt{N}}(\mathbf{h}_{\psi_L}) = \mathtt{softmax}(\mathbf{W}_L\,\mathbf{h}_{\psi_L}) \tag{10}$$

*where $\mathbf{W}_L \in \mathbb{R}^{|\mathbf{y}|\times|\psi_L|}$ is a trainable parameter.*

## E.2 Transformer Language Model

In this section, we provide a definition of decoder-only autoregressive language models (Radford et al., 2019; Vaswani et al., 2017). While many variations of transformer architectures have been developed, we focus on the original GPT-2 architecture (Radford et al., 2019). We highlight that the Pythia models explored in our experiments are slightly different from the original GPT-2 and use parallel attention (Chowdhery et al., 2023); however, we do not expect this change to strongly affect our results. We now define the different submodules that compose a transformer.

**Submodule 2.** *The* ***Embedding*** *layer in a transformer maps input tokens to vectors:*

$$f_{\mathtt{N}}^{0}(\mathbf{x}) = \mathbf{e}_{\mathbf{x}}, \qquad \texttt{where } \mathbf{e}_{\mathbf{x}} \in \mathbb{R}^{|\mathbf{x}|\times|\boldsymbol{\psi}_1|} \tag{11}$$

*In this equation,* $\mathbf{e} \in \mathbb{R}^{|\mathcal{X}|\times|\boldsymbol{\psi}_1|}$ *is a learned parameter matrix and* $\mathbf{x}$ *indexes into its rows.*[11]

**Submodule 3.** ***Multi-Head Self-Attention*** *with* $H$ *heads is defined as:*

$$\texttt{attn}(\mathbf{h}) = \texttt{concat}(\boldsymbol{\eta}_1(\mathbf{h}), \dots, \boldsymbol{\eta}_H(\mathbf{h}))\mathbf{W}^O \tag{12}$$

*where each head operates in dimension* $d_{\boldsymbol{\eta}} \ll |\boldsymbol{\psi}_\ell|$ *and computes:*

$$\boldsymbol{\eta}_i(\mathbf{h}) = \texttt{softmax}\left(\frac{(\mathbf{h}\mathbf{W}_i^Q)(\mathbf{h}\mathbf{W}_i^K)^\top}{\sqrt{d_k}}\right)\mathbf{h}\mathbf{W}_i^V \tag{13}$$

*with learned parameters* $\mathbf{W}^O \in \mathbb{R}^{Hd_{\boldsymbol{\eta}}\times|\boldsymbol{\psi}_\ell|}$, $\mathbf{W}_i^Q, \mathbf{W}_i^K, \mathbf{W}_i^V \in \mathbb{R}^{|\boldsymbol{\psi}_\ell|\times d_{\boldsymbol{\eta}}}$.

**Submodule 4.** ***Layer Normalization*** *applies per-feature normalization:*

$$\texttt{LN}(\mathbf{h}) = \gamma \odot \frac{\mathbf{h} - \mu}{\sigma} + \beta \tag{14}$$

*where* $\mu$ *and* $\sigma$ *are the mean and standard deviation across all features for a single input, and* $\gamma$, $\beta$ *are learned parameters.*

Using these submodules, we define a transformer block.

**Submodule 5.** *A* ***Transformer Block*** *chains together attention and MLP layers with residual connections:*

$$\mathbf{h}' = \mathbf{h} + \texttt{attn}(\texttt{LN}(\mathbf{h})), \qquad f_{\mathtt{N}}^{\ell}(\mathbf{h}) = \mathbf{h}' + \texttt{mlp}(\texttt{LN}(\mathbf{h}')) \tag{15}$$

*where* `mlp` *is applied to each token activations separatly as defined in Submodule 1.*

Finally, we define the complete transformer language model.

**DNN 2.** *A* ***transformer language model*** *consists of an embedding layer, transformer blocks, and an output layer:*

$$\mathbf{h}_{\boldsymbol{\psi}_1} = f_{\mathtt{N}}^{0}(\mathbf{x}) = \mathbf{e}_{\mathbf{x}} \tag{16a}$$

$$\mathbf{h}_{\boldsymbol{\psi}_{\ell+1}} = f_{\mathtt{N}}^{\ell}(\mathbf{h}_{\boldsymbol{\psi}_\ell}) \tag{16b}$$

$$p_{\mathtt{N}}(\mathbf{y} \mid \mathbf{x}) = f_{\mathtt{N}}^{L}(\mathbf{h}_{\boldsymbol{\psi}_L}) = \texttt{softmax}(\texttt{LN}(\mathbf{h}_{\boldsymbol{\psi}_L})_{-1}\,\mathbf{W}_L) \tag{16c}$$

*where* $\texttt{LN}(\mathbf{h}_{\boldsymbol{\psi}_L})_{-1}$ *selects the final token's position after layernorm is applied. For this model,* $\mathcal{H}_{\boldsymbol{\psi}_\ell} = \mathbb{R}^{|\mathbf{x}|\times|\boldsymbol{\psi}_\ell|}$ *for* $1 \leq \ell \leq L$ *and* $\mathcal{H}_{\boldsymbol{\psi}_{L+1}} = \Delta^{|\mathcal{Y}|-1}$.

## F Proof of Theorem 1

In this section, we prove our main theorem. For notational simplicity, we write in this section:

$$\underbrace{f_{\mathtt{N}}^{:\ell} \overset{\text{def}}{=} f_{\mathtt{N}}^{\boldsymbol{\psi}_\ell} = f_{\mathtt{N}}^{\ell-1} \circ \cdots \circ f_{\mathtt{N}}^{0},}_{\texttt{Run DNN up to layer } \ell} \qquad \texttt{and} \qquad \underbrace{f_{\mathtt{N}}^{\ell:} \overset{\text{def}}{=} f_{\mathtt{N}}^{L} \circ \cdots \circ f_{\mathtt{N}}^{\ell}}_{\texttt{Run DNN from layer } \ell} \tag{17}$$

We start by formally stating our assumptions.

**Assumption 1** (Countable input-space)**.** *We assume that the space of inputs (i.e.,* $\mathcal{X}$*) is countable.*

**Assumption 2** (Input-injectivity in all layers)**.** *We assume that* $f_{\mathtt{N}}^{:\ell}$ *is injective for all layers.*

**Assumption 3** (Strict output-surjectivity in all layers)**.** *We assume that the composition of* $f_{\mathtt{N}}^{\ell:}$ *and* $\tau_{\eta_{\mathbf{y}}}$ *is strictly surjective for all layers (we define strict surjectivity in Definition 10).*

**Assumption 4** (Algorithm and DNN have matchable partial-orderings)**.** *We assume that there exists a partitioning* $\{\boldsymbol{\psi}_\eta \mid \eta \in \boldsymbol{\eta}_{\texttt{int}}\} \cup \{\boldsymbol{\psi}_\perp\}$ *of* $\mathtt{N}$*'s neurons* $\boldsymbol{\psi}_{\texttt{int}}$*—where* $\boldsymbol{\psi}_\eta$ *are single neurons—which respects the partial-ordering of algorithm* $\mathtt{A}$*, i.e.,* $\eta \prec \eta' \implies \boldsymbol{\psi}_\eta \prec \boldsymbol{\psi}_{\eta'}$*. Further, for each layer at least one neuron is left unused in this partitioning, i.e.,* $\boldsymbol{\psi}_\perp \cap \boldsymbol{\psi}_\ell \neq \emptyset$.

[11] We ignore positional embeddings here for simplicity, as they do not affect our proofs of injectivity in App. G. Note that, in that section, we show injectivity on an entire layer's activations $\mathbf{h}_{\boldsymbol{\psi}_\ell}$. Position embeddings would be needed to show injectivity on a single position, e.g., the last token's position $(\mathbf{h}_{\boldsymbol{\psi}_\ell})_{-1}$; a property which we conjecture should also hold. Further, we note that position embeddings are used in our experiments.

**Assumption 5** (DNN solves the task)**.** *We assume that for any input* $\mathbf{x} \in \mathcal{X}$*, the neural network solves the task correctly, satisfying* $\mathtt{T}(\mathbf{x}) = \arg\max_{\mathbf{y}\in\mathcal{Y}} p_{\mathtt{N}}(\mathbf{y} \mid \mathbf{x})$.

We provide a longer discussion about why we think these assumptions are reasonable in App. F.1. For convenience, we also put a self-contained version of Definition 7 (input-restricted distributed abstraction) in App. F.2. Now, we restate our theorem and present its proof.

**Theorem 1.** *Given any algorithm* $\mathtt{A}$ *and any neural network* $\mathtt{N}$ *such that Assumps. 1 to 5 hold, we can show that* $\mathtt{A}$ *is an input-restricted distributed abstraction of* $\mathtt{N}$.

*Proof.* To show that an algorithm $\mathtt{A}$ is an input-restricted distributed abstraction of a neural network $\mathtt{N}$, we must show (according to Definition 7) that there exists a $\tau$ for which: $\mathtt{A}$ is an input-restricted $\boldsymbol{\tau}$-abstraction of $\mathtt{N}$; and $\tau$ is a distributed abstraction map. For $\tau$ to be a distributed abstraction map, we need a partition of hidden variables which allows us to independently compute it per node. Further, we need the partitioned hidden variables $\boldsymbol{\psi}^{\phi}_{\mathtt{int}}$ to be the output of an alignment map $\phi$ which is layer-wise decomposable. We thus have:

$$\underbrace{\mathbf{h}_{\boldsymbol{\psi}^{\phi}} = \phi(\mathbf{h}) = [\phi_\ell(\mathbf{h}_{\boldsymbol{\psi}_\ell})]_{\ell=0}^{L+1},}_{\texttt{Align DNN}} \quad \underbrace{\boldsymbol{\Psi}^{\phi} = \{\boldsymbol{\psi}^{\phi}_{\eta} \mid \eta \in \boldsymbol{\eta}_{\mathtt{int}}\} \cup \{\boldsymbol{\psi}^{\phi}_{\perp}\},}_{\texttt{Partition hidden variables}} \quad \underbrace{\tau(\mathbf{h}) = [\tau_\eta(\mathbf{h}_{\boldsymbol{\psi}^{\phi}_{\eta}})]_{\eta\in\boldsymbol{\eta}_{\mathtt{int}}}}_{\texttt{Compute abstraction}} \tag{18}$$

Therefore, to define a distributed abstraction map $\tau$, we must define the following three terms: (i) a set of layer-wise alignment maps $\{\phi_\ell\}_{\ell=1}^{L}$ (note that the alignment maps $\phi_0$ and $\phi_{L+1}$ are fixed by definition); (ii) a partition of hidden variables $\boldsymbol{\Psi}^{\phi}$; and (iii) a set of per-node functions $\{\tau_\eta\}_{\eta\in\boldsymbol{\eta}_{\mathtt{int}}}$. To prove this theorem, then, we must show that there exists a way to define these terms while ensuring that $\mathtt{A}$ is an input-restricted $\boldsymbol{\tau}$-abstraction of $\mathtt{N}$.

We now note that—given Assump. 4 and independently of our choice of alignment map $\phi$—there exists at least one partition $\boldsymbol{\Psi}^{\phi}$ of the hidden variables $\boldsymbol{\psi}^{\phi}_{\mathtt{int}}$ in $\mathtt{N}$ for which:

$$\underbrace{\forall_{\eta,\eta'\in\boldsymbol{\eta}_{\mathtt{int}}} : \eta \prec \eta' \implies \boldsymbol{\psi}^{\phi}_{\eta} \prec \boldsymbol{\psi}^{\phi}_{\eta'},}_{\texttt{respect partial ordering}} \quad \underbrace{\forall_{\eta\in\boldsymbol{\eta}_{\mathtt{int}}} : |\boldsymbol{\psi}^{\phi}_{\eta}| = 1,}_{\texttt{1 neuron per partition}} \quad \underbrace{\forall_{0<\ell\leq L} : \boldsymbol{\psi}^{\phi}_{\ell} \not\subseteq \bigcup_{\eta\in\boldsymbol{\eta}_{\mathtt{int}}} \boldsymbol{\psi}^{\phi}_{\eta}}_{\texttt{no layer is fully occupied}} \tag{19}$$

where we define $\boldsymbol{\psi}^{\phi}_{\ell}$ as the latent variables given when applying $\phi_\ell$ on $\boldsymbol{\psi}_\ell$. To facilitate our proof, we choose one such partition $\boldsymbol{\Psi}^{\phi}$ which we will keep fixed independently of our choice of alignment map $\phi$. Given partition $\boldsymbol{\Psi}^{\phi}$, we can assign each node $\eta$ to a specific layer $\ell$, as $\boldsymbol{\psi}^{\phi}_{\eta}$ contains a single hidden variable and therefore trivially belongs to a single layer. We therefore can define $\boldsymbol{\eta}_\ell$ as all nodes associated with layer $\ell$:

$$\eta \in \boldsymbol{\eta}_\ell \Leftrightarrow \boldsymbol{\psi}^{\phi}_{\eta} \subseteq \boldsymbol{\psi}^{\phi}_{\ell} \tag{20}$$

We now consider the application of interventions on $\mathtt{N}$ as layer-wise on $\boldsymbol{\psi}^{\phi}_{\eta} \subseteq \boldsymbol{\psi}^{\phi}_{\ell}$ for $\eta \in \boldsymbol{\eta}_\ell$. Let us therefore define $\boldsymbol{\mathcal{I}}^{\ell}_{\mathtt{N}}$ as the set of all interventions on $\boldsymbol{\psi}^{\phi}_{\eta}$ for $\eta \in \boldsymbol{\eta}_\ell$, where we note that $\boldsymbol{\mathcal{I}}^{\ell}_{\mathtt{N}}$ also includes an empty intervention (i.e., no intervention). For notational convenience, we will write the set of all interventions up to layer $\ell$ as $\boldsymbol{\mathcal{I}}^{:\ell}_{\mathtt{N}}$, and the set of all nodes associated with those layers as $\boldsymbol{\eta}_{:\ell}$:

$$\boldsymbol{\mathcal{I}}^{:\ell}_{\mathtt{N}} = \boldsymbol{\mathcal{I}}^{0}_{\mathtt{N}} \times \boldsymbol{\mathcal{I}}^{1}_{\mathtt{N}} \times \cdots \times \boldsymbol{\mathcal{I}}^{\ell}_{\mathtt{N}}, \qquad \boldsymbol{\eta}_{:\ell} = \boldsymbol{\eta}_0 \cup \boldsymbol{\eta}_1 \cup \cdots \cup \boldsymbol{\eta}_\ell \tag{21}$$

where $\times$ denotes a Cartesian product. We analogously define $\boldsymbol{\mathcal{I}}^{\ell}_{\mathtt{A}}$ and $\boldsymbol{\mathcal{I}}^{:\ell}_{\mathtt{A}}$.

Finally, we get to an induction proof that will complete this theorem. We will iteratively construct abstraction and alignment maps for each layer such that it holds that:

$$\underset{\substack{\mathbf{I}_{\mathtt{N}}\in\boldsymbol{\mathcal{I}}^{:\ell-1}_{\mathtt{N}} \\ \mathbf{x}\in\mathcal{X}}}{\forall} \ \underset{\eta\in\boldsymbol{\eta}_\ell}{\forall} : \underbrace{\tau_\eta(\mathbf{h}'_{\boldsymbol{\psi}^{\phi}_{\eta}}) = f^{\eta}_{\mathtt{A}}(\mathbf{x}, \mathbf{I}_{\mathtt{A}}),}_{\texttt{tested condition}} \qquad \texttt{where } \underbrace{\mathbf{h}'_{\boldsymbol{\psi}^{\phi}_{\ell}} = \phi_\ell(f^{:\ell}_{\mathtt{N}}(\mathbf{x}, \mathbf{I}_{\mathtt{N}})),}_{\texttt{pre-int. hidden variable, as } \mathbf{I}_{\mathtt{N}}\in\boldsymbol{\mathcal{I}}^{:\ell-1}_{\mathtt{N}}} \ \texttt{and } \mathbf{I}_{\mathtt{A}} = \omega_\tau(\mathbf{I}_{\mathtt{N}}) \quad (1)$$

where int. stands for intervention. Note that if this holds for all layers, we have proven that $\mathtt{A}$ is an input-restricted $\tau$-abstraction of $\mathtt{N}$, as we can perfectly reconstruct the behaviour of algorithm $\mathtt{A}$ from $\mathtt{N}$'s states under any intervention.[12] Also note, however, that our definition of abstraction map restricts $\tau_{\boldsymbol{\eta}_{\mathbf{y}}}(\mathbf{h}_{\boldsymbol{\psi}_{L+1}}) = \arg\max \mathbf{h}_{\boldsymbol{\psi}_{L+1}}$, so special care must be taken to guarantee that this last identity will

[12] The attentive reader may note condition (1) only guarantees we can reconstruct the behaviour of algorithm $\mathtt{A}$ from pre-intervention hidden variables. Lemma 1 shows the same holds for post-intervention hidden variables.

be preserved. We thus also require an additional condition to hold at each step:

$$\forall_{\substack{\mathbf{I}_{\mathtt{N}} \in \boldsymbol{\mathcal{I}}_{\mathtt{N}}^{:\ell} \\ \mathbf{x} \in \mathcal{X}}} : \underbrace{\tau_{\boldsymbol{\eta}_{\mathbf{y}}}(f_{\mathtt{N}}^{\ell:}(\mathbf{h}'_{\boldsymbol{\psi}_\ell})) = f_{\mathtt{A}}(\mathbf{x}, \mathbf{I}_{\mathtt{A}}),}_{\texttt{tested condition}} \quad \texttt{where } \underbrace{\mathbf{h}'_{\boldsymbol{\psi}_\ell} = f_{\mathtt{N}}^{:\ell}(\mathbf{x}, \mathbf{I}_{\mathtt{N}}),}_{\texttt{post-int. neurons, as } \mathbf{I}_{\mathtt{N}} \in \boldsymbol{\mathcal{I}}_{\mathtt{N}}^{:\ell}} \texttt{ and } \mathbf{I}_{\mathtt{A}} = \omega_\tau(\mathbf{I}_{\mathtt{N}}) \quad (2)$$

Finally, for convenience, we add a third condition to our inductive proof which will make the other two conditions easier to guarantee:

$$\forall_{\eta \in \boldsymbol{\eta}_{:\ell-1}} \exists g_\eta \forall_{\substack{\mathbf{I}_{\mathtt{N}} \in \boldsymbol{\mathcal{I}}_{\mathtt{N}}^{:\ell-1} \\ \mathbf{x} \in \mathcal{X}}} : \underbrace{g_\eta(\mathbf{h}'_{\boldsymbol{\psi}_\ell^\phi \cap \boldsymbol{\psi}_\perp^\phi}) = f_{\mathtt{A}}^{\eta}(\mathbf{x}, \mathbf{I}_{\mathtt{A}}),}_{\texttt{tested condition}} \texttt{ where } \underbrace{\mathbf{h}'_{\boldsymbol{\psi}_\ell^\phi} = \phi_\ell(f_{\mathtt{N}}^{:\ell}(\mathbf{x}, \mathbf{I}_{\mathtt{N}})),}_{\texttt{pre-int. hidden variable, as } \mathbf{I}_{\mathtt{N}} \in \boldsymbol{\mathcal{I}}_{\mathtt{N}}^{:\ell-1}} \text{and } \mathbf{I}_{\mathtt{A}} = \omega_\tau(\mathbf{I}_{\mathtt{N}}) \quad (3)$$

This condition guarantees that information about previous nodes (i.e., $\eta \in \boldsymbol{\eta}_{:\ell-1}$) is preserved in each layer's non-intervened neurons (i.e., $\boldsymbol{\psi}_\ell^\phi \cap \boldsymbol{\psi}_\perp^\phi$). This final condition will be useful to guarantee conditions (1) and (2) are preserved in future layers.

**Statement.** Conditions (1), (2), and (3) hold for all layers $\ell$ in a DNN.

**Base Case ($\ell = 0$).** For layer $\ell = 0$, we have $\boldsymbol{\eta}_\ell = \boldsymbol{\eta}_{\mathbf{x}}$. We also have both $\phi_\ell$ and $\tau_\eta$ as the identity function. Further, we consider $\boldsymbol{\mathcal{I}}^{:-1} = \{\emptyset\}$ and $\boldsymbol{\mathcal{I}}^{:0} = \{\emptyset\}$—where symbol $\emptyset$ here denotes an empty intervention—and we consider $f_{\mathtt{N}}^{:0}$ to be the identity on $\mathbf{x}$. (Note that layer $f_{\mathtt{N}}^{0}$ is not applied in $f_{\mathtt{N}}^{:0}$.) Now, it is easy to prove our base case:

- (1) follows trivially, as $f_{\mathtt{A}}^{\boldsymbol{\eta}_{\mathbf{x}}}(\mathbf{x}, \mathbf{I}_{\mathtt{A}}) = \mathbf{x}$ and $f_{\mathtt{N}}^{:0}(\mathbf{x}, \mathbf{I}_{\mathtt{N}}) = \mathbf{x}$.
- (2) follows from Assump. 5.
- (3) follows trivially given $\boldsymbol{\eta}_{:-1}$ is an empty set.

**Induction Step (given $\ell - 1$, then $\ell$).** Now, due to the inductive hypothesis, we assume that (1), (2) and (3) hold for layer $(\ell-1)$. Given this, we must now prove that these conditions also hold for layer $\ell$. We will consider two cases: $\boldsymbol{\eta}_\ell$ is either empty or not. Before doing so, however, we note that (1) and (3) hold for layer $(\ell-1)$'s pre-intervention hidden variables. In Lemmas 1 and 2, we show that the same applies for the post-intervention hidden variables.

Let's consider the **case where $\boldsymbol{\eta}_\ell$ is empty**.

In this case, we can simply define $\phi_\ell$ as the identity map. Further, given an empty $\boldsymbol{\eta}_\ell$, we know that there are no interventions in this layer, i.e., $\boldsymbol{\mathcal{I}}_{\mathtt{N}}^{\ell} = \{\emptyset\}$, and, as such, we have that: $\boldsymbol{\mathcal{I}}_{\mathtt{N}}^{:\ell} = \boldsymbol{\mathcal{I}}_{\mathtt{N}}^{:\ell-1} \times \boldsymbol{\mathcal{I}}_{\mathtt{N}}^{\ell} = \boldsymbol{\mathcal{I}}_{\mathtt{N}}^{:\ell-1}$. We can now prove the induction step for this case.

- (1) is true trivially, since $\boldsymbol{\eta}_\ell$ is empty.
- (2) follows using the inductive hypothesis. Let $\mathbf{I}_{\mathtt{N}} \in \boldsymbol{\mathcal{I}}_{\mathtt{N}}^{:\ell}$ and $\mathbf{x} \in \mathcal{X}$. Now, let $\mathbf{h}'_{\boldsymbol{\psi}_\ell} = f_{\mathtt{N}}^{:\ell}(\mathbf{x}, \mathbf{I}_{\mathtt{N}})$, $\mathbf{h}'_{\boldsymbol{\psi}_{\ell-1}} = f_{\mathtt{N}}^{:\ell-1}(\mathbf{x}, \mathbf{I}_{\mathtt{N}})$, and $\mathbf{I}_{\mathtt{A}} = \omega_\tau(\mathbf{I}_{\mathtt{N}})$. We can now show that:

$$\begin{aligned}
\tau_{\boldsymbol{\eta}_{\mathbf{y}}}\left(f_{\mathtt{N}}^{\ell:}(\mathbf{h}'_{\boldsymbol{\psi}_\ell})\right) &= \tau_{\boldsymbol{\eta}_{\mathbf{y}}}\left(f_{\mathtt{N}}^{\ell:}\left(f_{\mathtt{N}}^{:\ell}(\mathbf{x}, \mathbf{I}_{\mathtt{N}})\right)\right) && \text{definition of } \mathbf{h}'_{\boldsymbol{\psi}_\ell} && (22a)\\
&= \tau_{\boldsymbol{\eta}_{\mathbf{y}}}\left(f_{\mathtt{N}}^{\ell:}\left(f_{\mathtt{N}}^{\ell-1}(f_{\mathtt{N}}^{:\ell-1}(\mathbf{x}, \mathbf{I}_{\mathtt{N}}))\right)\right) && \text{no intervention at layer } \ell && (22b)\\
&= \tau_{\boldsymbol{\eta}_{\mathbf{y}}}\left(f_{\mathtt{N}}^{\ell-1:}\left(f_{\mathtt{N}}^{:\ell-1}(\mathbf{x}, \mathbf{I}_{\mathtt{N}})\right)\right) && \text{definition of } f_{\mathtt{N}}^{\ell-1:} && (22c)\\
&= \tau_{\boldsymbol{\eta}_{\mathbf{y}}}\left(f_{\mathtt{N}}^{\ell-1:}\left(\mathbf{h}'_{\boldsymbol{\psi}_{\ell-1}}\right)\right) && \text{definition of } \mathbf{h}'_{\boldsymbol{\psi}_{\ell-1}} && (22d)\\
&= f_{\mathtt{A}}(\mathbf{x}, \mathbf{I}_{\mathtt{A}}) && \text{inductive hypothesis on (2)} && (22e)
\end{aligned}$$

  This shows (2) holds for layer $\ell$ when $\boldsymbol{\eta}_\ell$ is empty.

- (3) follows using the inductive hypothesis. Let $\mathbf{I}_{\mathtt{N}} \in \boldsymbol{\mathcal{I}}_{\mathtt{N}}^{:\ell-1}$, $\mathbf{x} \in \mathcal{X}$, and $\eta \in \boldsymbol{\eta}_{:\ell-1}$. Now, let $\mathbf{h}'_{\boldsymbol{\psi}_\ell^\phi} = \phi_\ell(f_{\mathtt{N}}^{:\ell}(\mathbf{x}, \mathbf{I}_{\mathtt{N}}))$, $\mathbf{h}'_{\boldsymbol{\psi}_{\ell-1}^\phi} = \phi_{\ell-1}(f_{\mathtt{N}}^{:\ell-1}(\mathbf{x}, \mathbf{I}_{\mathtt{N}}))$, and $\mathbf{I}_{\mathtt{A}} = \omega_\tau(\mathbf{I}_{\mathtt{N}})$. Further, let

$g_{\eta}^{\ell-1}(\mathbf{h}'_{\boldsymbol{\psi}^{\phi}_{\ell-1}}) = f_{\mathtt{A}}^{:\eta}(\mathbf{x}, \mathbf{I}_{\mathtt{A}})$; we know such function exists due to the inductive hypothesis on ① and ③, together with Lemmas 1 and 2. Finally, since $f_{\mathtt{N}}^{:\ell}$ is injective (by Assump. 2) and since $f_{\mathtt{N}}^{:\ell} = f_{\mathtt{N}}^{\ell-1} \circ f_{\mathtt{N}}^{:\ell-1}$, we know that each $\mathbf{h}'_{\boldsymbol{\psi}^{\phi}_{\ell-1}}$ is mapped to a unique $\mathbf{h}'_{\boldsymbol{\psi}^{\phi}_{\ell}}$ in the next layer. We can thus define function $\text{\reflectbox{$f$}}_{\mathtt{N}}^{\ell-1}$ on the domain formed by these hidden variables which, given a hidden variable $\mathbf{h}'_{\boldsymbol{\psi}^{\phi}_{\ell}}$ returns its "parent" $\mathbf{h}'_{\boldsymbol{\psi}^{\phi}_{\ell-1}}$; in other words, $\text{\reflectbox{$f$}}_{\mathtt{N}}^{\ell-1}$ is an partial inverse of $f_{\mathtt{N}}^{\ell-1}$ defined only on its image. Defining now function $g_{\eta}^{\ell} = g_{\eta}^{\ell-1} \circ \text{\reflectbox{$f$}}_{\mathtt{N}}^{\ell-1}$, we can show:

$$
\begin{aligned}
g_{\eta}^{\ell}(\mathbf{h}'_{\boldsymbol{\psi}^{\phi}_{\ell} \cap \boldsymbol{\psi}^{\phi}_{\perp}}) &= g_{\eta}^{\ell}(\mathbf{h}'_{\boldsymbol{\psi}^{\phi}_{\ell}}) && \boldsymbol{\eta}_{\ell} \text{ is empty} \quad (23a)\\
&= g_{\eta}^{\ell}\left(\phi_{\ell}\left(f_{\mathtt{N}}^{\ell-1}(\mathbf{h}'_{\boldsymbol{\psi}^{\phi}_{\ell-1}})\right)\right) && \text{no intervention at layer } \ell \quad (23b)\\
&= g_{\eta}^{\ell}\left(f_{\mathtt{N}}^{\ell-1}(\mathbf{h}'_{\boldsymbol{\psi}^{\phi}_{\ell-1}})\right) && \phi_{\ell} \text{ is the identity function} \quad (23c)\\
&= g_{\eta}^{\ell-1}\left(\text{\reflectbox{$f$}}_{\mathtt{N}}^{\ell-1}\left(f_{\mathtt{N}}^{\ell-1}(\mathbf{h}'_{\boldsymbol{\psi}^{\phi}_{\ell-1}})\right)\right) && \text{definition of } g^{\ell} \quad (23d)\\
&= g_{\eta}^{\ell-1}(\mathbf{h}'_{\boldsymbol{\psi}^{\phi}_{\ell-1}}) && \text{definition of } \text{\reflectbox{$f$}}_{\mathtt{N}}^{\ell-1} \quad (23e)\\
&= f_{\mathtt{A}}^{:\eta}(\mathbf{x}, \mathbf{I}_{\mathtt{A}}) && \text{inductive hypothesis on ① and ③} \quad (23f)
\end{aligned}
$$

Let's now consider the **case when $\boldsymbol{\eta}_{\ell}$ is not empty**.

To show ①, ② and ③ for layer $\ell$ we need to find a suitable bijective $\phi_{\ell}$. We now show a careful way to construct this map which satisfies these conditions. To do so, we will again split this step of the proof into two parts. We will first take care of the case in which no interventions are applied to layer $\ell$, guaranteeing that the model behaves correctly in those cases. In that case, we must handle the set of **input-restricted pre-intervention hidden states** in layer $\ell$, which we define as:

$$
\mathcal{H}^{\blacklozenge}_{\boldsymbol{\psi}_{\ell}} \stackrel{\text{def}}{=} \{f_{\mathtt{N}}^{\boldsymbol{\psi}_{\ell}}(\mathbf{x}, \mathbf{I}_{\mathtt{N}}) \mid \mathbf{x} \in \mathcal{X}, \underbrace{\mathbf{I}_{\mathtt{N}} \in \boldsymbol{\mathcal{I}}_{\mathtt{N}}^{:\ell-1}}_{\texttt{pre-int., as we do not include } \boldsymbol{\mathcal{I}}^{\ell}}\} \tag{24}
$$

Notably, instead of defining the entire alignment map $\phi_{\ell}$ at once, we will first define its behaviour only on those hidden states. We will denote this domain-restricted function as $\phi_{\ell}^{\blacklozenge}$. Given this function, we will be able to define a set of **input-restricted pre-intervention hidden variables** in layer $\ell$ as:

$$
\mathcal{H}^{\blacklozenge}_{\boldsymbol{\psi}^{\phi}_{\ell}} \stackrel{\text{def}}{=} \{\phi_{\ell}^{\blacklozenge}(\mathbf{h}_{\boldsymbol{\psi}_{\ell}}) \mid \mathbf{h}_{\boldsymbol{\psi}_{\ell}} \in \mathcal{H}^{\blacklozenge}_{\boldsymbol{\psi}_{\ell}}\} \tag{25}
$$

where ◆ represents the non-intervened hidden states and variables, and we will use ❖ to represent the intervened instances. Note that $\mathcal{H}^{\blacklozenge}_{\boldsymbol{\psi}^{\phi}_{\ell}}$ is the set of representations output by alignment map $\phi_{\ell}^{\blacklozenge}$.

The second case we will consider will then handle interventions on this layer, and will again guarantee that the model behaves as expected in those cases. We thus define the set of **input-restricted post-intervention hidden variables** as:

$$
\mathcal{H}_{\boldsymbol{\psi}^{\phi}_{\ell}} \stackrel{\text{def}}{=} \{f_{\mathtt{N}}^{\boldsymbol{\psi}^{\phi}_{\ell}}(\mathbf{x}, \mathbf{I}_{\mathtt{N}}) \mid \mathbf{x} \in \mathcal{X}, \underbrace{\mathbf{I}_{\mathtt{N}} \in \boldsymbol{\mathcal{I}}_{\mathtt{N}}^{:\ell}}_{\texttt{post-int., as we include } \boldsymbol{\mathcal{I}}^{\ell}}\} \tag{26}
$$

Notably, what an intervention on layer $\ell$ does is re-combine the representations in $\mathcal{H}^{\blacklozenge}_{\boldsymbol{\psi}^{\phi}_{\ell}}$. We can thus write $\mathcal{H}_{\boldsymbol{\psi}^{\phi}_{\ell}}$ in terms of $\mathcal{H}^{\blacklozenge}_{\boldsymbol{\psi}^{\phi}_{\ell}}$ as:

$$
\mathcal{H}_{\boldsymbol{\psi}^{\phi}_{\ell}} = \left(\bigtimes_{\eta \in \boldsymbol{\eta}_{\ell}} \underbrace{\{\phi_{\ell}(\mathbf{h}_{\boldsymbol{\psi}_{\ell}})_{\boldsymbol{\psi}^{\phi}_{\eta}} \mid \mathbf{h}_{\boldsymbol{\psi}_{\ell}} \in \mathcal{H}^{\blacklozenge}_{\boldsymbol{\psi}_{\ell}}\}}_{\texttt{pre-int. h.v., projected to } \boldsymbol{\psi}^{\phi}_{\eta}}\right) \times \underbrace{\{\phi_{\ell}(\mathbf{h}_{\boldsymbol{\psi}_{\ell}})_{\boldsymbol{\psi}^{\phi}_{\perp} \cap \boldsymbol{\psi}^{\phi}_{\ell}} \mid \mathbf{h}_{\boldsymbol{\psi}_{\ell}} \in \mathcal{H}^{\blacklozenge}_{\boldsymbol{\psi}_{\ell}}\}}_{\texttt{pre-int. h.v., projected to } \boldsymbol{\psi}^{\phi}_{\perp}} \tag{27}
$$

We further define the set of **input-restricted intervention-only hidden variables** as:

$$
\mathcal{H}^{❖}_{\boldsymbol{\psi}^{\phi}_{\ell}} = \mathcal{H}_{\boldsymbol{\psi}^{\phi}_{\ell}} \setminus \mathcal{H}^{\blacklozenge}_{\boldsymbol{\psi}^{\phi}_{\ell}} \tag{28}
$$

By carefully defining the behaviour of $\phi_\ell$ on this set, we can guarantee the conditions above to hold. In particular, we will define this part of the function via its inverse $\phi_\ell^{\text{❖}-1}$, which maps these hidden variables back to hidden states. We therefore have $\phi_\ell$ and its partial inverse defined as:

$$\phi_\ell(\mathbf{h}) = \begin{cases} \phi_\ell^{\blacklozenge}(\mathbf{h}), & \texttt{if } \mathbf{h} \in \mathcal{H}^{\blacklozenge}_{\boldsymbol{\psi}_\ell} \\ \phi_\ell^{\text{❖}}(\mathbf{h}), & \texttt{if } \mathbf{h} \in \mathcal{H}^{\text{❖}}_{\boldsymbol{\psi}_\ell} \end{cases} \qquad \phi_\ell^{-1}(\mathbf{h}) = \begin{cases} \phi_\ell^{\blacklozenge-1}(\mathbf{h}), & \texttt{if } \mathbf{h} \in \mathcal{H}^{\blacklozenge}_{\boldsymbol{\psi}^\phi_\ell} \\ \phi_\ell^{\text{❖}-1}(\mathbf{h}), & \texttt{if } \mathbf{h} \in \mathcal{H}^{\text{❖}}_{\boldsymbol{\psi}^\phi_\ell} \end{cases} \tag{29}$$

We now define $\phi_\ell^{\blacklozenge}$.

**Definition 8.** *Partial map* $\phi_\ell^{\blacklozenge} : \mathcal{H}^{\blacklozenge}_{\boldsymbol{\psi}_\ell} \to \mathbb{R}^{|\boldsymbol{\psi}^\ell|}$ *is some fixed function that is injective on each dimension, i.e.,* $\forall_{i \in \{1,\ldots,|\boldsymbol{\psi}^\phi_\ell|\}} \forall_{\mathbf{h}_1,\mathbf{h}_2 \in \mathcal{H}^{\blacklozenge}_{\boldsymbol{\psi}_\ell}} : \mathbf{h}_1 \neq \mathbf{h}_2 \Rightarrow \phi_\ell^{\blacklozenge}(\mathbf{h}_1)_i \neq \phi_\ell^{\blacklozenge}(\mathbf{h}_2)_i$.

Such a function exists, because $\mathcal{H}^{\blacklozenge}_{\boldsymbol{\psi}_\ell}$ is countable (Lemma 4) and $\mathbb{R}$ is uncountable. Further, its partial inverse $\phi_\ell^{\blacklozenge-1}$, defined on the image $\mathcal{H}^{\blacklozenge}_{\boldsymbol{\psi}^\phi_\ell}$, exists because $\phi_\ell^{\blacklozenge}$ is injective.

We can now prove that conditions ① and ③ hold. We also prove that ② holds when $\mathbf{I}_{\mathtt{N}} \in \boldsymbol{\mathcal{I}}^{:\ell-1}_{\mathtt{N}}$, i.e., when there is no intervention in layer $\ell$.

- ① follows using the inductive hypothesis. Let $\mathbf{I}_{\mathtt{N}} \in \boldsymbol{\mathcal{I}}^{:\ell-1}_{\mathtt{N}}$, $\mathbf{x} \in \mathcal{X}$, and $\eta \in \boldsymbol{\eta}_\ell$. Further, let $\mathbf{h}'_{\boldsymbol{\psi}^\phi_\ell} = \phi_\ell(f^{:\ell}_{\mathtt{N}}(\mathbf{x}, \mathbf{I}_{\mathtt{N}}))$, $\mathbf{h}'_{\boldsymbol{\psi}^\phi_{\ell-1}} = \phi_{\ell-1}(f^{:\ell-1}_{\mathtt{N}}(\mathbf{x}, \mathbf{I}_{\mathtt{N}}))$, and $\mathbf{I}_{\mathtt{A}} = \omega_\tau(\mathbf{I}_{\mathtt{N}})$. Now, note that there exists a function $g_\eta$ for which $v_\eta = g_\eta(\mathbf{v}_{\boldsymbol{\eta}_{:\ell-1}})$, as the parents of $\eta$ are a subset of $\boldsymbol{\eta}_{:\ell-1}$. It now suffices to show that $\mathbf{h}'_{\boldsymbol{\psi}^\phi_\eta}$ encodes information about $\mathbf{v}_{\boldsymbol{\eta}_{:\ell-1}} = [f^{:\eta}_{\mathtt{A}}(\mathbf{x}, \mathbf{I}_{\mathtt{A}})]_{\eta \in \boldsymbol{\eta}_{:\ell-1}}$. By the inductive hypothesis on ① and ③, together with Lemmas 1 and 2, we know that $\mathbf{h}'_{\boldsymbol{\psi}^\phi_{\ell-1}}$ encodes information about $\mathbf{v}_{\boldsymbol{\eta}_{:\ell-1}}$; let $g_{\boldsymbol{\eta}_{:\ell-1}}$ be a function that extracts this information, i.e., $g_{\boldsymbol{\eta}_{:\ell-1}}(\mathbf{h}'_{\boldsymbol{\psi}^\phi_{\ell-1}}) = \mathbf{v}_{\boldsymbol{\eta}_{:\ell-1}}$. Now, since $f^{:\ell}_{\mathtt{N}}$ is injective, and $\phi_\ell^{\blacklozenge}$ is injective on each output dimension, we know that $\mathbf{h}'_{\boldsymbol{\psi}_\ell} = [\phi_\ell^{\blacklozenge}(f^{\ell-1}_{\mathtt{N}}(\phi^{-1}_{\ell-1}(\mathbf{h}'_{\boldsymbol{\psi}^\phi_{\ell-1}})))]_{\boldsymbol{\psi}^\phi_\eta}$ contains the same information as $\mathbf{h}'_{\boldsymbol{\psi}^\phi_{\ell-1}}$. We can thus construct (partial) inverses $\text{ɟ}^{\ell-1}_{\mathtt{N}}$, $\phi^{\blacklozenge-1}_{\ell,\boldsymbol{\psi}^\phi_\eta}$ and define $\tau_\eta$ as the composition $g_\eta \circ g_{:\eta-1} \circ \phi_{\ell-1} \circ \text{ɟ}^{\ell-1}_{\mathtt{N}} \circ \phi^{\blacklozenge-1}_{\ell,\boldsymbol{\psi}^\phi_\eta}$, which concludes this step of the proof:

$$\begin{aligned}
\tau_\eta([\mathbf{h}'_{\boldsymbol{\psi}^\phi_\ell}]_{\boldsymbol{\psi}^\phi_\eta}) &= \tau_\eta([\phi_\ell^{\blacklozenge}(f^{\ell-1}_{\mathtt{N}}(\phi^{-1}_{\ell-1}(\mathbf{h}'_{\boldsymbol{\psi}^\phi_{\ell-1}})))]_{\boldsymbol{\psi}^\phi_\eta}) && \text{definition of } \mathbf{h}'_{\boldsymbol{\psi}_\ell} && \text{(30a)} \\
&= g_\eta(g_{:\eta-1}(\phi_{\ell-1}(\text{ɟ}^{\ell-1}_{\mathtt{N}}(\phi^{\blacklozenge-1}_{\ell,\boldsymbol{\psi}^\phi_\eta}([\phi_\ell^{\blacklozenge}(f^{\ell-1}_{\mathtt{N}}(\phi^{-1}_{\ell-1}(\mathbf{h}'_{\boldsymbol{\psi}^\phi_{\ell-1}})))]_{\boldsymbol{\psi}^\phi_\eta}))))) && && \text{(30b)} \\
&= g_\eta(g_{:\eta-1}(\mathbf{h}'_{\boldsymbol{\psi}^\phi_{\ell-1}})) && && \text{(30c)} \\
&= f^{:\eta}_{\mathtt{A}}(\mathbf{x}, \mathbf{I}_{\mathtt{A}}) && && \text{(30d)}
\end{aligned}$$

- ② when $\mathbf{I}_{\mathtt{N}} \in \boldsymbol{\mathcal{I}}^{:\ell-1}_{\mathtt{N}}$ follows using the inductive hypothesis. Let $\mathbf{I}_{\mathtt{N}} \in \boldsymbol{\mathcal{I}}^{:\ell-1}_{\mathtt{N}}$ and $\mathbf{x} \in \mathcal{X}$. Now, let $\mathbf{h}'_{\boldsymbol{\psi}_\ell} = f^{:\ell}_{\mathtt{N}}(\mathbf{x}, \mathbf{I}_{\mathtt{N}})$, $\mathbf{h}'_{\boldsymbol{\psi}_{\ell-1}} = f^{:\ell-1}_{\mathtt{N}}(\mathbf{x}, \mathbf{I}_{\mathtt{N}})$, and $\mathbf{I}_{\mathtt{A}} = \omega_\tau(\mathbf{I}_{\mathtt{N}})$. We can show that:

$$\begin{aligned}
\tau_{\boldsymbol{\eta}_{\mathbf{y}}}\left(f^{\ell:}_{\mathtt{N}}(\mathbf{h}'_{\boldsymbol{\psi}_\ell})\right) &= \tau_{\boldsymbol{\eta}_{\mathbf{y}}}\left(f^{\ell:}_{\mathtt{N}}\left(f^{:\ell}_{\mathtt{N}}(\mathbf{x}, \mathbf{I}_{\mathtt{N}})\right)\right) && \text{definition of } \mathbf{h}'_{\boldsymbol{\psi}_\ell} && \text{(31a)} \\
&= \tau_{\boldsymbol{\eta}_{\mathbf{y}}}\left(f^{\ell:}_{\mathtt{N}}\left(f^{\ell-1}_{\mathtt{N}}\left(f^{:\ell-1}_{\mathtt{N}}(\mathbf{x}, \mathbf{I}_{\mathtt{N}})\right)\right)\right) && \text{no intervention at layer } \ell && \text{(31b)} \\
&= \tau_{\boldsymbol{\eta}_{\mathbf{y}}}\left(f^{\ell-1:}_{\mathtt{N}}\left(f^{:\ell-1}_{\mathtt{N}}(\mathbf{x}, \mathbf{I}_{\mathtt{N}})\right)\right) && \text{definition of } f^{\ell-1:}_{\mathtt{N}} && \text{(31c)} \\
&= \tau_{\boldsymbol{\eta}_{\mathbf{y}}}\left(f^{\ell-1:}_{\mathtt{N}}\left(\mathbf{h}'_{\boldsymbol{\psi}_{\ell-1}}\right)\right) && \text{definition of } \mathbf{h}'_{\boldsymbol{\psi}_{\ell-1}} && \text{(31d)} \\
&= f_{\mathtt{A}}(\mathbf{x}, \mathbf{I}_{\mathtt{A}}) && \text{inductive hypothesis on ②} && \text{(31e)}
\end{aligned}$$

  This shows ② holds for layer $\ell$ when there is no intervention in layer $\ell$.

- ③ follows using the inductive hypothesis. Let $\mathbf{I}_{\mathtt{N}} \in \boldsymbol{\mathcal{I}}^{:\ell-1}_{\mathtt{N}}$, $\mathbf{x} \in \mathcal{X}$, and $\eta \in \boldsymbol{\eta}_{:\ell-1}$. Now, let $\mathbf{h}'_{\boldsymbol{\psi}^\phi_\ell} = \phi_\ell(f^{:\ell}_{\mathtt{N}}(\mathbf{x}, \mathbf{I}_{\mathtt{N}}))$, $\mathbf{h}'_{\boldsymbol{\psi}^\phi_{\ell-1}} = \phi_{\ell-1}(f^{:\ell-1}_{\mathtt{N}}(\mathbf{x}, \mathbf{I}_{\mathtt{N}}))$, and $\mathbf{I}_{\mathtt{A}} = \omega_\tau(\mathbf{I}_{\mathtt{N}})$. Further, let

$g_\eta^{\ell-1}(\mathbf{h}'_{\boldsymbol{\psi}^\phi_{\ell-1}}) = f_{\texttt{A}}^{:\eta}(\mathbf{x}, \mathbf{I}_{\texttt{A}})$; we know such function exists due to the inductive hypothesis on ① and ③ together with Lemmas 1 and 2. Finally, since $f_{\texttt{N}}^{:\ell}$ and $\phi_\ell^{\blacklozenge}$ are injective and $\phi_{\ell-1}$ is bijective, we can define the partial inverse function $\daleth_{\texttt{N}}^{\ell-1}$ of their composition $\phi_\ell^{\blacklozenge} \circ f_{\texttt{N}}^{\ell} \circ \phi_{\ell-1}^{-1}$ (applied only to their image) which, given the hidden variable $\mathbf{h}'_{\boldsymbol{\psi}^\phi_\ell}$ returns its "parent" $\mathbf{h}'_{\boldsymbol{\psi}^\phi_{\ell-1}}$. Defining now a function $\widehat{g}_\eta^\ell = g_\eta^{\ell-1} \circ \daleth_{\texttt{N}}^{\ell-1}$ and $g_\eta^\ell(\mathbf{h}'_{\boldsymbol{\psi}^\phi_\ell \cap \boldsymbol{\psi}^\phi_\perp}) = \widehat{g}_\eta^\ell(\mathbf{h}'_{\boldsymbol{\psi}^\phi_\ell})$—which exists, as $\phi_\ell^{\blacklozenge}$ is injective on each dimension—we can show:

$$
\begin{aligned}
g_\eta^\ell(\mathbf{h}'_{\boldsymbol{\psi}^\phi_\ell \cap \boldsymbol{\psi}^\phi_\perp}) &= \widehat{g}_\eta^\ell(\mathbf{h}'_{\boldsymbol{\psi}^\phi_\ell}) && \phi_\ell^{\blacklozenge} \text{ is injective on all dimensions} \quad (32a)\\
&= \widehat{g}_\eta^\ell\left(\phi_\ell^{\blacklozenge}\left(f_{\texttt{N}}^{\ell-1}(\phi_{\ell-1}^{-1}(\mathbf{h}'_{\boldsymbol{\psi}^\phi_{\ell-1}}))\right)\right) && \text{no intervention at layer } \ell \quad (32b)\\
&= g_\eta^{\ell-1}\left(\daleth_{\texttt{N}}^{\ell-1}\left(\phi_\ell^{\blacklozenge}\left(f_{\texttt{N}}^{\ell-1}(\phi_{\ell-1}^{-1}(\mathbf{h}'_{\boldsymbol{\psi}^\phi_{\ell-1}}))\right)\right)\right) && \text{definition of } g^\ell \quad (32c)\\
&= g_\eta^{\ell-1}(\mathbf{h}'_{\boldsymbol{\psi}^\phi_{\ell-1}}) && \text{definition of } \daleth_{\texttt{N}}^{\ell-1} \quad (32d)\\
&= f_{\texttt{A}}^{:\eta}(\mathbf{x}, \mathbf{I}_{\texttt{A}}) && \text{inductive hypothesis on ① and ③} \quad (32e)
\end{aligned}
$$

We have now proved ① and ③. We have also partially proved ② for cases where there is no intervention in layer $\ell$.[13] We now finish our proof by considering cases where there is an intervention in this layer $\ell$. In the second case, we need to handle intervention-only representations $\mathcal{H}^{\text{❖}}_{\boldsymbol{\psi}^\phi_\ell}$. We will now define $\phi_\ell^{\text{❖}-1}$ on this domain $\mathcal{H}^{\text{❖}}_{\boldsymbol{\psi}^\phi_\ell}$ to fulfil ②.

**Definition 9.** *Partial map* $\phi_\ell^{\text{❖}-1} : \mathcal{H}^{\text{❖}}_{\boldsymbol{\psi}^\phi_\ell} \to \mathbb{R}^{|\boldsymbol{\psi}^\ell|}$ *is some fixed function such that it holds:*

1. $\phi_\ell^{\text{❖}-1}$ *maps to the set* $\mathcal{H}_{\boldsymbol{\psi}_\ell} \setminus \mathcal{H}^{\blacklozenge}_{\boldsymbol{\psi}_\ell}$
2. $\phi_\ell^{\text{❖}-1}$ *is an injective map*
3. *Let* $\mathbf{I}_{\texttt{N}} \in \boldsymbol{\mathcal{I}}_{\texttt{N}}^{:\ell} \setminus \boldsymbol{\mathcal{I}}_{\texttt{N}}^{:\ell-1}$ *and* $\mathbf{x} \in \mathcal{X}$. *Now, let* $\mathbf{h}'_{\boldsymbol{\psi}^\phi_\ell} = f_{\texttt{N}}^{\boldsymbol{\psi}^\phi_\ell}(\mathbf{x}, \mathbf{I}_{\texttt{N}})$, *and* $\mathbf{I}_{\texttt{A}} = \omega_\tau(\mathbf{I}_{\texttt{N}})$. *We have that* $f_{\texttt{N}}^{\ell:}(\phi_\ell^{\text{❖}-1}(\mathbf{h})) = f_{\texttt{A}}(\mathbf{x}, \mathbf{I}_{\texttt{A}})$.

Where the first two conditions ensure the necessary bijectivity of $\phi_\ell$ and the last characteristic ensures ②. Now, let $\mathbf{x} \in \mathcal{X}$ be any input and $\mathbf{I}_{\texttt{N}} \in \boldsymbol{\mathcal{I}}_{\texttt{N}}^{:\ell}$ any intervention. Further, let $\mathbf{h}'_{\boldsymbol{\psi}^\phi_\ell} = f_{\texttt{N}}^{\boldsymbol{\psi}^\phi_\ell}(\mathbf{x}, \mathbf{I}_{\texttt{N}})$, and $\mathbf{I}_{\texttt{A}} = \omega_\tau(\mathbf{I}_{\texttt{N}})$. We now note that—given ① and ③, and Lemmas 1 and 2—the value $v_\eta = f_{\texttt{A}}^{:\eta}(\mathbf{x}, \mathbf{I}_{\texttt{A}})$ for all nodes $\eta \in \boldsymbol{\eta}_{:\ell}$ are encoded in $\mathbf{h}'_{\boldsymbol{\psi}^\phi_\ell}$. This is enough information to determine the algorithm's output $\mathbf{y}^\star = f_{\texttt{A}}(\mathbf{x}, \mathbf{I}_{\texttt{A}})$. Now, define a function $g^\star$ which maps an element $\mathbf{h}'_{\boldsymbol{\psi}^\phi_\ell} \in \mathcal{H}^{\text{❖}}_{\boldsymbol{\psi}^\phi_\ell}$ to the output algorithm $\texttt{A}$ expects. Further, by Lemma 6 there exists an uncountably infinite set of hidden states $\mathcal{H}^{(\mathbf{y}^\star)}_{\boldsymbol{\psi}_\ell}$ such that:

$$\forall \mathbf{h} \in \mathcal{H}^{(\mathbf{y}^\star)}_{\boldsymbol{\psi}_\ell} : \mathbf{y}^\star = \operatorname*{argmax}_{\mathbf{y}' \in \mathcal{Y}} [f_{\texttt{N}}^{\ell:}(\mathbf{h}_{\boldsymbol{\psi}_\ell})]_{\mathbf{y}'} \tag{33}$$

We define $\hat{\mathcal{H}}^{(\mathbf{y}^\star)}_{\boldsymbol{\psi}_\ell} = \mathcal{H}^{(\mathbf{y}^\star)}_{\boldsymbol{\psi}_\ell} \setminus \mathcal{H}^{\blacklozenge}_{\boldsymbol{\psi}_\ell}$, which—as $\mathcal{H}^{\blacklozenge}_{\boldsymbol{\psi}_\ell}$ is countable—is still uncountably infinite. We can now map any $\mathbf{h} \in \mathcal{H}^{\text{❖}}_{\boldsymbol{\psi}^\phi_\ell}$ to an element in $\hat{\mathcal{H}}^{(g^\star(\mathbf{h}))}_{\boldsymbol{\psi}_\ell}$ fulfilling the third characteristic of Definition 9. That such a mapping exists adhering to the first and second characteristic of Definition 9 is ensured by the fact that $\hat{\mathcal{H}}^{(\mathbf{y}^\star)}_{\boldsymbol{\psi}_\ell}$ is uncountable and $\mathcal{H}^{\text{❖}}_{\boldsymbol{\psi}^\phi_\ell}$ is countable (shown in Lemma 5). Further, as $\phi_\ell^{\text{❖}-1}$ is injective, its partial inverse $\phi_\ell^{\text{❖}}$ on its image $\mathcal{H}^{\text{❖}}_{\boldsymbol{\psi}_\ell}$ exists.

[13] This also proves the result for any input $\mathbf{x} \in \mathcal{X}$ and intervention $\mathbf{I}_{\texttt{N}} \in \boldsymbol{\mathcal{I}}^\ell$ where $\mathbf{h} \in \mathcal{H}^{\blacklozenge}_{\boldsymbol{\psi}_\ell}$. By ③ at layer $\ell$, there exists $\mathbf{I}'_{\texttt{N}} \in \boldsymbol{\mathcal{I}}^{\ell-1}$—constructed by applying only the interventions from $\mathbf{I}_{\texttt{N}}$ on layers $< \ell$—such that $f_{\texttt{N}}^{:\ell}(\mathbf{x}, \mathbf{I}_{\texttt{N}}) = f_{\texttt{N}}^{:\ell}(\mathbf{x}, \mathbf{I}'_{\texttt{N}})$. Since both encode the same values on $\boldsymbol{\eta}_{<\ell}$ according to ③, which fully determine the output of $f_{\texttt{A}}$, and since ② holds for $(\mathbf{x}, \mathbf{I}'_{\texttt{N}})$ at layer $\ell$, it must also hold for $(\mathbf{x}, \mathbf{I}_{\texttt{N}})$.

```
def extract_unused_rep(h, H◆_{ψ_ℓ^φ} ∪ H❖_{ψ_ℓ^φ}):
    rep=h
    while rep∈ H◆_{ψ_ℓ^φ} ∪ H❖_{ψ_ℓ^φ}:
        rep=φ_ℓ^{-1}(rep)
    return rep
```

Figure 7: Pseudo-code for `extract_unused_rep`. This function returns an unique element in $(\mathcal{H}^{\blacklozenge}_{\boldsymbol{\psi}_\ell} \cup \mathcal{H}^{\maltese}_{\boldsymbol{\psi}_\ell}) \setminus (\mathcal{H}^{\blacklozenge}_{\boldsymbol{\psi}^{\phi}_\ell} \cup \mathcal{H}^{\maltese}_{\boldsymbol{\psi}^{\phi}_\ell})$ for each $\mathbf{h} \in (\mathcal{H}^{\blacklozenge}_{\boldsymbol{\psi}^{\phi}_\ell} \cup \mathcal{H}^{\maltese}_{\boldsymbol{\psi}^{\phi}_\ell}) \setminus (\mathcal{H}^{\blacklozenge}_{\boldsymbol{\psi}_\ell} \cup \mathcal{H}^{\maltese}_{\boldsymbol{\psi}_\ell})$ ensuring bijectivity of $\phi'_\ell(\mathbf{h})$.

The attentive reader may have noticed that we defined $\phi_\ell$ only over the domain $\mathcal{H}^{\blacklozenge}_{\boldsymbol{\psi}_\ell} \cup \mathcal{H}^{\maltese}_{\boldsymbol{\psi}_\ell}$ instead over $\mathbb{R}^{|\boldsymbol{\psi}_\ell|}$. We note that it is simple to extend $\phi_\ell$ to an $\phi'_\ell$ defined over $\mathbb{R}^{|\boldsymbol{\psi}_\ell|}$. Let `id` be the identity function and `extract_unused_rep` be defined by the algorithm given in Fig. 7. A bijective function $\phi'_\ell$ over $\mathbb{R}^{|\boldsymbol{\psi}_\ell|}$ mapping to $\mathbb{R}^{|\boldsymbol{\psi}_\ell|}$ can be defined as:

$$\phi'_\ell(\mathbf{h}) = \begin{cases} \phi_\ell(\mathbf{h}) & \texttt{if}\, \mathbf{h} \in \mathcal{H}^{\blacklozenge}_{\boldsymbol{\psi}_\ell} \cup \mathcal{H}^{\maltese}_{\boldsymbol{\psi}_\ell} \\ \texttt{extract_unused_rep}(\mathbf{h}, \mathcal{H}^{\blacklozenge}_{\boldsymbol{\psi}^{\phi}_\ell} \cup \mathcal{H}^{\maltese}_{\boldsymbol{\psi}^{\phi}_\ell}), & \texttt{if}\, \mathbf{h} \in (\mathcal{H}^{\blacklozenge}_{\boldsymbol{\psi}^{\phi}_\ell} \cup \mathcal{H}^{\maltese}_{\boldsymbol{\psi}^{\phi}_\ell}) \setminus (\mathcal{H}^{\blacklozenge}_{\boldsymbol{\psi}_\ell} \cup \mathcal{H}^{\maltese}_{\boldsymbol{\psi}_\ell}) \\ \mathbf{h} & \texttt{else} \end{cases} \tag{34}$$

which completes the proof. □

### F.1 Discussion about Assumptions

**Assump. 1 (Countable input-space).** While this assumption cannot be made on all neural networks like MLPs, it holds for models working on language and images. The set of all images with a specific resolution is finite, as it considers a finite number of pixels where each pixel has a finite number of channels (e.g. values for red, green and blue) and each channel is a number between 0 and 255. The set of all sequences in a language model is also countably infinite, as each set of sequences of some length is finite given finite tokens; so we have a set made out of the countable union of finite sets, which is still countable.

**Assump. 2 (Input-injectivity in all layers).** Neural network layers (e.g., MLP blocks) are not necessarily injective. The usage of learnable weights, activation functions like ReLU (Nair and Hinton, 2010) and information bottlenecks makes it possible to have a non-injective model. However, we prove in App. G that transformers, at least, are almost surely injective at initialisation on their inputs. Further, Nikolaou et al. (2025) recently published a proof—as well as empirical evidence—that transformers are almost surely injective in the hidden states of their last token, both at initialisation and after training. We also see in our empirical experiments in App. H that the MLPs we analyse are also, in practice, injective—or close enough to it that we observe no collisions in embedding space.

**Assump. 3 (Strict output-surjectivity in all layers).** Surjectivity can be defined on the output distribution $f^{\ell:}_{\texttt{N}} : \mathcal{H}_{\boldsymbol{\psi}_\ell} \to \Delta^{|\mathcal{Y}|-1}$, but that is a rather strong assumption. For our proofs, we will rely on strict surjectivity on the classification space instead ($\tau_{\boldsymbol{\eta}_{\mathbf{y}}} \circ f^{\ell:}_{\texttt{N}} : \mathcal{H}_{\boldsymbol{\psi}_\ell} \to |\mathcal{Y}|$), such that every class can be predicted. However, surjectivity on the classification space still does not necessarily hold for DNNs. LLMs have problems like the softmax bottleneck (Yang et al., 2018), which can lead to a model having insufficient capacity to predict all possible tokens. Grivas et al. (2022) also evaluate and find this problem, but show that surjectivity on the tokens is still likely in practice, making this a reasonable assumption in LLM settings.

**Assump. 4 (Algorithm and DNN have matchable partial-orderings).** We assume this since, for a neural network $\texttt{N}$ to be abstracted by the algorithm $\texttt{A}$, we need it to have this minimal width and depth.

**Assump. 5 (DNN solves the task).** We assume this because, if a neural network does not solve the given task, it will also not be abstracted by an algorithm which implements it.

### F.2 Detailed Version of Definition 7

Definition 7 can also be written without referring to previous definitions as following:

**Alternative Definition 1 (Equivalent to Definition 7).** *An algorithm* $\texttt{A}$ *is an* ***input-restricted distributed abstraction*** *of a neural network* $\texttt{N}$ *iff there exists an* $\tau$, $\boldsymbol{\mathcal{I}}_{\texttt{A}}$, *and* $\boldsymbol{\mathcal{I}}_{\texttt{N}}$ *such that*

- *$\tau$ is a distributed abstraction map. I.e., there exists an alignment map $\phi$, a latent-variable partition $\{\boldsymbol{\psi}^{\phi}_{\eta} \mid \eta \in \boldsymbol{\eta}_{\texttt{int}}\} \cup \{\boldsymbol{\psi}^{\phi}_{\perp}\}$ of $\boldsymbol{\psi}^{\phi}_{\texttt{int}}$ (with non-empty $\boldsymbol{\psi}^{\phi}_{\eta}$), and subabstraction maps $\{\tau_\eta \mid \eta \in \boldsymbol{\eta}_{\texttt{int}}\}$ such that $\tau$ is equivalent to computing the value of each node block-wise with $\tau_\eta(\mathbf{h}_{\boldsymbol{\psi}^{\phi}_{\eta}})$. An alignment map $\phi$ is a bijective function that maps the inner neurons $\boldsymbol{\psi}_{\texttt{int}}$ of $\texttt{N}$ onto an equal-sized set of latent variables $\boldsymbol{\psi}^{\phi}_{\texttt{int}}$, with $\phi$ respecting the network's computational order by being the combination of layer-wise bijections $\phi_\ell : \mathbb{R}^{|\boldsymbol{\psi}_\ell|} \to \mathbb{R}^{|\boldsymbol{\psi}_\ell|}$ applied to the neurons of each of the DNN's layers ($\ell$);*
- *$\boldsymbol{\mathcal{I}}_{\texttt{A}}$ and $\boldsymbol{\mathcal{I}}_{\texttt{N}}$ are a maximal input-restricted intervention set. A maximal input-restricted intervention set is composed of all interventions produced from other input-restricted interventions, i.e., it is a set with $\mathbf{h}_{\boldsymbol{\psi}^{\phi}} \leftarrow \mathbf{c}_{\boldsymbol{\psi}^{\phi}}$ (where $\boldsymbol{\psi}^{\phi} \subseteq \boldsymbol{\psi}^{\phi}_{\texttt{int}}$) or $\mathbf{v}_{\boldsymbol{\eta}} \leftarrow \mathbf{c}_{\boldsymbol{\eta}}$ (where $\boldsymbol{\eta} \subseteq \boldsymbol{\eta}_{\texttt{int}}$) where $\mathbf{c}_{\boldsymbol{\psi}^{\phi}}$ or $\mathbf{c}_{\boldsymbol{\eta}}$ arise from valid input-restricted computations (e.g., $\mathbf{c}_{\boldsymbol{\psi}^{\phi}} = f^{\boldsymbol{\psi}^{\phi}}_{\texttt{N}}(\mathbf{x})$ or $\mathbf{c}_{\boldsymbol{\eta}} = f^{:\boldsymbol{\eta}}_{\texttt{A}}(\mathbf{x}, \mathbf{c}_{\boldsymbol{\eta}} \leftarrow f^{:\boldsymbol{\eta}}_{\texttt{A}}(\mathbf{x}))$).*
- *$\tau$ is surjective;*
- *$\boldsymbol{\mathcal{I}}_{\texttt{A}} = \omega_\tau(\boldsymbol{\mathcal{I}}_{\texttt{N}})$;*
- *There exists a surjective $\tau_{\boldsymbol{\eta}_{\mathbf{x}}}$ such that*

$$\underset{\substack{\mathbf{x}\in\mathcal{X}\\ \mathbf{I}_{\texttt{N}}\in\boldsymbol{\mathcal{I}}_{\texttt{N}}}}{\forall} : \tau(f^{\boldsymbol{\psi}^{\phi}_{\texttt{int}}}_{\texttt{N}}(\mathbf{x}, \mathbf{I}_{\texttt{N}})) = f^{:\boldsymbol{\eta}_{\texttt{int}}}_{\texttt{A}}(\tau_{\boldsymbol{\eta}_{\mathbf{x}}}(\mathbf{x}), \mathbf{I}_{\texttt{A}}) \quad \texttt{where } \mathbf{I}_{\texttt{A}} = \omega_\tau(\mathbf{I}_{\texttt{N}}) \tag{35}$$

### F.3 Useful Definitions and Lemmas for Theorem 1

**Definition 10.** *We say the composition of a function $f : \mathbb{R}^d \to \Delta^{|\mathcal{Y}|-1}$ with* argmax *is **strictly surjective** if, for any output $\mathbf{y}^\star \in \mathcal{Y}$, there exists an input $\mathbf{h} \in \mathbb{R}^d$ for which $f$ outputs $\mathbf{y}^\star$ no matter how ties are broken in the* argmax*. Formally:*

$$\forall \mathbf{y}^\star \in \mathcal{Y}, \exists \mathbf{h} \in \mathbb{R}^d, \forall \mathbf{y}' \in \mathcal{Y} \setminus \{\mathbf{y}^\star\} : [f(\mathbf{h})]_{\mathbf{y}^\star} > [f(\mathbf{h})]_{\mathbf{y}'} \tag{36}$$

**Lemma 1.** *Let $\texttt{N}$ be a DNN and $\texttt{A}$ be an algorithm. Further, let $\tau$ be a distributed abstraction map with partition $\boldsymbol{\Psi}$ and $\{\tau_\eta\}_{\eta\in\boldsymbol{\eta}_{\texttt{int}}}$. If, for all $\eta \in \boldsymbol{\eta}_\ell$, $\tau_\eta$ satisfies the conditions in (1) (defined in Theorem 1's proof) applied on layer $\boldsymbol{\ell}$'s pre-intervention hidden variables, i.e., if:*

$$\underset{\substack{\mathbf{I}_{\texttt{N}}\in\boldsymbol{\mathcal{I}}^{:\ell-1}_{\texttt{N}}\\ \mathbf{x}\in\mathcal{X}}}{\forall} : \underbrace{\tau_\eta(\mathbf{h}'_{\boldsymbol{\psi}^{\phi}_{\eta}}) = f^{\eta}_{\texttt{A}}(\mathbf{x}, \mathbf{I}_{\texttt{A}}),}_{\texttt{tested condition}} \qquad \texttt{where } \underbrace{\mathbf{h}'_{\boldsymbol{\psi}^{\phi}_{\ell}} = f^{\boldsymbol{\psi}^{\phi}_{\ell}}_{\texttt{N}}(\mathbf{x}, \mathbf{I}_{\texttt{N}})),}_{\texttt{pre-int. hidden variable, as } \mathbf{I}_{\texttt{N}}\in\boldsymbol{\mathcal{I}}^{:\ell-1}_{\texttt{N}}} \texttt{ and } \mathbf{I}_{\texttt{A}} = \omega_\tau(\mathbf{I}_{\texttt{N}}) \tag{37}$$

*where we note that $f^{\boldsymbol{\psi}^{\phi}_{\ell}}_{\texttt{N}} = \phi_\ell \circ f^{:\ell}_{\texttt{N}}$ when no intervention is applied to layer $\ell$. Then $\tau_\eta$ also satisfies this condition when applied to layer $\ell$'s post-intervention hidden variables:*

$$\underset{\substack{\mathbf{I}_{\texttt{N}}\in\boldsymbol{\mathcal{I}}^{:\ell}_{\texttt{N}}\\ \mathbf{x}\in\mathcal{X}}}{\forall} : \underbrace{\tau_\eta(\mathbf{h}'_{\boldsymbol{\psi}^{\phi}_{\eta}}) = f^{\eta}_{\texttt{A}}(\mathbf{x}, \mathbf{I}_{\texttt{A}}),}_{\texttt{tested condition}} \qquad \texttt{where } \underbrace{\mathbf{h}'_{\boldsymbol{\psi}^{\phi}_{\ell}} = f^{\boldsymbol{\psi}^{\phi}_{\ell}}_{\texttt{N}}(\mathbf{x}, \mathbf{I}_{\texttt{N}})),}_{\texttt{post-int. hidden variable, as } \mathbf{I}_{\texttt{N}}\in\boldsymbol{\mathcal{I}}^{:\ell}_{\texttt{N}}} \texttt{ and } \mathbf{I}_{\texttt{A}} = \omega_\tau(\mathbf{I}_{\texttt{N}}) \tag{38}$$

*Proof.* Let $\tau_\eta$ be the abstraction map of $\eta \in \boldsymbol{\eta}_\ell$. By assumption, condition (1) holds for all pre-intervention hidden variables, i.e., hidden variables of the form $\mathbf{h}'_{\boldsymbol{\psi}^{\phi}_{\eta}} = [\phi_\ell(f^{:\ell}_{\texttt{N}}(\mathbf{x}, \mathbf{I}_{\texttt{N}}))]_{\boldsymbol{\psi}^{\phi}_{\eta}}$. We can show the same function applies to post-intervention hidden variables, i.e., hidden variables of the form:

$$\mathbf{h}'_{\boldsymbol{\psi}^{\phi}_{\eta}} = \begin{cases} \mathbf{c}_{\boldsymbol{\psi}^{\phi}_{\eta}} & \texttt{if } \mathbf{h}'_{\boldsymbol{\psi}^{\phi}_{\eta}} \leftarrow \mathbf{c}_{\boldsymbol{\psi}^{\phi}_{\eta}} \in \mathbf{I}_{\texttt{N}} \\ \left[\phi_\ell(f^{:\ell}_{\texttt{N}}(\mathbf{x}, \mathbf{I}_{\texttt{N}}))\right]_{\boldsymbol{\psi}^{\phi}_{\eta}} & \texttt{else} \end{cases} \tag{39}$$

Now let $\mathbf{I}_{\texttt{N}} \in \boldsymbol{\mathcal{I}}^{:\ell}_{\texttt{N}}$ be any intervention and $\mathbf{x} \in \mathcal{X}$ be any input. Further, let $\mathbf{I}_{\texttt{A}} = \omega_\tau(\mathbf{I}_{\texttt{N}})$. If $\mathbf{I}_{\texttt{N}} \in \boldsymbol{\mathcal{I}}^{:\ell-1}_{\texttt{N}}$, then the post-intervention hidden variable is identical to a pre-intervention one, and the conditions in (1) still hold, i.e.,: $\mathbf{h}'_{\boldsymbol{\psi}^{\phi}_{\eta}} = [\phi_\ell(f^{:\ell}_{\texttt{N}}(\mathbf{x}, \mathbf{I}_{\texttt{N}}))]_{\boldsymbol{\psi}^{\phi}_{\eta}}$ and $\tau_\eta$ is such that $\tau_\eta(\mathbf{h}'_{\boldsymbol{\psi}^{\phi}_{\eta}}) = f^{\eta}_{\texttt{A}}(\mathbf{x}, \mathbf{I}_{\texttt{A}})$. If $\mathbf{I}_{\texttt{N}} \notin \boldsymbol{\mathcal{I}}^{:\ell-1}_{\texttt{N}}$, for each node's hidden variables $\boldsymbol{\psi}^{\phi}_{\eta}$, we might or not intervene on it. If we do not intervene on node $\eta$, then we still have the case $\mathbf{h}'_{\boldsymbol{\psi}^{\phi}_{\eta}} = [\phi_\ell(f^{:\ell}_{\texttt{N}}(\mathbf{x}, \mathbf{I}_{\texttt{N}}))]_{\boldsymbol{\psi}^{\phi}_{\eta}}$ and thus $\tau_\eta$ still gives us the correct solution, i.e., $\tau_\eta(\mathbf{h}'_{\boldsymbol{\psi}^{\phi}_{\eta}}) = f^{\eta}_{\texttt{A}}(\mathbf{x}, \mathbf{I}_{\texttt{A}})$. If we intervene on $\eta$, then we know there exists an intervention of form

$\mathbf{h}_{\psi^{\phi}_{\eta}} \leftarrow f^{:\ell}_{\mathtt{N}}(\mathbf{x}', \mathbf{I}'_{\mathtt{N}})$ in $\mathbf{I}_{\mathtt{N}}$, for which $\mathbf{I}'_{\mathtt{N}} \in \mathcal{I}^{:\ell-1}_{\mathtt{N}}$, as our interventions are input-restricted. We also know (by eq. (4)) that there exists an equivalent intervention $v_\eta \leftarrow \tau_\eta(f^{:\ell}_{\mathtt{N}}(\mathbf{x}', \mathbf{I}'_{\mathtt{N}}))$ in $\mathbf{I}_{\mathtt{A}}$. We thus have that $\tau_\eta(\mathbf{h}'_{\psi^{\phi}_{\eta}}) = f^{\eta}_{\mathtt{A}}(\mathbf{x}, \mathbf{I}_{\mathtt{A}})$. □

**Lemma 2.** *Let* $\mathtt{N}$ *be a DNN and* $\mathtt{A}$ *be an algorithm. Further, let* $\tau$ *be a distributed abstraction map with partition* $\mathbf{\Psi}$*. If, for all* $\eta \in \boldsymbol{\eta}_{:\ell-1}$*, there exists a function* $g_\eta$ *which satisfies the conditions in* ③ *(defined in Theorem 1's proof) applied on layer* $\boldsymbol{\ell}$*'s pre-intervention hidden variables, i.e., if:*

$$\forall_{\substack{\mathbf{I}_{\mathtt{N}} \in \mathcal{I}^{:\ell-1}_{\mathtt{N}} \\ \mathbf{x} \in \mathcal{X}}} : \underbrace{g_\eta(\mathbf{h}'_{\psi^{\phi}_{\ell} \cap \psi^{\phi}_{\perp}}) = f^{\eta}_{\mathtt{A}}(\mathbf{x}, \mathbf{I}_{\mathtt{A}})}_{\texttt{tested condition}}, \qquad \texttt{where } \underbrace{\mathbf{h}'_{\psi^{\phi}_{\ell}} = f^{\psi^{\phi}_{\ell}}_{\mathtt{N}}(\mathbf{x}, \mathbf{I}_{\mathtt{N}})}_{\texttt{pre-int. hidden variable, as } \mathbf{I}_{\mathtt{N}} \in \mathcal{I}^{:\ell-1}_{\mathtt{N}}}, \texttt{ and } \mathbf{I}_{\mathtt{A}} = \omega_\tau(\mathbf{I}_{\mathtt{N}}) \tag{40}$$

*where we note that* $f^{\psi^{\phi}_{\ell}}_{\mathtt{N}} = \phi_\ell \circ f^{:\ell}_{\mathtt{N}}$ *when no intervention is applied to layer* $\ell$*. Then* $g_\eta$ *also satisfied this condition when applied to layer* $\ell$*'s post-intervention hidden variables:*

$$\forall_{\substack{\mathbf{I}_{\mathtt{N}} \in \mathcal{I}^{:\ell}_{\mathtt{N}} \\ \mathbf{x} \in \mathcal{X}}} : \underbrace{g_\eta(\mathbf{h}'_{\psi^{\phi}_{\ell} \cap \psi^{\phi}_{\perp}}) = f^{\eta}_{\mathtt{A}}(\mathbf{x}, \mathbf{I}_{\mathtt{A}})}_{\texttt{tested condition}}, \qquad \texttt{where } \underbrace{\mathbf{h}'_{\psi^{\phi}_{\ell}} = f^{\psi^{\phi}_{\ell}}_{\mathtt{N}}(\mathbf{x}, \mathbf{I}_{\mathtt{N}})}_{\texttt{post-int. hidden variable, as } \mathbf{I}_{\mathtt{N}} \in \mathcal{I}^{:\ell}_{\mathtt{N}}}, \texttt{ and } \mathbf{I}_{\mathtt{A}} = \omega_\tau(\mathbf{I}_{\mathtt{N}}) \tag{41}$$

*Proof.* Let $\eta \in \boldsymbol{\eta}_{:\ell-1}$ and $g_\eta$ be a function that satisfies condition ③ for it. Note that condition ③ holds for all pre-intervention hidden variables, i.e., hidden variables of the form $\mathbf{h}'_{\psi^{\phi}_{\ell} \cap \psi^{\phi}_{\perp}} = [\phi_\ell(f^{:\ell}_{\mathtt{N}}(\mathbf{x}, \mathbf{I}_{\mathtt{N}}))]_{\psi^{\phi}_{\ell} \cap \psi^{\phi}_{\perp}}$. We can show the same function $g_\eta$ also applies to post-intervention hidden variables, i.e., hidden variables of the form:

$$\mathbf{h}'_{\psi^{\phi}_{\ell} \cap \psi^{\phi}_{\perp}} = \begin{cases} \mathbf{c}_{\psi^{\phi}_{\ell} \cap \psi^{\phi}_{\perp}} & \texttt{if } \mathbf{h}'_{\psi^{\phi}_{\ell} \cap \psi^{\phi}_{\perp}} \leftarrow \mathbf{c}_{\psi^{\phi}_{\ell} \cap \psi^{\phi}_{\perp}} \in \mathbf{I}_{\mathtt{N}} \\ [\phi_\ell(f^{:\ell}_{\mathtt{N}}(\mathbf{x}, \mathbf{I}_{\mathtt{N}}))]_{\psi^{\phi}_{\ell} \cap \psi^{\phi}_{\perp}} & \texttt{else} \end{cases} \tag{42}$$

Now, let $\mathbf{I}_{\mathtt{N}} \in \mathcal{I}^{:\ell}_{\mathtt{N}}$, $\mathbf{x} \in \mathcal{X}$. Further, let $\mathbf{I}_{\mathtt{A}} = \omega_\tau(\mathbf{I}_{\mathtt{N}})$. If $\mathbf{I}_{\mathtt{N}} \in \mathcal{I}^{:\ell-1}_{\mathtt{N}}$, then the post-intervention hidden variable is identical to a pre-intervention one, and the conditions in ③ still hold, i.e.,: $\mathbf{h}'_{\psi^{\phi}_{\ell} \cap \psi^{\phi}_{\perp}} = [\phi_\ell(f^{:\ell}_{\mathtt{N}}(\mathbf{x}, \mathbf{I}_{\mathtt{N}}))]_{\psi^{\phi}_{\ell} \cap \psi^{\phi}_{\perp}}$ and $g_\eta$ is such that $g_\eta(\mathbf{h}'_{\psi^{\phi}_{\ell} \cap \psi^{\phi}_{\perp}}) = f^{\eta}_{\mathtt{A}}(\mathbf{x}, \mathbf{I}_{\mathtt{A}})$. If $\mathbf{I}_{\mathtt{N}} \notin \mathcal{I}^{:\ell-1}_{\mathtt{N}}$, it means that we intervene on at least one hidden variable in this layer $\psi^{\phi}_{\ell}$. However, we never intervene on neurons in $\psi^{\phi}_{\perp}$, meaning that for those we still have the case $\mathbf{h}'_{\psi^{\phi}_{\ell} \cap \psi^{\phi}_{\perp}} = [\phi_\ell(f^{:\ell}_{\mathtt{N}}(\mathbf{x}, \mathbf{I}_{\mathtt{N}}))]_{\psi^{\phi}_{\ell} \cap \psi^{\phi}_{\perp}}$ and thus the same function $g_\eta$ still satisfies our condition $g_\eta(\mathbf{h}'_{\psi^{\phi}_{\ell} \cap \psi^{\phi}_{\perp}}) = f^{\eta}_{\mathtt{A}}(\mathbf{x}, \mathbf{I}_{\mathtt{A}})$. □

**Lemma 3.** *Under Assump. 1 and given a fixed* $\phi$*, the set of input-restricted interventions* $\mathcal{I}_{\mathtt{N}}$ *is countable.*

*Proof.* This can be shown by induction. More specifically, we show that for any layer $\ell$, the set of input-restricted interventions $\mathcal{I}^{:\ell}_{\mathtt{N}}$ is countable for a specific $\phi$.

**Base Case ($\ell = 0$).** The base case can be proved trivially, as $\mathcal{I}^{:0}_{\mathtt{N}} = \{\emptyset\}$.

**Induction step ($\ell$ given $\ell - 1$).** By the induction hypothesis, $\mathcal{I}^{:\ell-1}_{\mathtt{N}}$ is countable. Now, note that $\mathcal{I}^{:\ell}_{\mathtt{N}}$ can be decomposed as:

$$\mathcal{I}^{:\ell}_{\mathtt{N}} = \mathcal{I}^{:\ell-1}_{\mathtt{N}} \times \mathcal{I}^{\ell}_{\mathtt{N}} \tag{43}$$

As the Cartesian product of two countable sets is itself countable, and as $\mathcal{I}^{:\ell-1}_{\mathtt{N}}$ is countable by the inductive hypothesis, we only need to show that $\mathcal{I}^{\ell}_{\mathtt{N}}$ is countable to complete our proof. This set $\mathcal{I}^{\ell}_{\mathtt{N}}$ is defined as the set of all input-restricted interventions to layer $\ell$. Given a set of neurons or hidden variables in this layer $\psi'$, we are thus dealing with interventions of the form: $\mathbf{h}_{\psi'} \leftarrow f^{\psi'}_{\mathtt{N}}(\mathbf{x}, \mathbf{I}_{\mathtt{N}})$, where: (i) $\psi' \subseteq \psi_\ell$ or $\psi' \subseteq \psi^{\phi}_{\ell}$; (ii) $\mathbf{x} \in \mathcal{X}$; and (iii) $\mathbf{I}_{\mathtt{N}} \in \mathcal{I}^{:\ell-1}_{\mathtt{N}}$. The set of all input-restricted interventions in this layer is thus bounded in size by the Cartesian product: $\bigtimes_{\psi \in \psi^{\phi}_{\ell}} \mathcal{X} \times \mathcal{I}^{:\ell-1}_{\mathtt{N}}$. These three sets are countable, and thus so is $\mathcal{I}^{\ell}_{\mathtt{N}}$. This concludes our proof. □

**Lemma 4.** *Under Assump. 1 and given a fixed* $\phi$*, the set of input-restricted pre-intervention hidden states in layer* $\ell$*, i.e.,* $\mathcal{H}^{\blacklozenge}_{\psi_\ell}$*, is countable.*

*Proof.* The set of input-restricted hidden states $\mathcal{H}^{\blacklozenge}_{\psi_\ell}$ is formed by hidden states $\mathbf{h}_{\psi_\ell} = f^{\psi_\ell}_{\mathtt{N}}(\mathbf{x}, \mathbf{I}_{\mathtt{N}})$, which we can write as:

$$\mathcal{H}^{\blacklozenge}_{\psi_\ell} = \{f^{\psi_\ell}_{\mathtt{N}}(\mathbf{x}, \mathbf{I}_{\mathtt{N}}) \mid \mathbf{x} \in \mathcal{X}, \mathbf{I}_{\mathtt{N}} \in \mathcal{I}^{:\ell-1}_{\mathtt{N}}\} \tag{44}$$

We thus have that the size of $\mathcal{H}^{\blacklozenge}_{\psi_\ell}$ is bounded by the size of the Cartesian product $\mathcal{X} \times \mathcal{I}^{:\ell}_{\mathtt{N}}$. As both of these sets are countable (by Assump. 1 and Lemma 3, respectively), $\mathcal{H}^{\blacklozenge}_{\psi_\ell}$ is also countable. This completes our proof. □

**Lemma 5.** *Under Assump. 1 and given a fixed $\phi$, the set of input-restricted intervention-only hidden variables in layer $\ell$, i.e., $\mathcal{H}^{\clubsuit}_{\psi^{\phi}_\ell}$, is countable.*

*Proof.* A similar proof to Lemma 4 applies here. In short, we have three relevant sets for this proof. First, the set of input-restricted pre-intervention hidden variables:

$$\mathcal{H}^{\blacklozenge}_{\psi^{\phi}_\ell} = \{\phi_\ell(\mathbf{h}_{\psi_\ell}) \mid \mathbf{h}_{\psi_\ell} \in \mathcal{H}^{\blacklozenge}_{\psi_\ell}\} \tag{45}$$

Second, we have the set of input-restricted post-intervention hidden variables:

$$\mathcal{H}_{\psi^{\phi}_\ell} = \left( \bigtimes_{\eta \in \boldsymbol{\eta}_\ell} \underbrace{\{\phi_\ell(\mathbf{h}_{\psi_\ell})_{\psi^{\phi}_\eta} \mid \mathbf{h}_{\psi_\ell} \in \mathcal{H}^{\blacklozenge}_{\psi_\ell}\}}_{\texttt{Pre-int. h.v., projected to } \psi^{\phi}_\eta} \right) \times \underbrace{\{\phi_\ell(\mathbf{h}_{\psi_\ell})_{\psi^{\phi}_\perp \cap \psi^{\phi}_\ell} \mid \mathbf{h}_{\psi_\ell} \in \mathcal{H}^{\blacklozenge}_{\psi_\ell}\}}_{\texttt{Pre-int. h.v., projected to } \psi^{\phi}_\perp} \tag{46}$$

Both sets above are countable, since $\mathcal{H}^{\blacklozenge}_{\psi_\ell}$ is countable (by Lemma 4), and $\boldsymbol{\eta}_\ell$ is finite. Third, we have the set of input-restricted intervention-only hidden variables, defined as:

$$\mathcal{H}^{\clubsuit}_{\psi^{\phi}_\ell} = \mathcal{H}_{\psi^{\phi}_\ell} \setminus \mathcal{H}^{\blacklozenge}_{\psi^{\phi}_\ell} \tag{47}$$

Since $\mathcal{H}_{\psi^{\phi}_\ell}$ is countable, $\mathcal{H}^{\clubsuit}_{\psi^{\phi}_\ell}$ is clearly also countable. This completes the proof. □

**Lemma 6.** *Under Assump. 3 and given a target output $\mathbf{y}^\star \in \mathcal{Y}$, we know that there is an uncountably infinite set $\mathcal{H}^{(\mathbf{y}^\star)}_{\psi_\ell}$ which predicts it, i.e.,:*

$$\mathbf{h} \in \mathcal{H}^{(\mathbf{y}^\star)}_{\psi_\ell} \Leftrightarrow \mathbf{y}^\star = \operatorname*{argmax}_{\mathbf{y}' \in \mathcal{Y}} [f^{\ell:}_{\mathtt{N}}(\mathbf{h})]_{\mathbf{y}'} \tag{48}$$

*Proof.* Under Assump. 3, we know that—for any target output $\mathbf{y}^\star \in \mathcal{Y}$—there is at least one hidden state $\mathbf{h}^\star \in \mathbb{R}^{|\psi_\ell|}$ which predicts it, i.e.:

$$\mathbf{y}^\star = \operatorname*{argmax}_{\mathbf{y}' \in \mathcal{Y}} [f^{\ell:}_{\mathtt{N}}(\mathbf{h}^\star)]_{\mathbf{y}'} \tag{49}$$

where we note that $f^{\ell:}_{\mathtt{N}}(\mathbf{h}^\star)$ outputs a probability distribution over $\mathcal{Y}$, i.e., $p_{\mathtt{N}}(\mathbf{y}' \mid \mathbf{h}^\star)$.

To show that we have an uncountably infinite set, let us first notice that

$$\forall \mathbf{h}' \in \mathbb{R}^{|\psi_\ell|} : ||f^{\ell:}_{\mathtt{N}}(\mathbf{h}) - f^{\ell:}_{\mathtt{N}}(\mathbf{h}')||_2 < \frac{m_1 - m_2}{2} \Rightarrow \mathbf{y}^\star = \operatorname*{argmax}_{\mathbf{y}' \in \mathcal{Y}} [f^{\ell:}_{\mathtt{N}}(\mathbf{h}')]_{\mathbf{y}'} \tag{50}$$

for $m_1$ be the max value of $f^{\ell:}_{\mathtt{N}}(\mathbf{h})$ and $m_2$ the second highest value of $f^{\ell:}_{\mathtt{N}}(\mathbf{h})$. $m_1 > m_2$ follows by the strict subjectivity mentioned in $Assump.$ 3. Eq. (50) follows by the definition of the euclidean norm ($||.||_2$), argmax and $f^{\ell:}_{\mathtt{N}}(\mathbf{h}') \in \Delta^{|\psi_L|-1}$ as $m_1$ has to be lowered at least $\frac{m_1-m_2}{2}$ to increase $m_2$ by $\frac{m_1-m_2}{2}$ for those two values to be the same. Increasing any other value in $f^{\ell:}_{\mathtt{N}}(\mathbf{h})$ would require $m_1$ being lowered more than $\frac{m_1-m_2}{2}$ or any other value increased by more than $\frac{m_1-m_2}{2}$. Now, given continuity of neural networks, we know that:

$$\begin{aligned} &\forall \epsilon > 0, \exists \delta > 0, \forall \mathbf{h}' \in \mathbb{R}^{|\psi_\ell|} : 0 < ||\mathbf{h} - \mathbf{h}'||_2 < \delta, \\ &\quad \Rightarrow ||f^{\ell:}_{\mathtt{N}}(\mathbf{h}) - f^{\ell:}_{\mathtt{N}}(\mathbf{h}')||_2 < \epsilon. \end{aligned} \tag{51}$$

Therefore, we see that:

$$\exists \delta > 0, \forall \mathbf{h}' \in \mathbb{R}^{|\boldsymbol{\psi}_\ell|} : 0 < ||\mathbf{h} - \mathbf{h}'||_2 < \delta,$$
$$\Rightarrow ||f_{\mathtt{N}}^{\ell:}(\mathbf{h}) - f_{\mathtt{N}}^{\ell:}(\mathbf{h}')||_2 < \frac{m_1 - m_2}{2} \tag{52a}$$
$$\Rightarrow \mathbf{y}^\star = \operatorname*{argmax}_{\mathbf{y}' \in \mathcal{Y}} [f_{\mathtt{N}}^{\ell:}(\mathbf{h}')]_{\mathbf{y}'} \tag{52b}$$

We notice that $||\mathbf{h} - \mathbf{h}'||_2 < \delta$ for $\delta > 0$ denotes a continuous region in $\mathbb{R}^{|\boldsymbol{\psi}_\ell|}$ which therefore includes uncountably infinite points. □

## G Transformers at Initialisation are Almost Surely Injective on each Layer

**Theorem 2.** *Transformers like DNN 2 with randomly independent initialised from a continuous distribution (riicd.) weights are almost surely injective at initialisation up to each layer* $0 \leq \ell < L$.

*Proof.* To show injectivity up to a layer $\ell'$ in a transformer, it suffices to show that $f_{\mathtt{N}}^{\ell}$ is injective on any countable subset $\mathcal{H}$ of its domain for all layers $\ell$ $(0 \leq \ell \leq \ell')$. This suffices as we assume the set of inputs $\mathcal{X}$ is countable, and the composition of injective functions is injective. Let $\Theta$ be the random variable representing the transformer's weights. To show (almost sure) injectivity on layer $\ell$ for any fixed input set $\mathcal{H}$, we need that (because of Lemma 10):[14]

$$p_\Theta\left(\forall \mathbf{h}_1, \mathbf{h}_2 \in \mathcal{H}, \mathbf{h}_1 \neq \mathbf{h}_2 : f_{\mathtt{N}}^{\ell}(\mathbf{h}_1) \neq f_{\mathtt{N}}^{\ell}(\mathbf{h}_2)\right) = 1 \tag{53}$$

Since the transformer operates over sequences of tokens, any element $\mathbf{h} \in \mathcal{H}$ has its first dimension indexing the sequence length. Let $|\mathbf{h}|$ denote the sequence length and $[\mathbf{h}]_t$ refer to the $t$-th element in $\mathbf{h}$. Let $T$ be the set of token positions $T = \{1, \ldots, \mathtt{min}(|\mathbf{h}_1|, |\mathbf{h}_2|)\}$. For injectivity, it suffices to show that:

$$p_\Theta(\forall \mathbf{h}_1, \mathbf{h}_2 \in \mathcal{H}, t \in T, [\mathbf{h}_1]_t \neq [\mathbf{h}_2]_t : [f_{\mathtt{N}}^{\ell}(\mathbf{h}_1)]_t \neq [f_{\mathtt{N}}^{\ell}(\mathbf{h}_2)]_t) = 1 \tag{54}$$

Note that eq. (54) only ensures injectivity when $|\mathbf{h}_1| = |\mathbf{h}_2|$. However, this is sufficient because when $|\mathbf{h}_1| \neq |\mathbf{h}_2|$, eq. (53) follows trivially: since $|f_{\mathtt{N}}^{\ell}(\mathbf{h}_1)| \neq |f_{\mathtt{N}}^{\ell}(\mathbf{h}_2)|$, we immediately have $f_{\mathtt{N}}^{\ell}(\mathbf{h}_1) \neq f_{\mathtt{N}}^{\ell}(\mathbf{h}_2)$. When $|\mathbf{h}_1| = |\mathbf{h}_2|$, we can show that eq. (54) implies eq. (53) as follows: if $\mathbf{h}_1 \neq \mathbf{h}_2$, then there exists at least one token position $t' \in T$ where $[\mathbf{h}_1]_{t'} \neq [\mathbf{h}_2]_{t'}$. By eq. (54), this implies $[f_{\mathtt{N}}^{\ell}(\mathbf{h}_1)]_{t'} \neq [f_{\mathtt{N}}^{\ell}(\mathbf{h}_2)]_{t'}$ almost surely, and therefore $f_{\mathtt{N}}^{\ell}(\mathbf{h}_1) \neq f_{\mathtt{N}}^{\ell}(\mathbf{h}_2)$ almost surely.

We observe that a transformer's input set $\mathcal{X}$ consists of all sequences formed from a finite token vocabulary, which is countably infinite. Since transformers are deterministic functions, the input set $\mathcal{H}$ encountered at any sublayer is also countably infinite. Therefore, it suffices to prove eq. (54) for any fixed countably infinite input set $\mathcal{H}$.

We show that Eq. (54) holds for any fixed countably infinite subset $\mathcal{H}$ of the layer's domain. This is established for the embedding layer ($f_{\mathtt{N}}^{\ell}(\mathbf{h}) = \mathbf{e}_{\mathbf{h}}$), the MLP layer ($f_{\mathtt{N}}^{\ell}(\mathbf{h}) = \mathbf{h} + \mathtt{mlp}(\mathtt{LN}(\mathbf{h}))$), and the attention layer ($f_{\mathtt{N}}^{\ell}(\mathbf{h}) = \mathbf{h} + \mathtt{attn}(\mathtt{LN}(\mathbf{h}))$) by Lemma 7, Lemma 8, and Lemma 9, respectively. □

The 3 theorems facilitating the proof above are:

**Lemma 7.** *Lets assume we have an embedding layer randomly independent initialized from a continuous distribution (riicd.) weights and any countably infinite input sets (in embeddings token indexes). We denote the set of random variables over the weights as* $\Theta$*. We then can show for any fixed countably infinite input set* $\mathcal{H}$ *that this Layer is injective almost surely.*

$$p_\Theta(\forall \mathbf{h}_1, \mathbf{h}_2 \in \mathcal{H}, t \in T, [\mathbf{h}_1]_t \neq [\mathbf{h}_2]_t : [\mathbf{e}_{\mathbf{h}_1}]_t \neq [\mathbf{e}_{\mathbf{h}_2}]_t) = 1 \tag{55}$$

*Proof.* See App. G.2. □

**Lemma 8.** *Lets assume we have a sub-block consisting of an MLP with a residual connection and layer norm (i.e.,* $\mathbf{h} + (\mathtt{mlp}(\mathtt{LN}(\mathbf{h})))$ *with riicd. weights. We can show that, for any fixed countably*

[14] We note that $p_Z(\forall z \in \mathcal{Z} : z)$, where $\mathcal{Z}$ is a set of events, is the same as $p_Z(\cap_{z \in \mathcal{Z}} \{z\})$ formally.

*infinite input set $\mathcal{H}$, this layer is injective almost surely:*

$$p_\Theta(\forall \mathbf{h}_1, \mathbf{h}_2 \in \mathcal{H}, t \in T, [\mathbf{h}_1]_t \neq [\mathbf{h}_2]_t : \\ [\mathbf{h}_1 + \mathtt{mlp}(\mathtt{LN}(\mathbf{h}_1))]_t \neq [\mathbf{h}_2 + \mathtt{mlp}(\mathtt{LN}(\mathbf{h}_2))]_t) = 1 \tag{56}$$

*Proof.* See App. G.3. □

**Lemma 9.** *Lets assume we have a sub-block consisting of a self-attention with a residual connection and layer norm (i.e.,* $\mathbf{h} + \mathtt{attn}(\mathtt{LN}(\mathbf{h}))$*) with riicd. weights. We can show that, for any fixed countably infinite input set $\mathcal{H}$, this layer is injective almost surely:*

$$p_\Theta(\forall \mathbf{h}_1, \mathbf{h}_2 \in \mathcal{H}, t \in T, [\mathbf{h}_1]_t \neq [\mathbf{h}_2]_t : \\ [\mathbf{h}_1 + \mathtt{attn}(\mathtt{LN}(\mathbf{h}_1))]_t \neq [\mathbf{h}_2 + \mathtt{attn}(\mathtt{LN}(\mathbf{h}_2))]_t) = 1 \tag{57}$$

*Proof.* See App. G.4 □

### G.1 Fundamental Lemmas

In this section we present some fundamental lemmas used to prove G.2 to G.4.

**Lemma 10.** *For a layers function $f^\ell_{\mathtt{N}}$ to be injective on its input set $\mathcal{H}$, it has to hold that:*

$$p_\Theta(\forall \mathbf{h}_1, \mathbf{h}_2 \in \mathcal{H} : \mathbf{h}_1 \neq \mathbf{h}_2 \Rightarrow f^\ell_{\mathtt{N}}(\mathbf{h}_1) \neq f^\ell_{\mathtt{N}}(\mathbf{h}_2)) = 1 \tag{58}$$

*This can equivalently be written as:*

$$p_\Theta\left(\forall \mathbf{h}_1, \mathbf{h}_2 \in \mathcal{H}, \mathbf{h}_1 \neq \mathbf{h}_2 : f^\ell_{\mathtt{N}}(\mathbf{h}_1) \neq f^\ell_{\mathtt{N}}(\mathbf{h}_2)\right) = 1 \tag{59}$$

*Proof.* We can derive eq. (59) from eq. (58):

$$p_\Theta(\forall \mathbf{h}_1, \mathbf{h}_2 \in \mathcal{H} : \mathbf{h}_1 \neq \mathbf{h}_2 \Rightarrow f^\ell_{\mathtt{N}}(\mathbf{h}_1) \neq f^\ell_{\mathtt{N}}(\mathbf{h}_2))$$

$$= p_\Theta\left(\bigcap_{\mathbf{h}_1, \mathbf{h}_2 \in \mathcal{H}} \{\mathbf{h}_1 \neq \mathbf{h}_2 \Rightarrow f^\ell_{\mathtt{N}}(\mathbf{h}_1) \neq f^\ell_{\mathtt{N}}(\mathbf{h}_2)\}\right) \tag{60a}$$

$$= p_\Theta\left(\bigcap_{\mathbf{h}_1, \mathbf{h}_2 \in \mathcal{H}} \{\left((\mathbf{h}_1 \neq \mathbf{h}_2) \wedge f^\ell_{\mathtt{N}}(\mathbf{h}_1) \neq f^\ell_{\mathtt{N}}(\mathbf{h}_2)\right) \vee (\mathbf{h}_1 = \mathbf{h}_2)\}\right) \tag{60b}$$

$$= p_\Theta\left(\bigcap_{\mathbf{h}_1, \mathbf{h}_2 \in \mathcal{H}, \mathbf{h}_1 \neq \mathbf{h}_2} \{(\mathbf{h}_1 \neq \mathbf{h}_2) \wedge f^\ell_{\mathtt{N}}(\mathbf{h}_1) \neq f^\ell_{\mathtt{N}}(\mathbf{h}_2)\} \cap \underbrace{\bigcap_{\mathbf{h}_1, \mathbf{h}_2 \in \mathcal{H}, \mathbf{h}_1 = \mathbf{h}_2} \{(\mathbf{h}_1 = \mathbf{h}_2)\}}_{\texttt{always true}}\right) \tag{60c}$$

$$= p_\Theta\left(\bigcap_{\mathbf{h}_1, \mathbf{h}_2 \in \mathcal{H}, \mathbf{h}_1 \neq \mathbf{h}_2} \left\{\underbrace{(\mathbf{h}_1 \neq \mathbf{h}_2)}_{\texttt{always true}} \wedge f^\ell_{\mathtt{N}}(\mathbf{h}_1) \neq f^\ell_{\mathtt{N}}(\mathbf{h}_2)\right\}\right) \tag{60d}$$

$$= p_\Theta\left(\bigcap_{\mathbf{h}_1, \mathbf{h}_2 \in \mathcal{H}, \mathbf{h}_1 \neq \mathbf{h}_2} \{f^\ell_{\mathtt{N}}(\mathbf{h}_1) \neq f^\ell_{\mathtt{N}}(\mathbf{h}_2)\}\right) \tag{60e}$$

$$= p_\Theta\left(\forall \mathbf{h}_1, \mathbf{h}_2 \in \mathcal{H}, \mathbf{h}_1 \neq \mathbf{h}_2 : f^\ell_{\mathtt{N}}(\mathbf{h}_1) \neq f^\ell_{\mathtt{N}}(\mathbf{h}_2)\right) \tag{60f}$$

□

**Lemma 11.** *If we have a countable set $\mathcal{Z}$ of almost sure events $z$, we know that their intersection is also almost surely. Formally:*

$$\left(\forall z \in \mathcal{Z} : p(z) = 1\right) \implies \left(p(\forall z \in \mathcal{Z} : z) = 1\right) \tag{61}$$

*Proof.* First, observe that

$$p\left(\bigcap_{z\in\mathcal{Z}} z\right) = 1 - p\left(\left(\bigcap_{z\in\mathcal{Z}} z\right)^c\right) = 1 - p\left(\bigcup_{z\in\mathcal{Z}} z^c\right) \tag{62}$$

where $z^c$ is the complement of an event $z$. Since $p(z) = 1$ for all $z \in \mathcal{Z}$, it follows that $p(z^c) = 0$ for all $z \in \mathcal{Z}$ By the countable subadditivity of probability measures:

$$p\left(\bigcup_{z\in\mathcal{Z}} z^c\right) \leq \sum_{z\in\mathcal{Z}} p(z^c) = \sum_{z\in\mathcal{Z}} 0 = 0 \tag{63}$$

Therefore,

$$p(\forall z \in \mathcal{Z} : z) = p\left(\bigcap_{z\in\mathcal{Z}} z\right) = 1 - 0 = 1 \,\square \tag{64}$$

### G.2 Proof of Lemma 7

In this section, we will prove Lemma 7 which states that the embedding layer is almost surely injective on countably infinite inputs.

**Lemma 7.** *Lets assume we have an embedding layer randomly independent initialized from a continuous distribution (riicd.) weights and any countably infinite input sets (in embeddings token indexes). We denote the set of random variables over the weights as $\Theta$. We then can show for any fixed countably infinite input set $\mathcal{H}$ that this Layer is injective almost surely.*

$$p_\Theta(\forall \mathbf{h}_1, \mathbf{h}_2 \in \mathcal{H}, t \in T, [\mathbf{h}_1]_t \neq [\mathbf{h}_2]_t : [\mathbf{e}_{\mathbf{h}_1}]_t \neq [\mathbf{e}_{\mathbf{h}_2}]_t) = 1 \tag{55}$$

*Proof.* We can apply Lemma 11 three times (on $\mathbf{h}_1, \mathbf{h}_2$ and $t$) to show that eq. (55) is equivalent to, for any $\mathbf{h}_1, \mathbf{h}_2 \in \mathcal{H}$ and $t \in T$ for which $[\mathbf{h}_1]_t \neq [\mathbf{h}_2]_t$, it holding that:

$$p_\Theta([\mathbf{e}_{\mathbf{h}_1}]_t \neq [\mathbf{e}_{\mathbf{h}_2}]_t) = 1 \tag{65}$$

We further note that, by the definition of an embedding block (Submodule 2):

$$p_\Theta([\mathbf{e}_{\mathbf{h}_1}]_t \neq [\mathbf{e}_{\mathbf{h}_2}]_t) = 1 \quad \Leftrightarrow \quad p_\Theta(\mathbf{e}_{[\mathbf{h}_1]_t} \neq \mathbf{e}_{[\mathbf{h}_2]_t}) = 1 \tag{66}$$

We can thus apply the law of total probability by defining $\Theta'$ as all the random variables $\Theta$ except the one for the first element of $\mathbf{e}_{[\mathbf{h}_1]_t}$, i.e., except $[\mathbf{e}_{[\mathbf{h}_1]_t}]_1$,[15] and $\mathbf{e}'_{[\mathbf{h}_1]_t}$ as the embedding of $[\mathbf{h}_1]_t$ without the first element:

$$\int p_{\Theta\setminus\Theta'}(\mathbf{e}_{[\mathbf{h}_1]_t} \neq \mathbf{e}_{[\mathbf{h}_2]_t} \mid \mathbf{e}'_{[\mathbf{h}_1]_t}, \mathbf{e}_{[\mathbf{h}_2]_t}) \mathrm{p}_{\Theta'}(\mathbf{e}'_{[\mathbf{h}_1]_t} \cup \mathbf{e}_{[\mathbf{h}_2]_t}) d(\mathbf{e}'_{[\mathbf{h}_1]_t} \cup \mathbf{e}_{[\mathbf{h}_2]_t}) = 1 \tag{67}$$

It therefore suffices to show that, for any $\mathbf{e}'_{[\mathbf{h}_1]_t}$ and $\mathbf{e}_{[\mathbf{h}_2]_t}$:

$$p_{\Theta\setminus\Theta'}(\mathbf{e}_{[\mathbf{h}_1]_t} \neq \mathbf{e}_{[\mathbf{h}_2]_t} \mid \mathbf{e}'_{[\mathbf{h}_1]_t}, \mathbf{e}_{[\mathbf{h}_2]_t}) = 1 \tag{68}$$

This holds trivially when any embedding dimension other than the first of $\mathbf{e}_{[\mathbf{h}_1]_t}$ and $\mathbf{e}_{[\mathbf{h}_2]_t}$ differs. When all dimensions except the first are equal, we apply:

$$\begin{aligned} & p_{\Theta\setminus\Theta'}([\mathbf{e}_{[\mathbf{h}_1]_t}]_1 \neq [\mathbf{e}_{[\mathbf{h}_2]_t}]_1 \mid \mathbf{e}'_{[\mathbf{h}_1]_t}, \mathbf{e}_{[\mathbf{h}_2]_t}) = 1 \\ \Leftrightarrow \quad & p_{\Theta\setminus\Theta'}([\mathbf{e}_{[\mathbf{h}_1]_t}]_1 = [\mathbf{e}_{[\mathbf{h}_2]_t}]_1 \mid \mathbf{e}'_{[\mathbf{h}_1]_t}, \mathbf{e}_{[\mathbf{h}_2]_t}\}) = 0 \end{aligned} \tag{69}$$

The right-hand side $[\mathbf{e}_{[\mathbf{h}_2]_t}]_1$ is a constant while the left-hand side $[\mathbf{e}_{[\mathbf{h}_1]_t}]_1$ is a random variable over a continuous region; this event has measure 0, resulting in probability 0. $\square$

### G.3 Proof of Lemma 8

In this section, we will prove Lemma 8, which will show that the block consisting of an MLP, residual connection and layer norm is almost sure injective on its countably infinite inputs.

**Lemma 8.** *Lets assume we have a sub-block consisting of an MLP with a residual connection and layer norm (i.e., $\mathbf{h} + (\mathtt{mlp}(\mathtt{LN}(\mathbf{h})))$) with riicd. weights. We can show that, for any fixed countably*

[15] By this we refer to the first element of the embedding vector of $[\mathbf{h}_1]_t$.

*infinite input set $\mathcal{H}$, this layer is injective almost surely:*

$$p_\Theta(\forall \mathbf{h}_1, \mathbf{h}_2 \in \mathcal{H}, t \in T, [\mathbf{h}_1]_t \neq [\mathbf{h}_2]_t : \\ [\mathbf{h}_1 + \mathtt{mlp}(\mathtt{LN}(\mathbf{h}_1))]_t \neq [\mathbf{h}_2 + \mathtt{mlp}(\mathtt{LN}(\mathbf{h}_2))]_t) = 1 \tag{56}$$

*Proof.* For notational convenience, let $\mathbf{m}_i = \mathtt{mlp}(\mathtt{LN}(\mathbf{h}_i))$. Given Lemma 11, it suffices to prove that for any $\mathbf{h}_1, \mathbf{h}_2 \in \mathcal{H}$ and $t \in T$, where $[\mathbf{h}_1]_t \neq [\mathbf{h}_2]_t$, we have:

$$p_\Theta\Big([\mathbf{h}_1 + \mathbf{m}_1]_t \neq [\mathbf{h}_2 + \mathbf{m}_2]_t\Big) = 1 \tag{70}$$

Without loss of generality, fix one such $\mathbf{h}_1, \mathbf{h}_2 \in \mathcal{H}$ and $t \in T$. We can manipulate this probability distribution as:

$$\begin{aligned}
&p_\Theta\Big([\mathbf{h}_1 + \mathbf{m}_1]_t \neq [\mathbf{h}_2 + \mathbf{m}_2]_t\Big) \\
&= p_\Theta\Big([\mathbf{h}_1]_t + [\mathbf{m}_1]_t \neq [\mathbf{h}_2]_t + [\mathbf{m}_2]_t\Big) \\
&= p_\Theta\Big([\mathbf{m}_1]_t \neq [\mathbf{m}_2]_t\Big)\, p_\Theta\Big([\mathbf{h}_1]_t + [\mathbf{m}_1]_t \neq [\mathbf{h}_2]_t + [\mathbf{m}_2]_t \mid [\mathbf{m}_1]_t \neq [\mathbf{m}_2]_t\Big) &\text{(71a)} \\
&\quad + p_\Theta\Big([\mathbf{m}_1]_t = [\mathbf{m}_2]_t)\Big)\, \underbrace{p_\Theta\Big([\mathbf{h}_1]_t + [\mathbf{m}_1]_t \neq [\mathbf{h}_2]_t + [\mathbf{m}_2]_t \mid [\mathbf{m}_1]_t = [\mathbf{m}_2]_t\Big)}_{=1,\text{ since } [\mathbf{h}_1]_t \neq [\mathbf{h}_2]_t} \\
&= p_\Theta\Big([\mathbf{m}_1]_t \neq [\mathbf{m}_2]_t\Big) p_\Theta\Big([\mathbf{h}_1]_t + [\mathbf{m}_1]_t \neq [\mathbf{h}_2]_t + [\mathbf{m}_2]_t \mid [\mathbf{m}_1]_t \neq [\mathbf{m}_2]_t\Big) &\text{(71b)} \\
&\quad + p_\Theta\Big([\mathbf{m}_1]_t = [\mathbf{m}_2]_t)\Big)
\end{aligned}$$

Therefore, it suffices to show that:

$$p_\Theta\Big([\mathbf{h}_1]_t + [\mathbf{m}_1]_t \neq [\mathbf{h}_2]_t + [\mathbf{m}_2]_t \mid [\mathbf{m}_1]_t \neq [\mathbf{m}_2]_t\Big) = 1 \tag{72}$$

since $p_\Theta\Big([\mathbf{m}_1]_t \neq [\mathbf{m}_2]_t)\Big) + p_\Theta\Big([\mathbf{m}_1]_t = [\mathbf{m}_2]_t)\Big)$ is trivially 1. We now unfold the last layer of the MLP as $\mathbf{m}_i = \mathbf{W}_L(\mathbf{m}'_i) + \mathbf{b}_L$, where $\mathbf{m}'_i = \sigma(f^{:L-1}_{\mathtt{N}_{\mathtt{MLP}}}(\mathtt{LN}(\mathbf{h}_i)))$. We can rewrite eq. (72) as:

$$\begin{aligned}
&p_\Theta\Big([\mathbf{h}_1]_t + [\mathbf{W}_L\mathbf{m}'_1 + \mathbf{b}_L]_t \neq [\mathbf{h}_2]_t + [\mathbf{W}_L\mathbf{m}'_2 + \mathbf{b}_L]_t \mid [\mathbf{m}_1]_t \neq [\mathbf{m}_2]_t\Big) \\
&= p_\Theta\Big([\mathbf{h}_1]_t + \mathbf{W}_L[\mathbf{m}'_1]_t \neq [\mathbf{h}_2]_t + \mathbf{W}_L[\mathbf{m}'_2]_t \mid [\mathbf{m}_1]_t \neq [\mathbf{m}_2]_t\Big) &\text{(73a)} \\
&\overset{(1)}{=} p_\Theta\Big([\mathbf{h}_1]_t + \mathbf{W}_L[\mathbf{m}'_1]_t \neq [\mathbf{h}_2]_t + \mathbf{W}_L[\mathbf{m}'_2]_t \mid [\mathbf{m}'_1]_{[t,i]} \neq [\mathbf{m}'_2]_{[t,i]}\Big) &\text{(73b)} \\
&\overset{(2)}{\geq} p_\Theta\Big([\mathbf{h}_1]_{[t,1]} + [\mathbf{W}_L\mathbf{m}'_1]_{[t,1]} \neq [\mathbf{h}_2]_{[t,1]} + [\mathbf{W}_L\mathbf{m}'_2]_{[t,1]} \mid [\mathbf{m}'_1]_{[t,i]} \neq [\mathbf{m}'_2]_{[t,i]}\Big) &\text{(73c)} \\
&= p_\Theta\Big([\mathbf{h}_1]_{[t,1]} + \sum_{j=1}^{|[\mathbf{h}_1]_t|}[\mathbf{W}_L]_{[1,j]}[\mathbf{m}'_1]_{[t,j]} \neq [\mathbf{h}_2]_{[t,1]} + \sum_{j=1}^{|[\mathbf{h}_2]_t|}[\mathbf{W}_L]_{[1,j]}[\mathbf{m}'_2]_{[t,j]} \mid [\mathbf{m}'_1]_{[t,i]} \neq [\mathbf{m}'_2]_{[t,i]}\Big) &\text{(73d)} \\
&\overset{(3)}{=} \int_{\Theta\setminus\Theta'} p_{\Theta'}\Big([\mathbf{h}_1]_{[t,1]} + \sum_{j=1}^{|[\mathbf{h}_1]_t|}[\mathbf{W}_L]_{[1,j]}[\mathbf{m}'_1]_j \neq [\mathbf{h}_2]_{[t,1]} + &\text{(73e)} \\
&\qquad \sum_{j=1}^{|[\mathbf{h}_2]_t|}[\mathbf{W}_L]_{[1,j]}[\mathbf{m}'_2]_{[t,j]} \mid [\mathbf{m}'_1]_{[t,i]} \neq [\mathbf{m}'_2]_{[t,i]}, \theta\Big)\, \mathrm{p}_{\Theta\setminus\Theta'}(\theta \mid [\mathbf{m}'_1]_{[t,i]} \neq [\mathbf{m}'_2]_{[t,i]})d\theta
\end{aligned}$$

where equality (1) holds since $[\mathbf{m}_1]_t \neq [\mathbf{m}_2]_t$ implies there exists some index $i$ such that $[\mathbf{m}'_1]_{[t,i]} \neq [\mathbf{m}'_2]_{[t,i]}$..[16] (2) holds because if the inequality is satisfied for a single component of the vector, it must also be satisfied for the entire vector. In (3), we define $\Theta'$ as the random variable responsible for the value of $[\mathbf{W}_L]_{[1,i]}$ and $\theta$ as a realisation of the random variables $\Theta \setminus \Theta'$. Therefore, to prove

[16] $[\mathbf{m}'_1]_{[t,i]}$ represents a two dimensional indexing, referring to the $i$-th element of the representation of the $t$-th token.

eq. (72), it suffices to show:

$$p_{\Theta'}\Big([\mathbf{h}_1]_{[t,1]} + \sum_{j=1}^{|[\mathbf{h}_1]_t|}[\mathbf{W}_L]_{[1,j]}[\mathbf{m}'_1]_j \neq [\mathbf{h}_2]_{[t,1]} + \sum_{j=1}^{|[\mathbf{h}_2]_t|}[\mathbf{W}_L]_{[1,j]}[\mathbf{m}'_2]_{[t,j]} \mid [\mathbf{m}'_1]_{[t,i]} \neq [\mathbf{m}'_2]_{[t,i]}, \theta\Big) = 1 \tag{74}$$

For brevity, we omit repeating the conditions in the following probabilities as they remain unchanged to the previous equation:

$$p_{\Theta'}\Big([\mathbf{h}_1]_{[t,1]} + \sum_{j=1}^{|[\mathbf{h}_1]_t|}[\mathbf{W}_L]_{[1,j]}[\mathbf{m}'_1]_j \neq [\mathbf{h}_2]_{[t,1]} + \sum_{j=1}^{|[\mathbf{h}_2]_t|}[\mathbf{W}_L]_{[1,j]}[\mathbf{m}'_2]_{[t,j]} \mid \dots\Big) = 1 \tag{75a}$$

$$\Leftrightarrow p_{\Theta'}\Big([\mathbf{h}_1]_{[t,1]} + \sum_{j=1}^{|[\mathbf{h}_1]_t|}[\mathbf{W}_L]_{[1,j]}[\mathbf{m}'_1]_j = [\mathbf{h}_2]_{[t,1]} + \sum_{j=1}^{|[\mathbf{h}_2]_t|}[\mathbf{W}_L]_{[1,j]}[\mathbf{m}'_2]_{[t,j]} \mid \dots\Big) = 0 \tag{75b}$$

$$\Leftrightarrow p_{\Theta'}\Big([\mathbf{W}_L]_{[1,i]}[\mathbf{m}'_1]_{[t,i]} - [\mathbf{W}_L]_{[1,i]}[\mathbf{m}'_2]_{[t,i]} = [\mathbf{h}_2]_{[t,1]} - [\mathbf{h}_1]_{[t,1]} + \sum_{j=1, j\neq i}^{|[\mathbf{h}_1]_t|}[\mathbf{W}_L]_{[1,j]}[\mathbf{m}'_2]_{[t,j]} - \sum_{j=1, j\neq i}^{|[\mathbf{h}_2]_t|}[\mathbf{W}_L]_{[1,j]}[\mathbf{m}'_1]_{[t,j]} \mid \dots\Big) = 0 \tag{75c}$$

$$\Leftrightarrow p_{\Theta'}\Big([\mathbf{W}_L]_{[1,i]} = \frac{1}{[\mathbf{m}'_1]_{[t,i]} - [\mathbf{m}'_2]_{[t,i]}}\Big([\mathbf{h}_2]_{[t,1]} - [\mathbf{h}_1]_{[t,1]} + \sum_{j=1, j\neq i}^{|[\mathbf{h}_2]_t|}[\mathbf{W}_L]_{[1,j]}[\mathbf{m}'_2]_{[t,j]} - \sum_{j=1, j\neq i}^{|[\mathbf{h}_1]_t|}[\mathbf{W}_L]_{[1,j]}[\mathbf{m}'_1]_{[t,j]}\Big) \mid \dots\Big) = 0 \tag{75d}$$

where the last step follows from the condition $[\mathbf{m}'_1]_{[t,i]} \neq [\mathbf{m}'_2]_{[t,i]}$ which ensures the denominator is non-zero. Now, Eq. (75d) holds because the right-hand side is a constant (since its elements are fixed given the conditions of the probability) while the left-hand side is a random variable drawn from a continuous distribution (since the weights are riicd.). Therefore, the probability that this equality holds is zero, as the event has measure zero. □

### G.4 Proof of Lemma 9

In this Section, we prove Lemma 9, which establishes that the self-attention sub-block (consisting of attention, residual connection, and layer normalisation) is almost surely injective on countably infinite inputs. The proof structure parallels that of Lemma 8, so we highlight the key differences and necessary adaptations without repeating the full derivation.

**Lemma 9.** *Lets assume we have a sub-block consisting of a self-attention with a residual connection and layer norm (i.e.,* $\mathbf{h}+\texttt{attn}(\texttt{LN}(\mathbf{h}))$*) with riicd. weights. We can show that, for any fixed countably infinite input set* $\mathcal{H}$*, this layer is injective almost surely:*

$$p_\Theta(\forall \mathbf{h}_1, \mathbf{h}_2 \in \mathcal{H}, t \in T, [\mathbf{h}_1]_t \neq [\mathbf{h}_2]_t : [\mathbf{h}_1 + \texttt{attn}(\texttt{LN}(\mathbf{h}_1))]_t \neq [\mathbf{h}_2 + \texttt{attn}(\texttt{LN}(\mathbf{h}_2))]_t) = 1 \tag{57}$$

*Proof.* We follow a proof strategy analogous to that of Lemma 8 in App. G.3. Following the same steps up to eq. (72), it suffices to show for this lemma that for any $\mathbf{h}_1, \mathbf{h}_2 \in \mathcal{H}$ and $t \in T$, where $[\mathbf{h}_1]_t \neq [\mathbf{h}_2]_t$, we have:

$$p_\Theta\Big([\mathbf{h}_1]_t + [\texttt{attn}(\texttt{LN}(\mathbf{h}_1))]_t \neq [\mathbf{h}_2]_t + [\texttt{attn}(\texttt{LN}(\mathbf{h}_2))]_t \mid [\texttt{attn}(\texttt{LN}(\mathbf{h}_1))]_t \neq [\texttt{attn}(\texttt{LN}(\mathbf{h}_2))]_t\Big) = 1 \tag{76}$$

We can write this according to the definition of an attention block (Submodule 1):

$$p_\Theta\Big([\mathbf{h}_1]_t + (\mathbf{W}^O)^T[\mathbf{h}'_1]_t \neq [\mathbf{h}_2]_t + (\mathbf{W}^O)^T[\mathbf{h}'_2]_t \mid [\texttt{attn}(\texttt{LN}(\mathbf{h}_1))]_t \neq [\texttt{attn}(\texttt{LN}(\mathbf{h}_2))]_t\Big) \tag{77}$$

where $\mathbf{h}'_1$ and $\mathbf{h}'_2$ are the hidden states after concatenation in the self-attention mechanism (see eq. (12)). The remainder of the proof follows the same approach as the proof of Lemma 8 in App. G.3, starting from eq. (73a). □

| | All Pairs | Same Output | Not Same Output | Same Variables | Not Same Variables |
|---|---|---|---|---|---|
| Input | $8.5e{-2}_{\pm 1.1e-2}$ | $8.5e{-2}_{\pm 1.1e-2}$ | $1.7e{-1}_{\pm 1.9e-2}$ | $8.5e{-2}_{\pm 1.1e-2}$ | $1.6e{-1}_{\pm 1.9e-2}$ |
| Layer 1 | $5.7e{-4}_{\pm 4.5e-4}$ | $5.7e{-4}_{\pm 4.5e-4}$ | $2.4e{-2}_{\pm 6.2e-3}$ | $5.7e{-4}_{\pm 4.5e-4}$ | $1.1e{-2}_{\pm 4.3e-3}$ |
| Layer 2 | $4.5e{-4}_{\pm 3.8e-4}$ | $4.5e{-4}_{\pm 3.8e-4}$ | $3.2e{-2}_{\pm 1.2e-2}$ | $4.5e{-4}_{\pm 3.8e-4}$ | $1.2e{-2}_{\pm 6.7e-3}$ |
| Layer 3 | $3.3e{-4}_{\pm 2.8e-4}$ | $3.3e{-4}_{\pm 2.8e-4}$ | $8.9e{-2}_{\pm 4.3e-2}$ | $3.3e{-4}_{\pm 2.8e-4}$ | $1.5e{-2}_{\pm 1.0e-2}$ |

Table 1: An approximation of the minimal Euclidean distance of the trained MLP model in the hierarchical equality task using 1,280,000 samples. The minimal Euclidean distance is computed between all samples to a randomly selected subset of 10,000 samples. We compute it over 10 different random seeds and present the mean and standard deviation (the number after $\pm$).

# H MLP Injectivity in Hierarchical Equality Task

We see in Fig. 2 that the IIA remains low for the `identity of first argument` algorithm on a fully trained model even when using a $\phi^{\texttt{nonlin}}$ alignment map (based on `revnet`). A reasonable assumption for why would be that the fully trained model does not fulfil some assumption required by our proof of Theorem 1 (any algorithm is an input-restricted distributed abstraction for any model) given in §4. In this section, we present follow-up experiments investigating the reason for this disagreement between our empirical results on the `identity of first argument` algorithm and the theoretical result of Theorem 1.

Let us first note that to prove Theorem 1 we rely on an existence proof: showing there exists a function $\phi$ which satisfies the conditions for a DNN to be abstracted by an algorithm. It says nothing, however, about this function being learnable in practice. Our experiments, however, measure IIA on an unseen test set—which requires $\phi^{\texttt{nonlin}}$ to not only fit a training set, but generalise to new data. Therefore, following our proof of Theorem 1 we explore the IIA on the train set. However, on the normal training set (with 1,280,000 samples), we still do not get an IIA over 0.55. On the other hand, if we repeat the experiment with only 1,000 training samples, we see $\phi^{\texttt{nonlin}}$ achieves an IIA of over 0.99 on the training set. Therefore, it is likely that the `revnet` used when defining $\phi^{\texttt{nonlin}}$ does not have enough capacity to fit the overly complex function our proof describes.

To further analyse why the capacity of the used `revnet` is not sufficient, we analyse the injectivity of the evaluated MLP by investigating its hidden representations. We first evaluate 1,280,000 randomly sampled inputs and their hidden states, checking if they are all unique. In these 1,280,000 samples (and repeating this experiment with 10 different random seeds), no collisions were found, implying the evaluated MLP is (at least close to) injective.

We now examine the supposition that the model finds it more difficult to distinguish between hidden states that share the same values for the variables in `both equality relations` than between those that do not. To this end, we compute the minimal Euclidean distance between hidden states across the entire set of 1,280,000 samples to a randomly selected subset of 10,000 samples. Specifically, we measure the minimal pairwise Euclidean distance among: (i) all sample pairs, (ii) sample pairs sharing the same output, (iii) sample pairs sharing the same values for both equality variables in `both equality relations`, (iv) sample pairs that do not share the same output and (v) sample pairs that do not share the same values for both equality variables. The results are presented in Table 1. We observe that the minimal Euclidean distances are smaller for pairs sharing the same output or the same equality-variable values compared to pairs that do not. This suggests that, although injectivity is preserved, a RevNet likely will find it more challenging to separate hidden states that share variable values.

# I Additional Experiment Details

In this section, we present additional details about our hierarchical equality task (in App. I.1) and indirect object identification (in App. I.2) experiments. We also present details and results on the distributive law task (in App. I.3).

## I.1 Hierarchical Equality Task

**Task 1 (from Geiger et al., 2024b).** *The **hierarchical equality task** is defined as follows. Let* $\mathbf{x} = \mathbf{x}_1 \circ \mathbf{x}_2 \circ \mathbf{x}_3 \circ \mathbf{x}_4$ *be a 16-dimensional vector, where each* $\mathbf{x}_i \in \mathbb{R}^4$ *for* $i \in \{1, 2, 3, 4\}$*, and* $\circ$ *denotes vector concatenation. The input space is* $\mathcal{X} = [-0.5, 0.5]^{16}$*, and the output space is*

$\mathcal{Y} = \{\texttt{false}, \texttt{true}\}$. *The task function is:*

$$\mathtt{T}(\mathbf{x}) = \big((\mathbf{x}_1 == \mathbf{x}_2) == (\mathbf{x}_3 == \mathbf{x}_4)\big), \tag{78}$$

*where the equality* $(\mathbf{x}_i == \mathbf{x}_j)$ *holds if and only if* $\mathbf{x}_i$ *and* $\mathbf{x}_j$ *are equal as vectors in* $\mathbb{R}^4$.

#### I.1.1 Algorithms

We define the following three candidate algorithms in detail.

**Alg 1.** *The* ***both equality relations alg.*** *to solve Task 1 has* $\boldsymbol{\eta}_{\texttt{inner}}=\{\eta_{\mathbf{x}_1==\mathbf{x}_2}, \eta_{\mathbf{x}_3==\mathbf{x}_4}\}$ *and:*

$$f_{\mathtt{A}}^{\eta_{\mathbf{x}_1==\mathbf{x}_2}}(\mathbf{v}_{\mathtt{par}_{\mathtt{A}}(\eta_{\mathbf{x}_1==\mathbf{x}_2})}) = (v_{\eta_{\mathbf{x}_1}} == v_{\eta_{\mathbf{x}_2}})$$
$$f_{\mathtt{A}}^{\eta_{\mathbf{x}_3==\mathbf{x}_4}}(\mathbf{v}_{\mathtt{par}_{\mathtt{A}}(\eta_{\mathbf{x}_3==\mathbf{x}_4})}) = (v_{\eta_{\mathbf{x}_3}} == v_{\eta_{\mathbf{x}_4}})$$
$$f_{\mathtt{A}}^{\eta_y}(\mathbf{v}_{\mathtt{par}_{\mathtt{A}}(\eta_y)}) = (v_{\eta_{\mathbf{x}_1==\mathbf{x}_2}} == v_{\eta_{\mathbf{x}_3==\mathbf{x}_4}})$$

**Alg 2.** *The* ***left equality relation alg.*** *to solve Task 1 has* $\boldsymbol{\eta}_{\texttt{inner}}=\{\eta_{\mathbf{x}_1==\mathbf{x}_2}\}$ *and:*

$$f_{\mathtt{A}}^{\eta_{\mathbf{x}_1==\mathbf{x}_2}}(\mathbf{v}_{\mathtt{par}_{\mathtt{A}}(\eta_{\mathbf{x}_1==\mathbf{x}_2})}) = (v_{\eta_{\mathbf{x}_1}} == v_{\eta_{\mathbf{x}_2}})$$
$$f_{\mathtt{A}}^{\eta_y}(\mathbf{v}_{\mathtt{par}_{\mathtt{A}}(\eta_y)}) = (v_{\eta_{\mathbf{x}_1==\mathbf{x}_2}} == (v_{\eta_{\mathbf{x}_3}} == v_{\eta_{\mathbf{x}_4}}))$$

**Alg 3.** *The* ***identity of first argument alg.*** *to solve Task 1 has* $\boldsymbol{\eta}_{\texttt{inner}}=\{\eta_{\mathbf{x}_1}\}$ *and:*

$$f_{\mathtt{A}}^{\eta_{\mathbf{x}_1}}(\mathtt{par}_{\mathtt{A}}(\eta_{\mathbf{x}_1})) = \mathbf{x}_1$$
$$f_{\mathtt{A}}^{\eta_y}(\mathtt{par}_{\mathtt{A}}(\eta_y)) = ((v_{\eta_{\mathbf{x}_1}} == \mathbf{x}_2) == (\mathbf{x}_3 == \mathbf{x}_4))$$

#### I.1.2 Training Details

For the hierarchical equality task, we use a 3-layer MLP with $|\boldsymbol{\psi}_1| = |\boldsymbol{\psi}_2| = |\boldsymbol{\psi}_3| = 16$. The model is trained using the Adam optimiser with learning rate 0.001 and cross-entropy loss. We use a batch size of 1024 and train on 1,048,576 samples, with 10,000 samples each for evaluation and testing. Training runs for a maximum of 20 epochs with early stopping after 3 epochs of no improvement.

For the training progression experiments, we use the same configuration but limit training to 2 epochs.

When training the alignment maps $\phi$, we use a batch size of 6400 and train for up to 50 epochs with early stopping after 5 epochs of no improvement (using a threshold of 0.001 for the required change, compared to 0 for MLP training). We use the Adam optimiser with learning rate 0.001 and cross-entropy loss. To generate the datasets for DAS, for Alg. 1 we intervene with a probability of 1/3 on $\eta_{\mathbf{x}_1==\mathbf{x}_2}$, 1/3 on $\eta_{\mathbf{x}_3==\mathbf{x}_4}$, and 1/3 on both variables. The samples for the base and source inputs are generated such that $(\mathbf{x}_1 == \mathbf{x}_2)$ and $(\mathbf{x}_3 == \mathbf{x}_4)$ each hold 50% of the time. For Alg. 2 and Alg. 3 we intervene on $\eta_{\mathbf{x}_1==\mathbf{x}_2}$ and $\eta_{\mathbf{x}_1}$ for all samples, respectively. For each algorithm, we sample 1,280,000 interventions for training, 10,000 for evaluation, and 10,000 for testing.

#### I.1.3 Additional Results

We present results for the three candidate algorithms for the hierarchical equality task, analysing the effect of hidden size $d_{\text{rn}}$ and intervention size $|\boldsymbol{\psi}_\eta^\phi|$ across all MLP layers.

For the `both equality relations` algorithm, Fig. 8a, 9a and 9b demonstrate that the hidden size experiment aligns with previously reported trends, while also showing how alignment maps $\phi$ of increasing complexity perform across training epochs, layers, and intervention sizes.

For the `left equality relation` algorithm, as shown in Fig. 8b, 9c and 9d, we observe similar patterns: increasing hidden size and intervention size improves performance, and alignment is generally more successful in later layers during early training.

For the `identity of first argument` algorithm, Fig. 8c, 9e and 9f reveal that, interestingly, some alignment is achieved—especially in layer 3—during the first half of training, but this effect diminishes in the second half.

Overall, these results demonstrate that the hidden size experiment is consistent with the findings reported in the main paper. They also show that it is easier, in untrained models, to find an alignment map for later layers, and that transient alignment can occur in specific layers and algorithms during the initial stages of training.

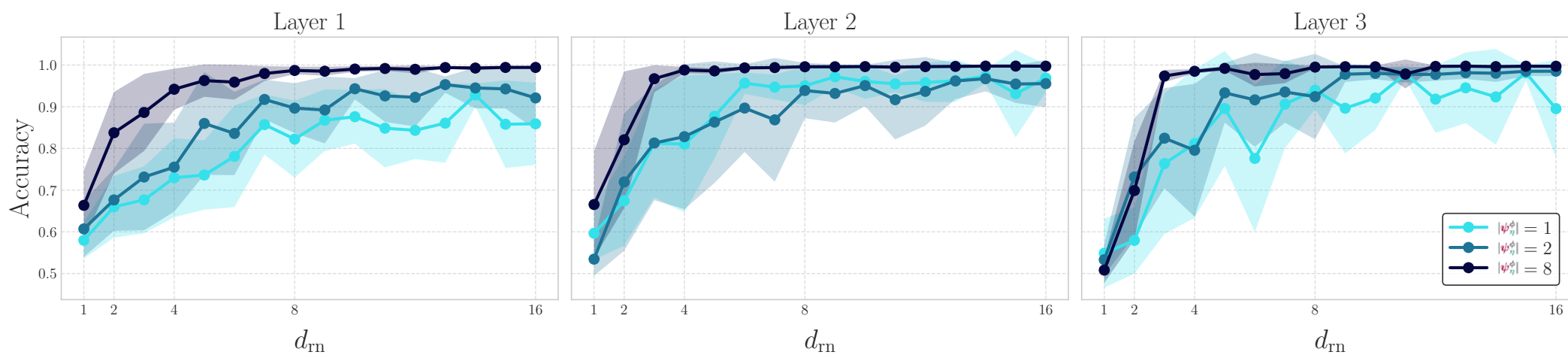

(a) `both equality relations`

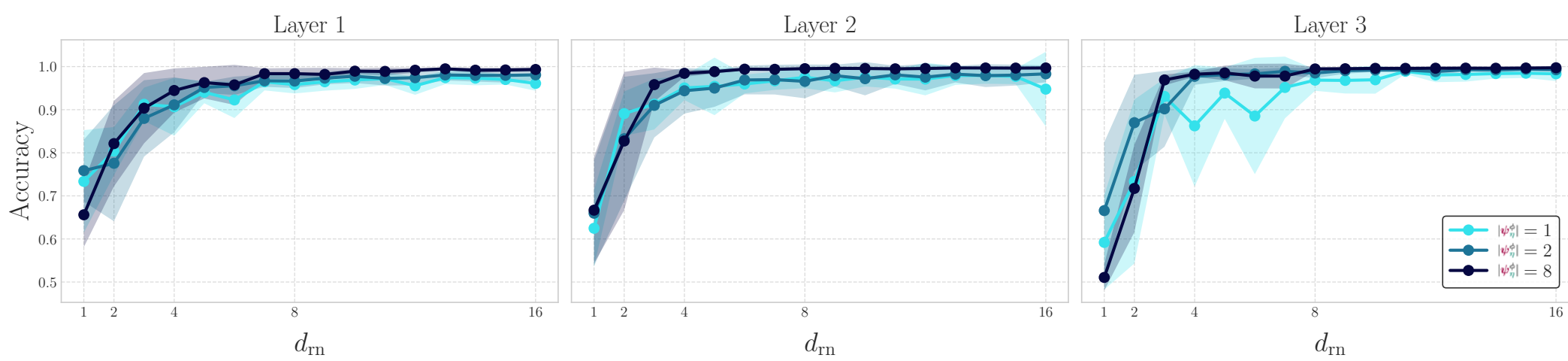

(b) `left equality relation`

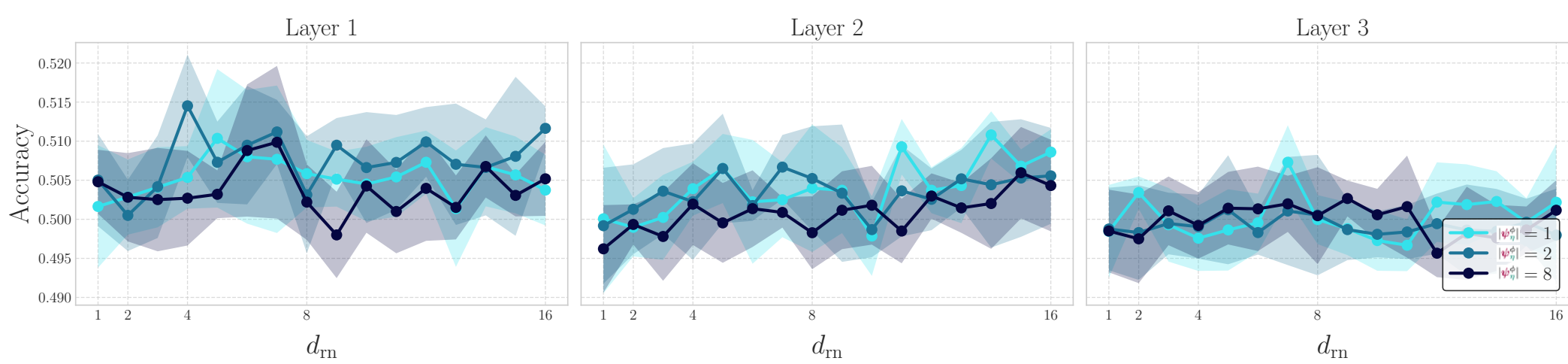

(c) `identity of first argument`

Figure 8: Mean IIA over 5 seeds using $\phi^{\texttt{nonlin}}$ ($L_{\text{rn}} = 1$) on the trained DNN. Performance improves with larger hidden dimension $d_{\text{rn}}$ and intervention size $|\psi^{\phi}_{\eta}|$. Each subplot corresponds to one of the three candidate algorithms for the hierarchical equality task, showing how the model's representational capacity influences performance.

### I.2 Indirect Object Identification Task

**Task 2.** *The **Indirect Object Identification** (IOI) task involves predicting the indirect object in sentences with a specific structure. Each input $\mathbf{x} \in \mathcal{X}$ consists of a text where a subject* (`S`) *and an indirect object* (`IO`) *are introduced, followed by the* `S` *giving something to the* `IO`*. For example:*

*"Friends Juana and Kristi found a mango at the bar. Kristi gave it to" $\Rightarrow$ "Juana"*

*Here, "Juana" and "Kristi" are introduced, with "Kristi"* (`S`) *appearing again before giving something to "Juana"* (`IO`)*. The output set $\mathcal{Y}$ consists of the first tokens of the two names:*

$$\mathcal{Y} = \{\texttt{first_token}(\texttt{S}), \texttt{first_token}(\texttt{IO})\} \tag{79}$$

#### I.2.1 Algorithm

For this task, we evaluate the **ABAB-ABBA** algorithm. Denoting the two names in the story as A and B, this algorithm determines whether the sentence follows an ABAB pattern (e.g., *"Friends Juana and Kristi found a mango at the bar. Juana gave it to Kristi"*) or an ABBA pattern (e.g., *"Friends Juana and Kristi found a mango at the bar. Kristi gave it to Juana"*). If the pattern is ABAB (where B is the indirect object `IO`), the algorithm predicts the first token of B. Conversely, for an ABBA pattern, it predicts the first token of A. In our experiments, we intervene on whether an input follows the ABAB pattern or not.

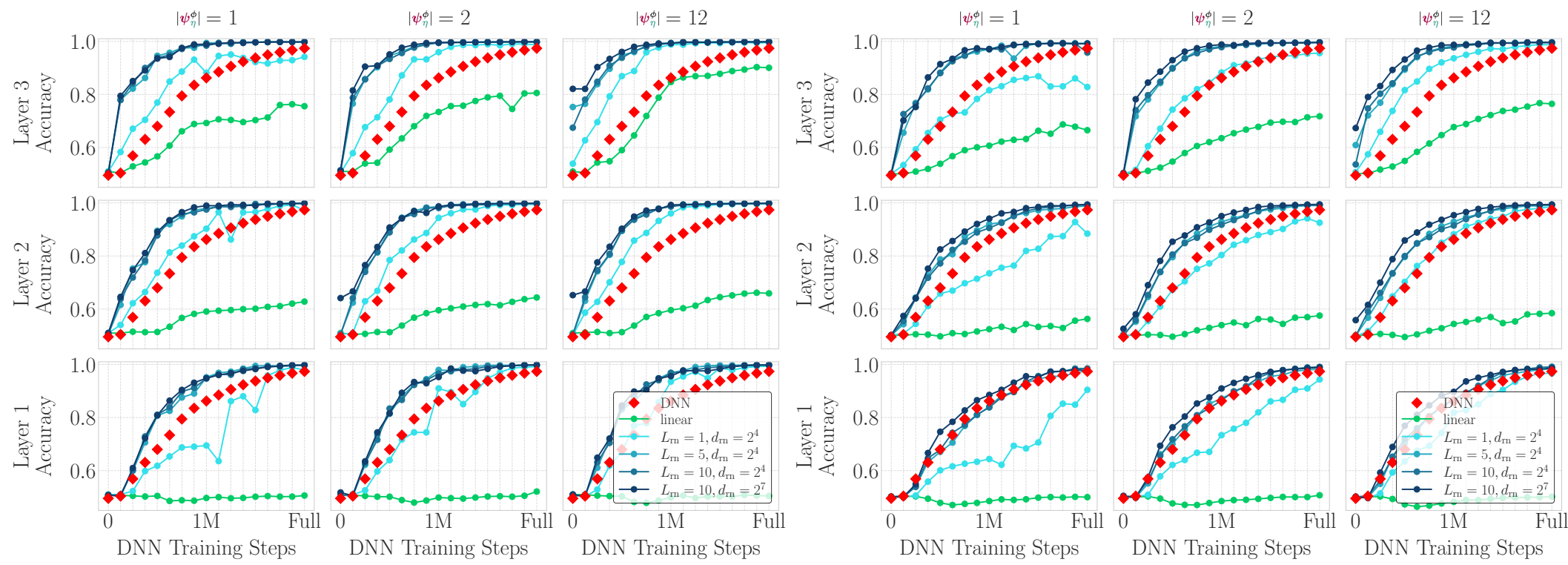

(a) Max IIA for `both equality relations` (b) Mean IIA for `both equality relations`

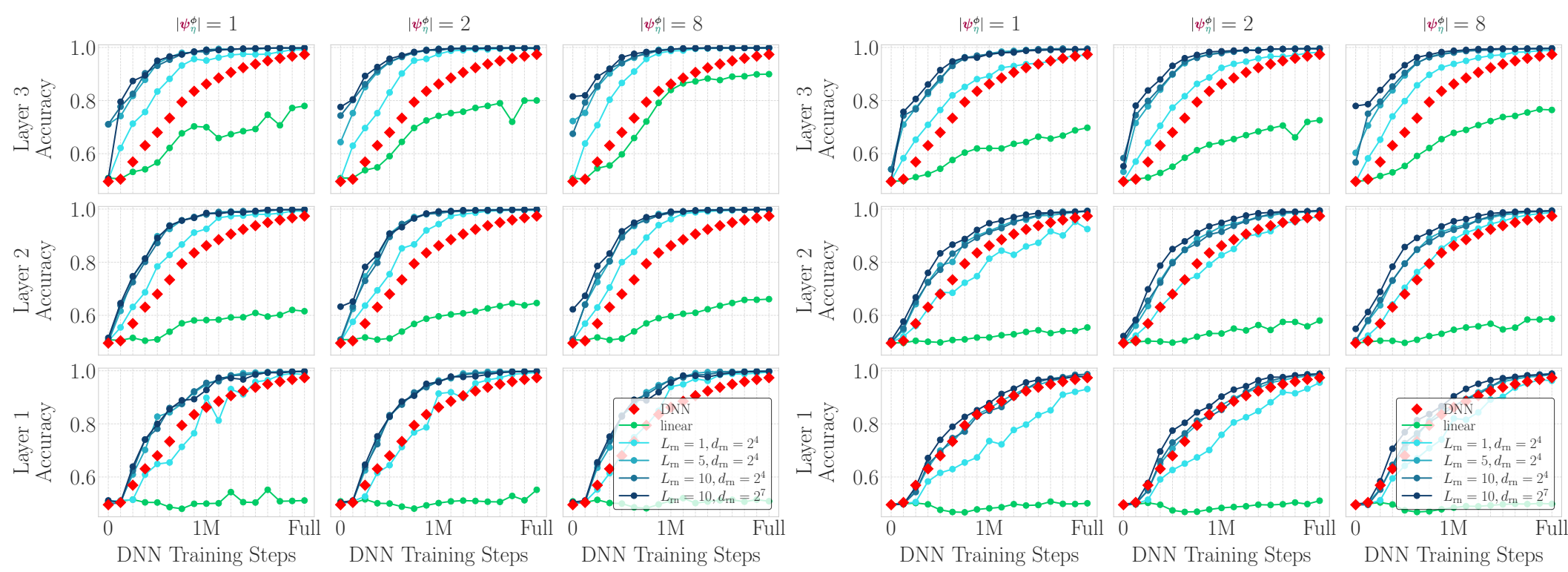

(c) Max IIA for `left equality relation` (d) Mean IIA for `left equality relation`

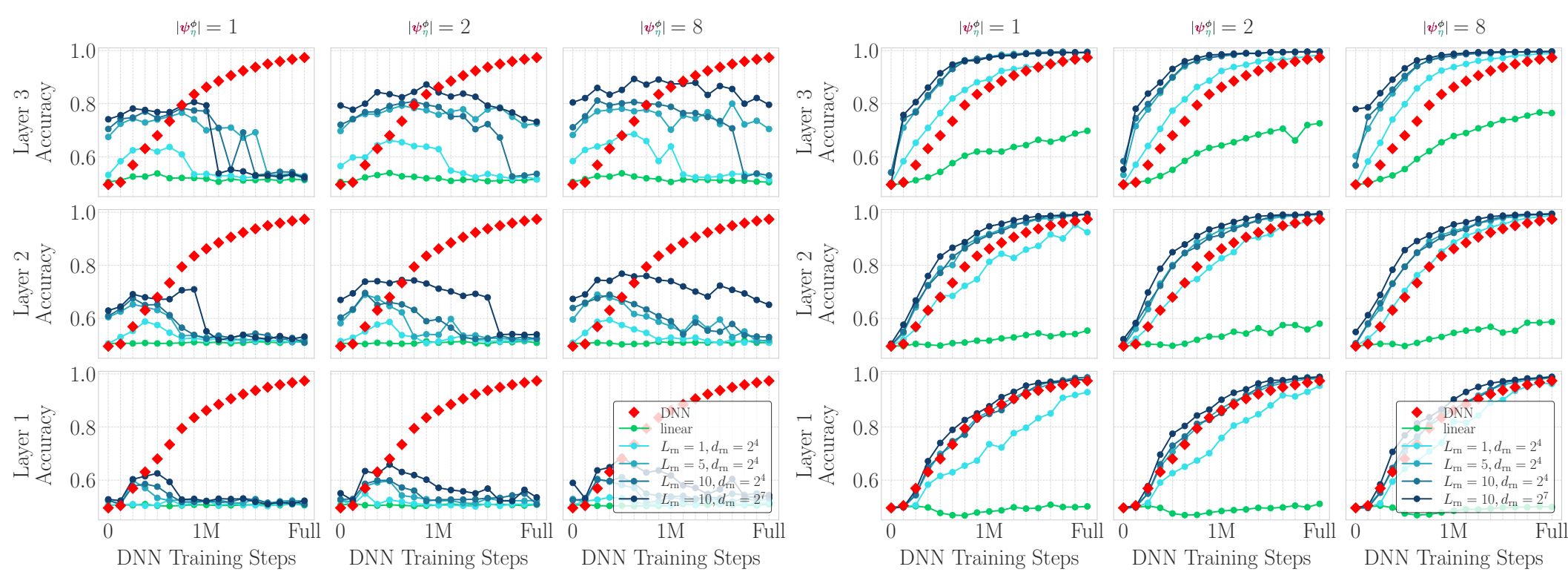

(e) Max IIA for `identity of first argument` (f) Mean IIA for `identity of first argument`

Figure 9: IIA over 5 seeds for each combination of MLP layer (rows) and intervention size (columns) during training progression for the tested algorithms.

**Alg 4.** *The* ABAB-ABBA *algorithm for the IOI task has one inner node* $\eta_{\texttt{inner}} = \{\eta_1\}$ *and is defined as follows:*

$$f_{\texttt{A}}^{\eta_1}(\texttt{par}_{\texttt{A}}(\eta_1)) = \texttt{check_is_abab_pattern}(v_{\eta_{\mathbf{x}}}) \tag{80a}$$

$$f_{\texttt{A}}^{\eta_y}(\texttt{par}_{\texttt{A}}(\eta_y)) = \begin{cases} \texttt{first_token}(\texttt{get_name_b}(v_{\eta_{\mathbf{x}}})) & \textit{if } v_{\eta_1} = \texttt{true} \\ \texttt{first_token}(\texttt{get_name_a}(v_{\eta_{\mathbf{x}}})) & \textit{if } v_{\eta_1} = \texttt{false} \end{cases} \tag{80b}$$

*Here,* $\texttt{get_name_a}(\mathbf{x})$ *extracts the first name (denoted A, e.g. Juana in our example) and* $\texttt{get_name_b}(\mathbf{x})$ *extracts the second name (denoted B, e.g. Kristi in our example) from the input sentence* $\mathbf{x}$*. The function* $\texttt{check_is_abab_pattern}(\mathbf{x})$ *returns* `true` *if the sentence follows an "ABAB" structure (e.g., "A and B ... A gave to B", meaning B is the* IO*) and* `false` *if it follows*

*an "ABBA" structure (e.g., "A and B ... B gave to A", meaning A is the* IO*).* first_token(*Name*) *returns the first token of the specified name. The output* **y** *is the first token of the indirect object.*

### I.2.2 Training Details

We use models from the Pythia suite (Biderman et al., 2023) to evaluate the IIA performance of the different $\phi$ on the IOI task. Specifically, we employ the Pythia 31M, 70M, 160M, and 410M parameter models. We also examine different training checkpoints provided by these models to analyse how IIA evolves during training. To assess robustness, we replicate a subset of experiments using alternative Pythia model seeds from (van der Wal et al., 2025).

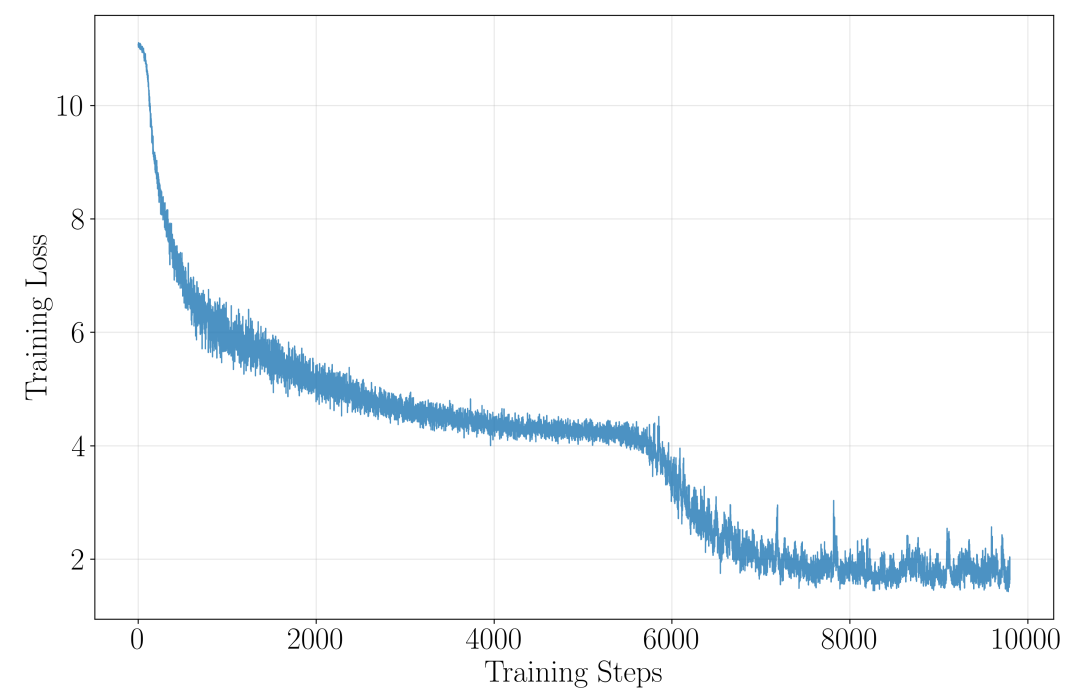

Figure 10: Cross Entropy loss ($y$-axis) during training ($x$-axis) of the alignment map for the randomly initialised *Pythia-31m*. The loss plateaus between 4k and 6k steps, and suddenly drops after 6k steps.

We train all alignment maps on 2 epochs of $10^6$ interventions based on data from Muhia (2022), with a batch size of 256 and a learning rate of $10^{-4}$. For all experiments, we set the intervention size $|\psi^{\phi}_{\eta}|$ to half of the DNN's hidden dimension. For smaller models (31M, 70M), we train using float64 precision and a learning rate of $10^{-3}$, as these adjustments proved crucial for convergence. We also note that we observed quite severe grokking behaviour, where models had low IIA for a long time, which quickly jumped to high IIA values at a certain point of training (see Fig. 10; wandb run).

### I.2.3 Additional Results

**Robustness across random seeds.** In Fig. 11, we examine how our main results from §6 generalise across multiple training seeds of the Pythia model. The key trends hold consistently across all 5 seeds - we can find perfect alignments using $\phi^{\texttt{nonlin}}$ in most cases. However, we observe two notable exceptions. For one seed, the DNN fails to learn the IOI task even after full training. For another seed, we cannot find an alignment using even complex alignment maps under our current setup.[17] All other seeds achieve perfect alignment under $\phi^{\texttt{nonlin}}$. We hypothesise that the alignment failure case is primarily due to suboptimal training of the alignment map. Due to computational constraints, we did not perform extensive hyperparameter tuning that might have achieved convergence.

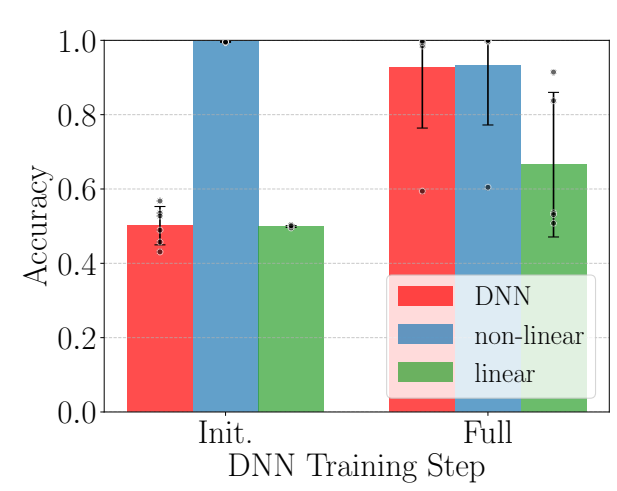

Figure 11: IIA of alignment between the ABAB-ABBA algorithm and the *Pythia-410m* model across multiple seeds (seeds 1 to 5 from van der Wal et al., 2025), with interventions at layer 12. We evaluate the IIA of both $\phi^{\texttt{lin}}$ and $\phi^{\texttt{nonlin}}$ (with $d_{\texttt{h}} = 64, K = 1$) on randomly initialised (Init.) and fully trained (Full) DNNs.

**Generalisation across distinct name sets.** In the main paper, we split the dataset from Muhia (2022) by ensuring that no two sentences appear in both the training and evaluation sets. However, this splitting strategy does not guarantee that the names themselves are distinct between training and evaluation sets. In Fig. 12, we examine the results when using completely different sets of names for training and evaluation. The results differ substantially: we cannot find an alignment using even complex alignment maps for the randomly initialised DNN. This suggests that IIA on the randomly initialised DNN may depend critically on overlap between the specific entities encountered during

[17]These failures occur in different seeds: seed 3 shows poor IIA despite learning the task, while seed 4 fails to learn the IOI task.

training and evaluation. For the fully trained DNN, we observe perfect alignment using $\phi^{\texttt{nonlin}}$ and reasonably high alignment using $\phi^{\texttt{lin}}$.

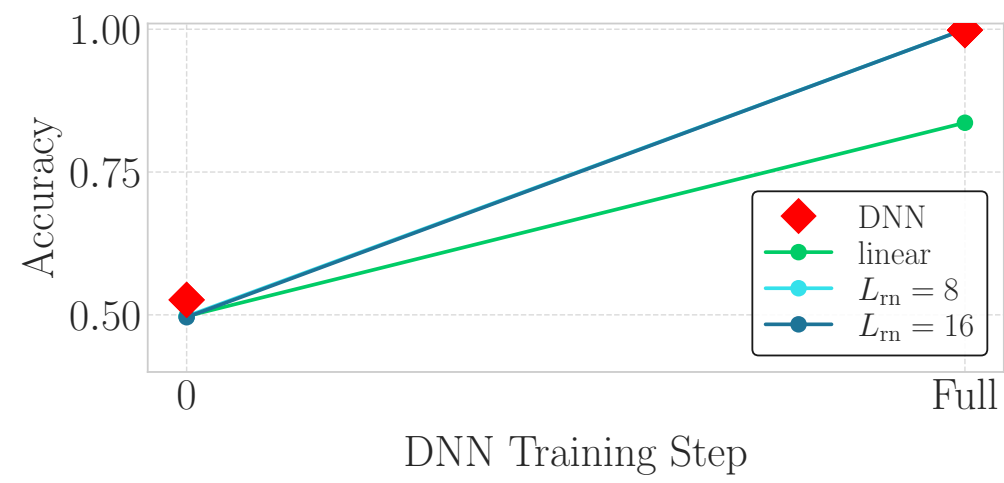

Figure 12: IIA of alignment between the ABAB-ABBA algorithm and the *Pythia-410m* model using a different set of names for training and evaluation, with interventions at layer 12. We evaluate the IIA of both $\phi^{\texttt{lin}}$ and $\phi^{\texttt{nonlin}}$ (with $d_{\texttt{h}} = 64, K = 1$) on randomly initialised and fully trained (Full) DNNs.

### I.3 Distributive Law Task

We now study a similar task to the hierarchical equality (in App. I.1), based on the distributive law of and ($\wedge$) and or ($\vee$).

**Task 3.** *The* ***distributive law task*** *is defined as follows. Let* $\mathbf{x} = \mathbf{x}_1 \circ \mathbf{x}_2 \circ \mathbf{x}_3 \circ \mathbf{x}_4 \circ \mathbf{x}_5 \circ \mathbf{x}_6$ *be a 24-dimensional vector, where each* $\mathbf{x}_i \in [-0.5, 0.5]^4$ *for* $i \in \{1, 2, 3, 4, 5, 6\}$*, and* $\circ$ *denotes vector concatenation. The input space is* $\mathcal{X} = [-0.5, 0.5]^{24}$*, and the output space is* $\mathcal{Y} = \{\texttt{false}, \texttt{true}\}$*. The task function is*

$$\texttt{T}(\mathbf{x}) = \big((\mathbf{x}_1 == \mathbf{x}_2) \wedge (\mathbf{x}_3 == \mathbf{x}_4)\big) \vee \big((\mathbf{x}_3 == \mathbf{x}_4) \wedge (\mathbf{x}_5 == \mathbf{x}_6)\big), \tag{81}$$

*where the equality* $(\mathbf{x}_i == \mathbf{x}_j)$ *holds if and only if* $\mathbf{x}_i$ *and* $\mathbf{x}_j$ *are equal as vectors in* $\mathbb{R}^4$.

#### I.3.1 Algorithms

We define the following two candidate algorithms.

**Alg 5.** *The* **`And-Or-And alg.`** *to solve Task 3 has*

$$\eta_{\texttt{inner}} = \{\eta_{(\mathbf{x}_1 == \mathbf{x}_2)\wedge(\mathbf{x}_3 == \mathbf{x}_4)}, \eta_{(\mathbf{x}_3 == \mathbf{x}_4)\wedge(\mathbf{x}_5 == \mathbf{x}_6)}\}$$

*and it is defined as follows:*

$$f_{\texttt{A}}^{\eta_{(\mathbf{x}_1 == \mathbf{x}_2)\wedge(\mathbf{x}_3 == \mathbf{x}_4)}}(\mathbf{v}_{\texttt{par}_{\texttt{A}}(\eta_{(\mathbf{x}_1 == \mathbf{x}_2)\wedge(\mathbf{x}_3 == \mathbf{x}_4)})}) = (v_{\eta_{\mathbf{x}_1}} == v_{\eta_{\mathbf{x}_2}}) \wedge (v_{\eta_{\mathbf{x}_3}} == v_{\eta_{\mathbf{x}_4}})$$
$$f_{\texttt{A}}^{\eta_{(\mathbf{x}_3 == \mathbf{x}_4)\wedge(\mathbf{x}_5 == \mathbf{x}_6)}}(\mathbf{v}_{\texttt{par}_{\texttt{A}}(\eta_{(\mathbf{x}_3 == \mathbf{x}_4)\wedge(\mathbf{x}_5 == \mathbf{x}_6)})}) = (v_{\eta_{\mathbf{x}_3}} == v_{\eta_{\mathbf{x}_4}}) \wedge (v_{\eta_{\mathbf{x}_5}} == v_{\eta_{\mathbf{x}_6}})$$
$$f_{\texttt{A}}^{\eta_y}(\mathbf{v}_{\texttt{par}_{\texttt{A}}(\eta_y)}) = v_{\eta_{(\mathbf{x}_1 == \mathbf{x}_2)\wedge(\mathbf{x}_3 == \mathbf{x}_4)}} \vee v_{\eta_{(\mathbf{x}_3 == \mathbf{x}_4)\wedge(\mathbf{x}_5 == \mathbf{x}_6)}}$$

**Alg 6.** *The* **`And-Or alg.`** *to solve Task 3 has*

$$\eta_{\texttt{inner}} = \{\eta_{\mathbf{x}_3 == \mathbf{x}_4}, \eta_{(\mathbf{x}_1 == \mathbf{x}_2)\vee(\mathbf{x}_5 == \mathbf{x}_6)}\}$$

*and it is defined as follows:*

$$f_{\texttt{A}}^{\eta_{\mathbf{x}_3 == \mathbf{x}_4}}(\mathbf{v}_{\texttt{par}_{\texttt{A}}(\eta_{\mathbf{x}_3 == \mathbf{x}_4})}) = (v_{\eta_{\mathbf{x}_3}} == v_{\eta_{\mathbf{x}_4}})$$
$$f_{\texttt{A}}^{\eta_{(\mathbf{x}_1 == \mathbf{x}_2)\vee(\mathbf{x}_5 == \mathbf{x}_6)}}(\mathbf{v}_{\texttt{par}_{\texttt{A}}(\eta_{(\mathbf{x}_1 == \mathbf{x}_2)\vee(\mathbf{x}_5 == \mathbf{x}_6)})}) = (v_{\eta_{\mathbf{x}_1}} == v_{\eta_{\mathbf{x}_2}}) \vee (v_{\eta_{\mathbf{x}_5}} == v_{\eta_{\mathbf{x}_6}})$$
$$f_{\texttt{A}}^{\eta_y}(\mathbf{v}_{\texttt{par}_{\texttt{A}}(\eta_y)}) = v_{\eta_{\mathbf{x}_3 == \mathbf{x}_4}} \wedge v_{\eta_{(\mathbf{x}_1 == \mathbf{x}_2)\vee(\mathbf{x}_5 == \mathbf{x}_6)}}$$

#### I.3.2 Training Details

For the distributive law task, we use a 3-layer MLP (see App. E.1) with an input dimensionality of 24, hidden layers of dimensionality $|\psi_1| = |\psi_2| = |\psi_3| = 24$, and an output dimensionality of 2. The model is trained using the Adam optimiser with a learning rate of 0.001 and cross-entropy loss. We use a batch size of 1024. The datasets are generated by randomly sampling input vectors $\mathbf{x} = \mathbf{x}_1 \circ \cdots \circ \mathbf{x}_6$ such that the target label $\bar{y} = \texttt{T}(\mathbf{x})$ is true 50% of the time. We sample 1,048,576 samples for training, 10,000 for evaluation, and 10,000 for testing. Training runs for a maximum of 20 epochs with early stopping after 3 epochs of no improvement.

For training $\phi$, we use a batch size of 6400 and train for up to 50 epochs with early stopping after 5 epochs of no improvement (using a threshold of 0.001 for the required change). We use the Adam

optimiser with learning rate 0.001 and cross-entropy loss. To generate the intervened datasets: For Alg. 5, we intervene with a probability of 1/3 on $\eta_{(\mathbf{x}_1==\mathbf{x}_2)\wedge(\mathbf{x}_3==\mathbf{x}_4)}$, 1/3 on $\eta_{(\mathbf{x}_3==\mathbf{x}_4)\wedge(\mathbf{x}_5==\mathbf{x}_6)}$, and 1/3 on both variables. For Alg. 6, we intervene with a probability of 1/3 on $\eta_{\mathbf{x}_3==\mathbf{x}_4}$, 1/3 on $\eta_{(\mathbf{x}_1==\mathbf{x}_2)\vee(\mathbf{x}_5==\mathbf{x}_6)}$, and 1/3 on both variables. For both algorithms, the samples for the base and source inputs are generated such that the output of the intervention changes compared to the base input 50% of the time. We sample 1,280,000 interventions for training, 10,000 for evaluation , and 10,000 for testing for each algorithm.

### I.3.3 Results

In this section, we discuss the results on the distributed law task using the And-or-And and And-Or algorithms. Our findings corroborate the results presented in the main paper. As shown in Fig. 13, using linear and identity alignment maps reveals distinct dynamics. The `And-Or` algorithm achieves higher IIA using $\phi^{\texttt{lin}}$, particularly in later layers where the IIA of $\phi^{\texttt{lin}}$ on the `And-Or-And` algorithm approaches 0.5. However, these dynamics completely vanish when using a more complex alignment map like $\phi^{\texttt{nonlin}}$, where we achieve almost perfect IIA everywhere.

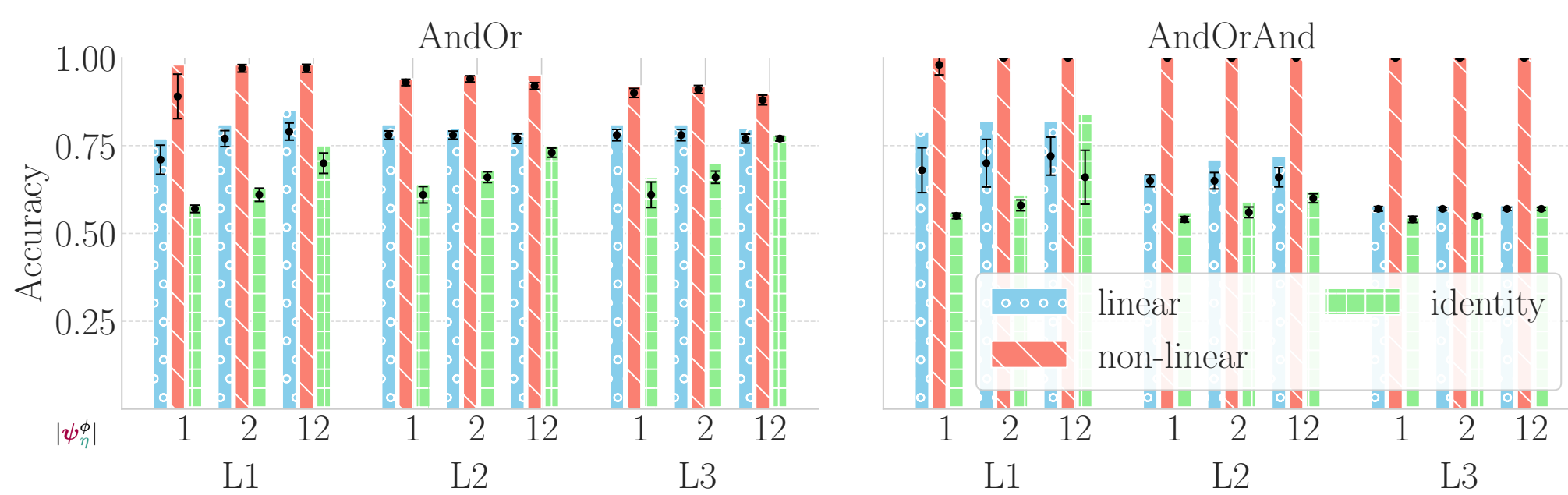

Figure 13: IIA in the distributive law task for causal abstractions trained with different alignment maps $\phi$. The figure shows results for both analysed algorithms for this task. The bars represent the max IIA across 10 runs with different random seeds. The black lines represent mean IIA with 95% confidence intervals. The $|\psi_\eta^\phi|$ denotes the intervention size per node. Without interventions, all DNNs reach 100% accuracy. The used $\phi^{\texttt{nonlin}}$ uses $L_{\text{rn}} = 10$ and $d_{\text{rn}} = 24$.

Fig. 14a and 14c present the evaluated IIA throughout model training. These training progression plots show that randomly initialised models often achieve IIA above 0.8 with non-linear alignment maps, supporting our insight that when the notion of causal abstraction is equipped with $\phi^{\texttt{nonlin}}$ it may identify algorithms which are not necessarily implemented by the underlying model. In Fig. 14b and 14d, we plot the mean IIA over 5 seeds instead of the maximum IIA.

The hidden size experiments (Fig. 15a and 15b) show that even RevNets with small $d_{\texttt{h}}$ of 4 achieve near-perfect IIA for And-Or-And, while And-Or never reaches perfect IIA in the second layer, regardless of the $d_{\texttt{h}}$. The training progression plots suggest a possible explanation: IIA for And-Or-And initially increases in the last two layers but then decreases, while RevNets maintain near-perfect IIA. This may indicate that And-Or-And is first implemented with simple encodings detectable by linear $\phi$s, before evolving into non-linear encodings that only RevNets can detect. The fact that And-Or never achieves high IIA in later layers further suggests it may not be a true abstraction of the model's behaviour, though we note this remains a hypothesis requiring further investigation.

**And-Or-And Training.** In this section, we analyse a DNN when this model is trained specifically to rely on the `And-Or-And` algorithm (and, consequently, to encode the values of its hidden nodes). We do so with the method from Geiger et al. (2022), training the DNN to encode `And-Or-And`'s hidden nodes' values in its second layer, with an intervention size of 12. This method is similar to how we train $\phi$ (see App. I.3.2), but $\phi$ is fixed to the identity function, and the DNN itself is trained; further, the training dataset is composed of 1/4 non-intervened samples, 1/4 samples with interventions on $\eta_{(\mathbf{x}_1==\mathbf{x}_2)\wedge(\mathbf{x}_3==\mathbf{x}_4)}$, 1/4 on $\eta_{(\mathbf{x}_3==\mathbf{x}_4)\wedge(\mathbf{x}_5==\mathbf{x}_6)}$, and 1/4 on both variables. We then evaluate if this DNN abstracts both the `And-Or-And` and `And-Or` using different $\phi$ (as before, after freezing the DNN). The IIA performance of these $\phi$ is presented in Fig. 16. We can see here that, when using identity and linear alignment maps $\phi$, IIA scores suggest that the `And-Or-And` algorithm seems to be implemented perfectly given the second layer, where we have only around 0.75 IIA for

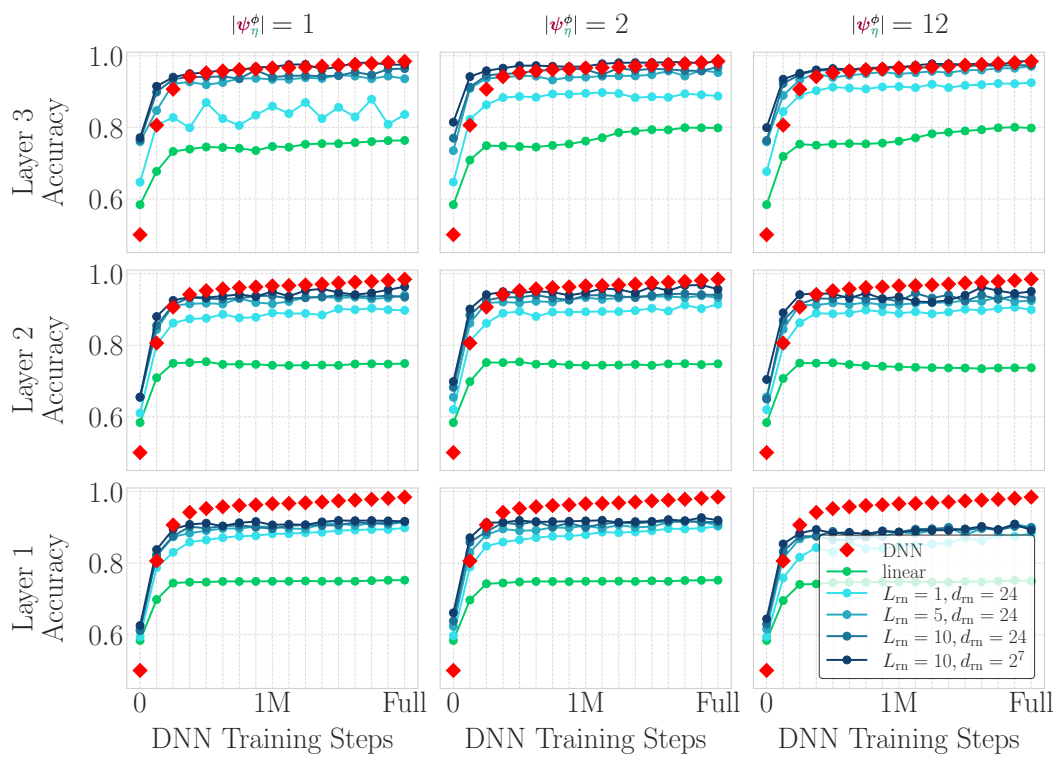

(a) Max IIA for `And-Or`

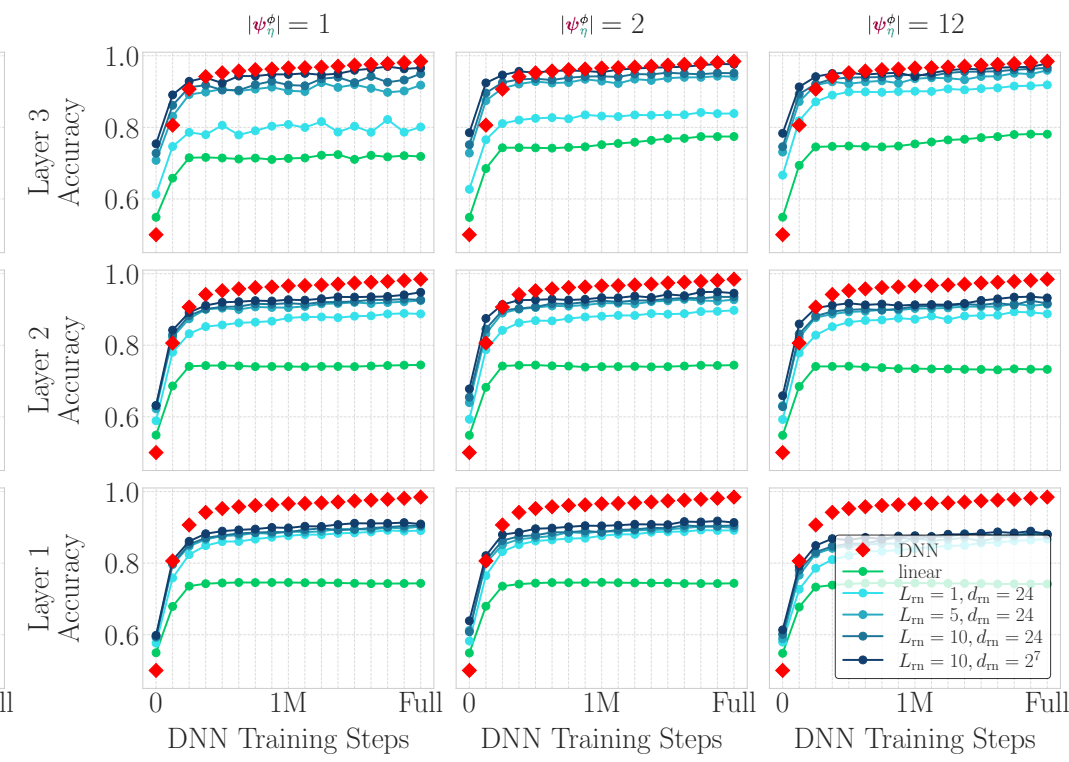

(b) Max IIA for `And-Or`

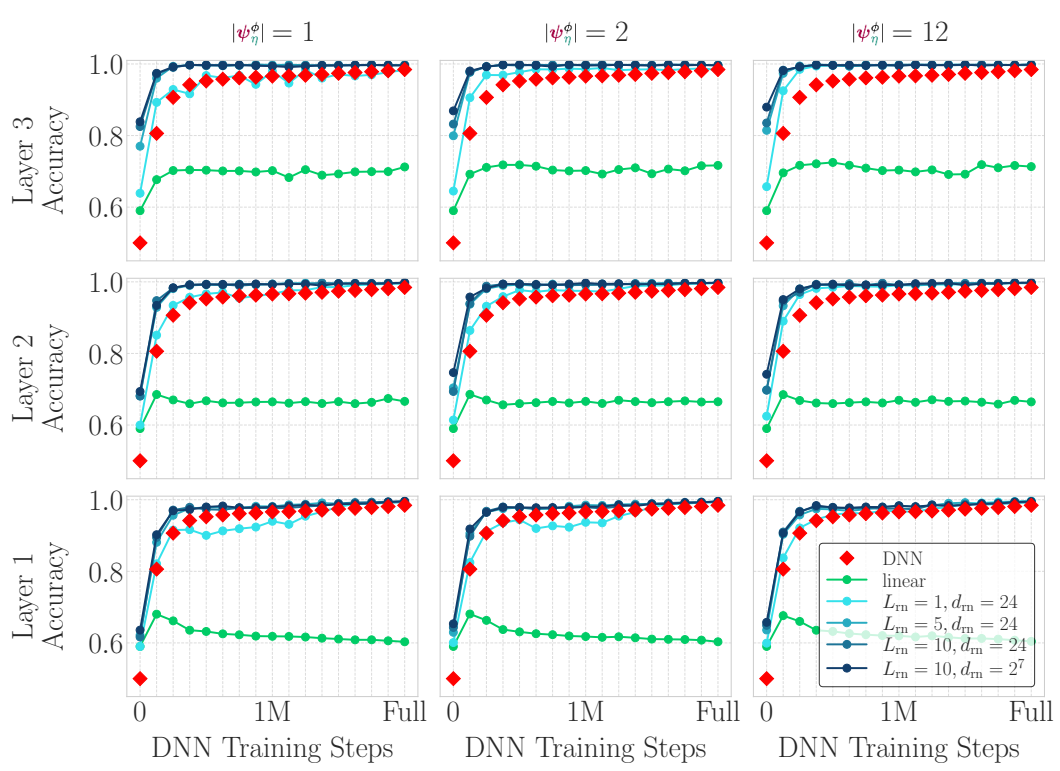

(c) Max IIA for `And-Or-And`

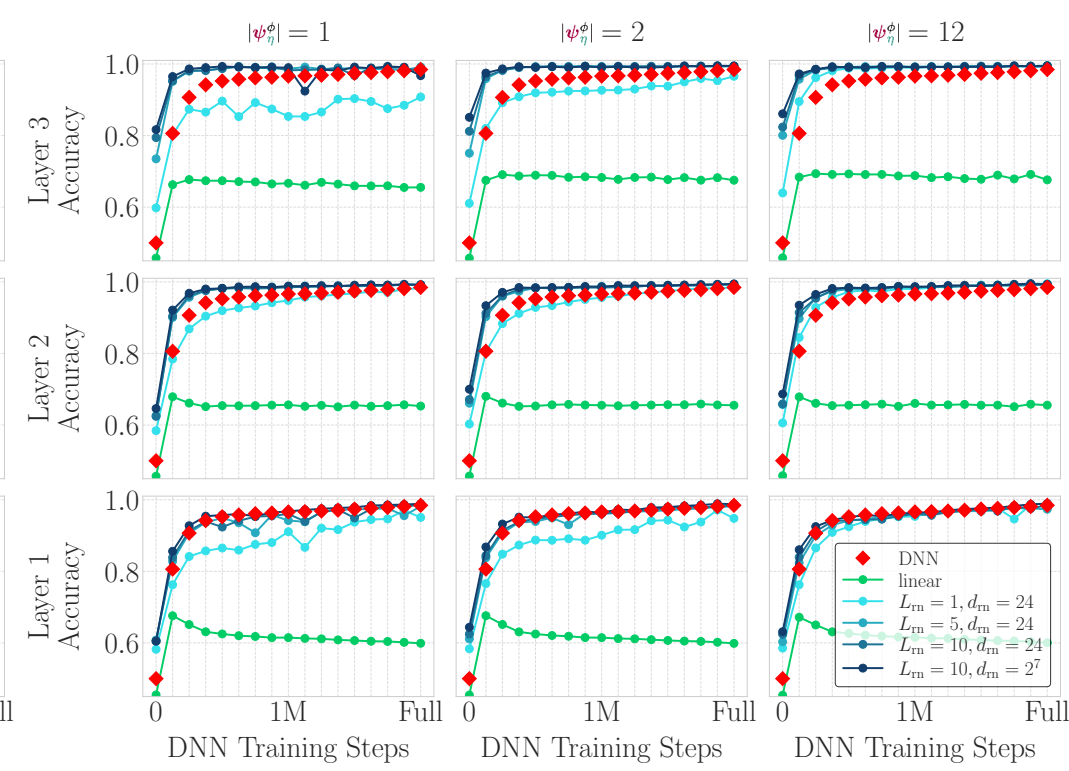

(d) Max IIA for `And-Or-And`

Figure 14: IIA over 5 seeds for each combination of MLP layer (rows) and intervention size (columns) during training progression for the evaluated algorithms.

the `And-Or` algorithm. However, these differences vanish almost completely using $\phi^{\texttt{nonlin}}$ as our alignment map.

**And-Or Training.** In this section, we report an experiment similar to the above, but we train our DNN to rely on the `And-Or` algorithm instead. These results are shown in Fig. 17. In this figure, we again see that, when using identity and linear as alignment map $\phi$, IIA performance suggests that the `And-Or` algorithm seems to be implemented perfectly given the second layer, where we have only around 0.65 IIA for the `And-Or-And` algorithm. These differences however vanish when using $\phi^{\texttt{nonlin}}$ as alignment map, which leads to perfect IIA scores with either algorithm.

# J Computational Resources

The experiments on MLP were executed on CPU (10 computers with i7-4770 or newer) over 3 weeks, as we noticed that DAS on small MLPs are faster on CPU than on GPU. The experiments on the Pythia models were executed on a single A100 GPU with 80GB of memory using approximately 30 GPU hours, including the hyperparameter tuning.

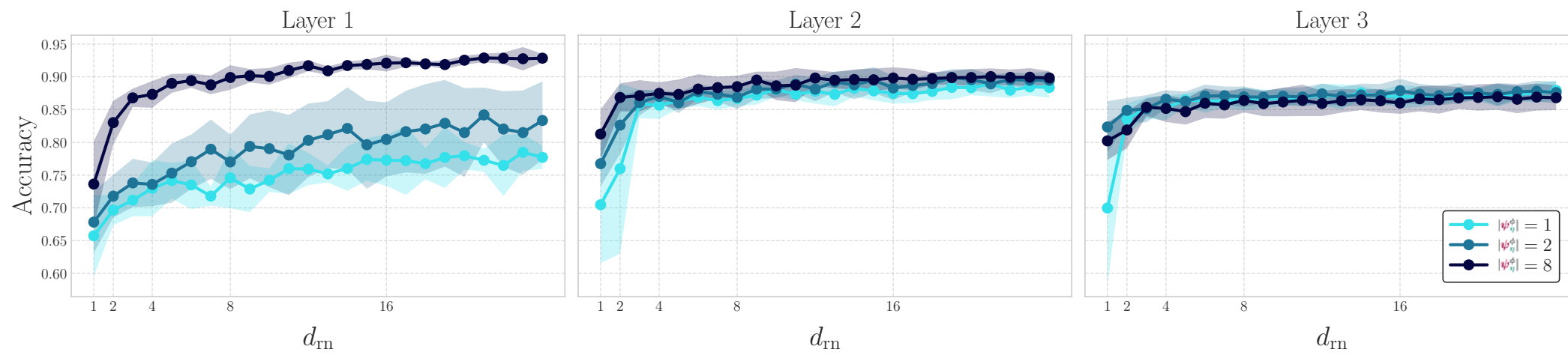

(a) `And-Or`

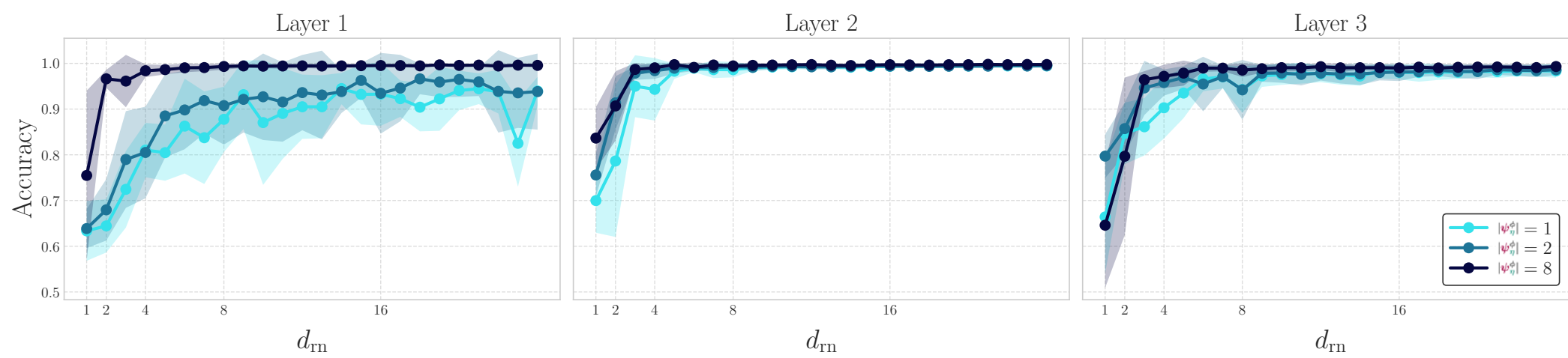

(b) `And-Or-And`

Figure 15: Mean IIA over 5 seeds using $\phi^{\texttt{nonlin}}$ ($L_{\text{rn}} = 1$) on the trained DNN. Performance improves with larger hidden dimension $d_{\text{rn}}$ and intervention size $|\psi_\eta^\phi|$. Each subplot corresponds to one of the two candidate algorithms for the distributed law task, showing how $\phi$'s representational capacity influences performance.

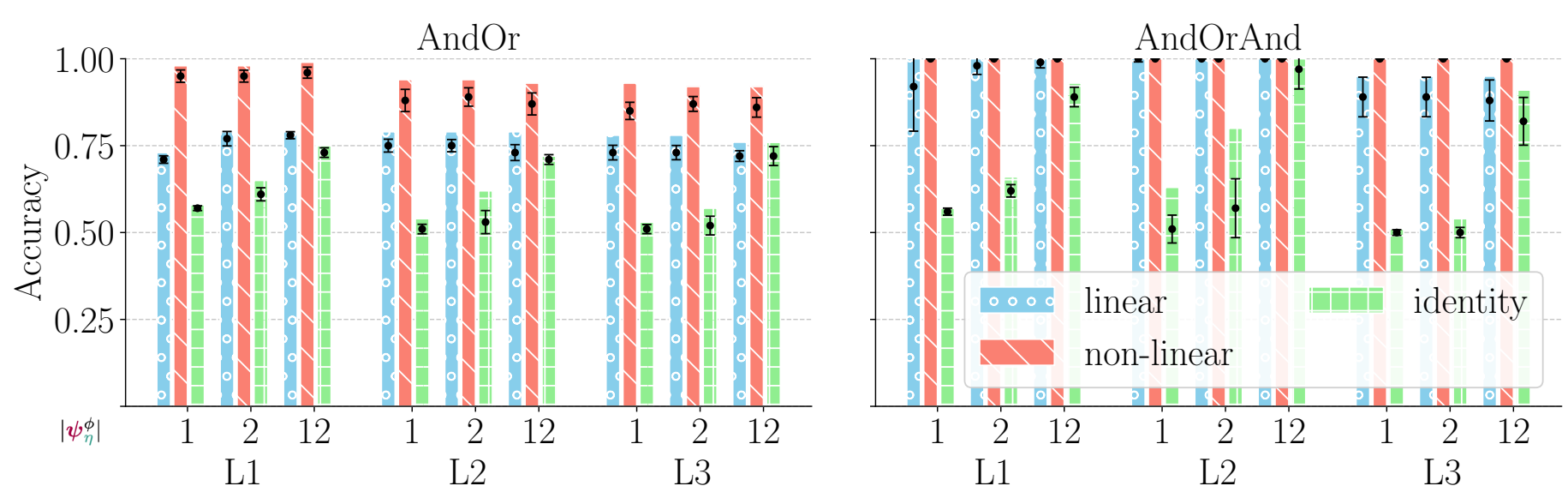

Figure 16: IIA in the distributive law task for causal abstractions trained with different alignment maps $\phi$ and a DNN trained to use the `And-Or-And` algorithm. The figure shows results when evaluating if the DNN encodes either of the analysed algorithms for this task. The bars represent the max IIA across 10 runs with different random seeds. The black lines represent mean IIA with 95% confidence intervals. The $|\psi_\eta^\phi|$ denotes the intervention size per node. All DNNs reach >99.9% accuracy after training. The used $\phi^{\texttt{nonlin}}$ uses $L_{\text{rn}} = 10$ and $d_{\text{rn}} = 16$.

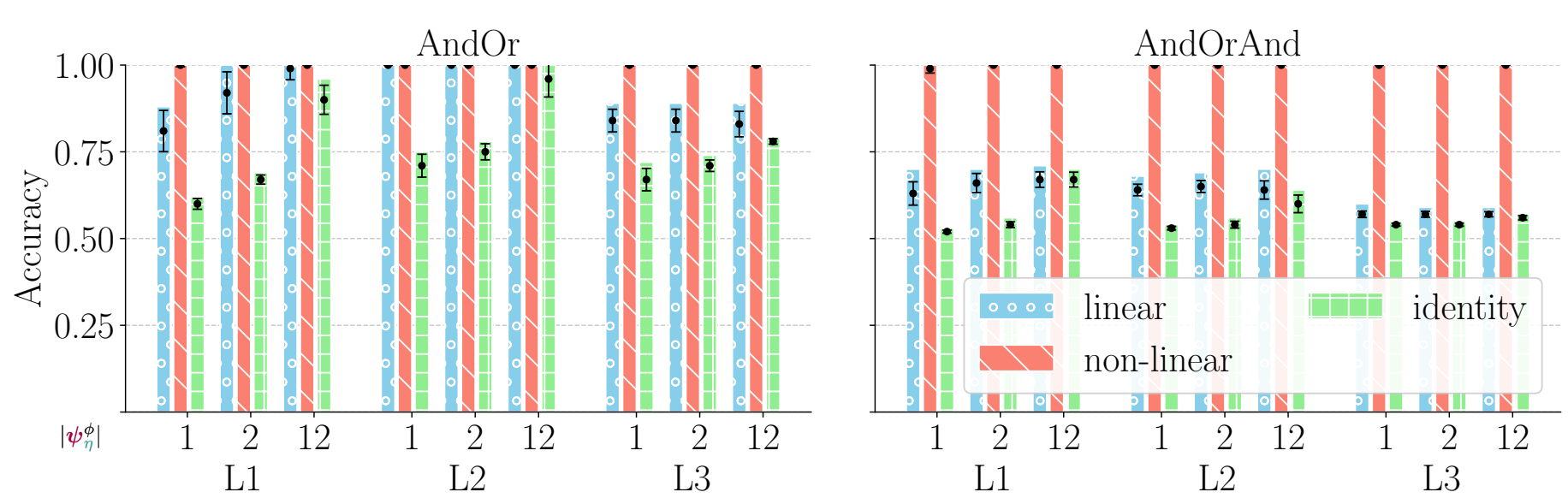

Figure 17: IIA in the distributive law task for causal abstractions trained with different alignment maps $\phi$ and a DNN trained to use the `And-Or` algorithm. The figure shows IIA results when evaluating if the DNN encodes either of the analysed algorithms for this task. The bars represent the max IIA across 10 runs with different random seeds. The black lines represent mean IIA with 95% confidence intervals. The $|\psi_\eta^\phi|$ denotes the intervention size per node. Without interventions, all DNNs reach >99.9% accuracy. The used $\phi^{\texttt{nonlin}}$ uses $L_{\text{rn}} = 10$ and $d_{\text{rn}} = 16$.

