# OpenReview forum: "The Non-Linear Representation Dilemma: Is Causal Abstraction Enough for Mechanistic Interpretability?"
_NeurIPS.cc/2025/Conference — NeurIPS 2025 spotlight_

### Official Review · Reviewer_vv44 · 2025-06-26

**Clarity:** 3
**Significance:** 3
**Originality:** 3
**Rating:** 5
**Confidence:** 3

**Summary:**

This paper critically examines the framework of causal abstraction for mechanistic interpretability. The authors prove that, under reasonable assumptions, allowing for arbitrarily powerful (non-linear) alignment maps renders the notion of causal abstraction vacuous, as any neural network can be perfectly mapped to any algorithm. This theoretical claim is supported by strong empirical evidence on both synthetic tasks and language models, where near-perfect interchange-intervention accuracy (IIA) is achieved even for randomly initialized models that cannot solve the task. The paper frames this finding as the "non-linear representation dilemma," highlighting the fundamental trade-off between the expressive power of an alignment map and the meaningfulness of the resulting explanation.

**Questions:**

In your conclusion, you state an "intuition" to support the linear representation hypothesis (LRH) but acknowledge that you "cannot make this intuition formal." This difficulty may arise because your work seems to make the general LRH intractable from both a falsification and verification standpoint:
* To falsify the universal LRH, one must demonstrate a feature that is provably non-linear. This requires showing both (i) the failure of linear maps and (ii) the success of a non-linear one. However, (i) might never be finished, and for (ii) your theorem shows that a successful alignment with a powerful nonlinear map is a method-induced artifact, providing no evidence that the network actually represents the feature non-linearly. Thus, a required counterexample can never be convincingly established.
* Simultaneously, any verification by a linear alignment map is weakened. It can always be challenged as a mere approximation of a more complex, unfalsifiable non-linear ground truth with perfect IIA.

Given that the universal LRH is rendered practically unprovable and unfalsifiable within this framework, shouldn't the paper's main takeaway be a stronger recommendation to abandon it as a useful scientific hypothesis? Instead of relying on intuition, shouldn't the community be guided to formulate and test a portfolio of more modest, but strictly falsifiable, hypotheses? For example:
* (a) a local, circuit-specific claim (e.g., "The IOI circuit in GPT-2 Small uses a linearly decodable representation for the 'ABAB pattern' variable").
* (b) An instrumental, Pareto-optimal claim (e.g., "Within a fixed complexity budget, a linear decoder provides the best explanation on the Pareto front of IIA vs. complexity").

**Ethical Concerns:**

["NO or VERY MINOR ethics concerns only"]

**Final Justification:**

The authors basically addressed all my questions. I think the paper is solid and well motivated. Therefore I keep the score.

**Limitations:**

yes

**Quality:**

4

**Strengths And Weaknesses:**

Strengths:
1. The paper tackles a foundational and timely problem in interpretability research. The central theorem, proving the vacuity of unconstrained causal abstraction, is a rigorous and significant contribution.
2. The empirical results provide compelling support for the theoretical claims. The demonstration that complex alignment maps can find perfect abstractions even in randomly initialized or incapable models is a strong piece of evidence.

Weaknesses:
1. The paper's conclusion feels somewhat tentative and does not fully embrace the strong implications of its own findings. The authors state an "intuition" to support the Linear Representation Hypothesis (LRH) but claim they cannot formalize it. This seems to miss an opportunity to propose more nuanced, falsifiable versions of the LRH as a path forward, as I will detail in the questions.

---

> ### Author Rebuttal · Authors · 2025-07-30
>
> Thank you very much for your thoughtful and constructive review. We greatly appreciate your detailed feedback and suggestions. Please find our response below.
>
>
> > Summary of Feedback:
>
> > The paper's conclusion feels somewhat tentative and does not fully embrace the strong implications of its own findings. The authors state an "intuition" to support the Linear Representation Hypothesis (LRH) but claim they cannot formalize it. This seems to miss an opportunity to propose more nuanced, falsifiable versions of the LRH as a path forward, as I will detail in the questions.
>
> > Given that the universal LRH is rendered practically unprovable and unfalsifiable within this framework, shouldn't the paper's main takeaway be a stronger recommendation to abandon it as a useful scientific hypothesis? Instead of relying on intuition, shouldn't the community be guided to formulate and test a portfolio of more modest, but strictly falsifiable, hypotheses?
>
>
> We would like to clarify that—while we believe some of our results suggest the linear representation hypothesis to produce more intuitively reliable interpretability insights—we do not fully endorse this linear hypothesis as a strong or sufficient hypothesis for causal abstraction. Further, we agree that more modest and falsifiable hypotheses can be more scientifically productive.
> As we see it, researchers would ideally—as the reviewer suggests—formulate and test a portfolio of modest, but strictly falsifiable, hypotheses (relying on local or instrumental claims). However, by themselves these modest hypotheses do not directly answer the core question: *What has the model actually implemented?* Based on a large-ish battery of such tests, it may then be reasonable for researchers to speculate about grander claims (such as the linear representation hypothesis). Many great interpretability papers already follow this format, and we see no harm in this—as long as there is a clear separation between factual and speculative claims.
>
>
>
> > (ii) your theorem shows that a successful alignment with a powerful nonlinear map is a method-induced artifact.
>
> Regarding this statement,
> it is true that successful alignment maps (under arbitrarily complex mappings) are guaranteed by our theorem, and thus they are indeed a method-induced artifact.
> However, this interpretation is not always entirely accurate. Our theorem demonstrates that there **exists** an **arbitrarily complex** alignment map such that any algorithm can be shown to be an input-restricted distributed abstraction of any model (given some assumptions). However, our theorem:
>
> * offers **no generalisation guarantees** to unseen data.
> * **does not explain** the empirical success of nonlinear alignment maps that **are not complex enough to overfit** a specific task.
>
> In our experiments, we use train–test splits, evaluating IIA on held-out data.
> Thus, when we observe high IIA using such non-linear alignment maps, we cannot simply dismiss them as artifacts. Therefore, we believe focusing on generalisation to be a promising step for future research. We expanded on this in the discussion section of the revised paper with the following paragraph:
>
>  **The Role of Generalisation.** We now highlight that Theorem 1 provides an existence proof for a perfect abstraction map (thus guaranteeing perfect IIA) between a DNN and an algorithm. This existence proof, however, leverages complex interactions between the intervened hidden states and the DNN’s structure, requiring perfect information about both and thus representing a form of extreme overfitting. Crucially, this theorem offers no guarantees regarding the learnability of the alignment map ϕ from limited data or its generalisation to unseen inputs. This gap between theoretical existence and practical learnability becomes evident in practice. For instance, in an additional experiment on the IOI task [added to the appendix in the revised version], we show that when training and test sets contain disjoint sets of names, the learned alignment map fails to generalise, resulting in low IIA on the test set. This suggests that generalisation should play a crucial role in causal abstraction analysis, as the ability to learn abstraction maps that transfer beyond training data seems fundamental to interpreting a model, distinguishing a genuine understanding of its inner workings from mere training pattern memorisation.
>
> We also believe that generalisation is necessary but not sufficient for faithful abstraction. Therefore, we are actively investigating now (as future work)  complex alignment maps that achieve high IIA, and whether they actually uncover meaningful mechanisms implemented by the DNNs, or whether they “hack” the model to achieve high performance.
> To continue this research, we are studying the carry one variable from the Mechanistic Interpretability Benchmark (MIB). This ongoing preliminary work illustrates the complexity of defining faithful alignment maps. So far, we've observed that a linear (but non-orthogonal) alignment map achieves IIA above 0.96 on this task, but does not appear to reflect any true mechanism implemented by the model. These preliminary findings raise more questions than they resolve—highlighting the early stage of this line of research. As such, we are hesitant to offer definitive recommendations just yet, but we aim to do so in future work.
>
>
>
>
>
>
>
>
>
> To conclude, the main message of this paper is therefore: alignment maps must be selected with care, and causal abstraction—though a powerful framework—should not be treated as a silver bullet.  While we offer some practical insights based on our findings, this work represents (to the best of our knowledge) the first systematic evaluation of the nonlinear representation dilemma in causal abstraction. We intentionally avoid making broad or strongly prescriptive claims at this stage. Instead, we see this paper as laying the groundwork for more practical, grounded guidelines, which we aim to develop in future research as our understanding of alignment mechanisms evolves.
>
> ---
>
> Please feel free to let us know if we have misunderstood any part of your feedback or if any clarification is needed. We would be happy to continue the discussion or elaborate further on any point.

---

> > ### Comment · Reviewer_vv44 · 2025-08-04
> >
> > I thank the authors for the response. My questions are basically addressed, and I will keep the score.

---

### Official Review · Reviewer_9y8j · 2025-07-02

**Clarity:** 3
**Significance:** 3
**Originality:** 3
**Rating:** 5
**Confidence:** 4

**Summary:**

This paper examines a fundamental limitation in causal abstraction methods for mechanistic interpretability. The authors prove that when alignment maps between neural networks and algorithms are allowed to be arbitrarily complex (non-linear), any neural network can be mapped to any algorithm under reasonable assumptions, making causal abstraction vacuous. They demonstrate this through Theorem 1 and empirical validation showing that complex alignment maps achieve perfect interchange intervention accuracy (IIA) even on randomly initialized models incapable of solving target tasks. The work identifies the "non-linear representation dilemma" - the inherent trade-off between alignment map complexity and interpretability validity - and argues that causal abstraction implicitly relies on strong linearity assumptions about neural information encoding.

**Questions:**

1. Can you provide a more systematic analysis of when the key assumptions (particularly input-injectivity and output-surjectivity) hold in practice? What are the concrete implications when these assumptions fail, and how common are such failures across different architectures and tasks?

2. Given that you've identified the non-linear representation dilemma, what specific recommendations would you provide to practitioners currently using causal abstraction methods? How to choose appropriate alignment map complexity? Relatedly, what are the most promising research directions for addressing the identified limitations?

3. Do your findings extend to other interpretability methods beyond causal abstraction? Are there fundamental insights here that apply more broadly to the interpretability research program?

**Ethical Concerns:**

["NO or VERY MINOR ethics concerns only"]

**Limitations:**

yes

**Quality:**

3

**Strengths And Weaknesses:**

### Strengths

technical
- theoretical contribution is mathematically rigorous with well-stated assumptions that are reasonable for practical neural networks (countable input spaces, input-injectivity, output-surjectivity)
- proof structure is sound and the empirical validation effectively demonstrates the theoretical predictions across multiple tasks
- experimental design systematically explores alignment map complexity using identity, linear, and non-linear (RevNet) parameterizations, providing evidence that complex maps can achieve high IIA regardless of model capability.

presentation
- makes inherently complex mathematical concepts accessible to the NeurIPS audience
- progressive introduction of concepts from basic causal models to distributed abstractions is well-structured
-  hierarchical equality and IOI tasks provide concrete illustrations of abstract theoretical points
- figures effectively communicate the key findings

novelty and significance
- focus on alignment map complexity in causal abstraction is novel
- non-linear representation dilemma provides a framework that could redirect research toward principled constraints for alignment maps.

### Weaknesses
- key assumptions (particularly input-injectivity) may fail in practice (as the authors admit) - "identity of first argument" experimental results also show this → somewhat limits the theoretical result
- empirical validation limited to relatively simple tasks and specific model architectures
- for Pythia, single-seed reduces statistical confidence

- math in Section 3 is quite dense and could benefit from more intuitive explanations
- notation becomes complex and inconsistent in places (particularly the intervention notation in equations 4-6)
- practical implications for practitioners could be articulated more clearly (what is the concrete guidance?)

significance and novelty
- limited to causal abstraction methods
- complexity-accuracy trade-off, while formalized in a novel context, echoes similar debates in the probing literature (but actually the authors were fairly thorough in citing the complexity-accuracy trade-off literature from probing)

minor:
 - throughout the paper, in figures and text, I'd advise to use different colors/patterns that are colorblind-friendly - you are using red-green contrast, in general the color scheme doesn't seem coherent to me (but well done on the different shadings in figure 1 to distinguish the categories without color). I would probably swap the red-green for blue-orange (given that you use blue in the plots). Overall this is minor though as in the figures you made sure to distinguish the elements without color.
 - figure captions could better explain what readers should conclude from each result
 - l. 99: The notation "Δ|Y|-1" is unclear - clarify this is the probability simplex
 - l. 182: Ensure the bijection notation is consistent with later usage
 - Add more intuitive explanations alongside formal definitions (especially Definitions 1-2, 6-7)
 - maybe include a glossary of key terms in the appendix,
 - maybe include a notation table in the appendix listing all symbols
 - proof sketches in the main text for Theorem 1, even if full proof is in appendix
 - consistent subscript/superscript conventions throughout (some inconsistencies in ω vs ε notation)
 - In the introduction, better motivate why this problem matters for the broader ML community

---

> ### Author Rebuttal · Authors · 2025-07-30
>
> Thank you for your review and your helpful suggestions. We address your points below.
>
> > key assumptions (particularly input-injectivity) may fail in practice (as the authors admit) - "identity of first argument" experimental results also show this → somewhat limits the theoretical result
>
> We agree that the assumptions in Theorem 1 generally do not hold exactly. However, we aimed to convey that input-injectivity and output-surjectivity often hold approximately in practice, which is sufficient for many algorithms. To clarify this point, we revised the following parts of Section 4:
> * Input-injectivity: We changed “This assumption is also present in prior work (e.g., Pimentel et al., 2020b) and seems to be well approximated in practice (Morris et al., 2023). Further, we show in App. F that this is almost surely true for transformers at initialisation—assuming real-valued weights and activations (and thus ignoring floating point precision).” to “This assumption is also present in prior work (e.g., Pimentel et al., 2020b) and we show in App. F—assuming real-valued weights and activations—that this is almost surely true for transformers at initialisation. Due to floating point precision and neural collapse (Papyan et al., 2020), it is likely to not hold fully in practice; however, it still seems to be well-approximated in many empirical settings (e.g., see Morris et al. (2023) and App. H).” App. H (newly added) contains new experiments on input-injectivity for the hierarchical equality task.
> * Output-surjectivity: We changed “In practice, however, even with large vocabulary sizes, all outputs seem to still be producible by language models (Grivas et al., 2022).” to “[...] even with large vocabulary sizes, it seems that most outputs can still be produced by language models (Grivas et al., 2022) which is enough for these DNNs to be abstracted by many algorithms.”
>
> As noted, we had hypothesised that imperfect alignment from "identity of first argument" might stem from the MLP not being injective or surjective at initialisation due to the ReLU activation. However, further analysis (newly introduced App. H) ruled out input-injectivity as the main issue: across 1.28M inputs and 10 seeds, we found no hidden state collisions, suggesting that input-injectivity holds in practice for this setting.
> This raises the question: why didn’t we achieve perfect IIA, particularly for the “identity of first argument” intervention (Fig. 1, right)? Since the model successfully performs the task, output-surjectivity appears satisfied. Moreover, high IIA in the Both Equality Relations and Left Equality Relations settings suggests other assumptions are also met. We believe two factors explain the lower IIA:
>
> 1. **Generalization Gap.**
> Theorem 1 is an existence proof, does not discuss the learnability of this function and can be seen as a form of “strong overfitting”. In our experiments—where training and test sets are separate—the generalization gap becomes apparent. For instance, with only 1,000 training examples, the RevNet could overfit and reach >99% training IIA.
>
> 2. **Model Capacity Limitations.**
> Theorem 1 assumes arbitrarily expressive models, but our RevNet has limited capacity. It fails to fit large “identity of first argument” datasets (e.g., 1.28M samples, max training IIA ≈ 0.55), yet overfits smaller ones (training IIA of >99% on 1,000 examples).
>
> > empirical validation limited to relatively simple tasks and specific model architectures
>
> We agree that broader empirical validation would strengthen the work. However, in line with prior work, we focused on a diverse but tractable set of tasks to enable in-depth analysis. While most tasks are simple, we also include results on a real-world language modelling task—Indirect Object Identification (IOI)—using Pythia models. Despite its simplicity, IOI offers meaningful insights into the non-linear representation dilemma in LLMs.
>
> > for Pythia, single-seed reduces statistical confidence
>
> We agree that this is an important point. To clarify: we do provide multi-seed results in **Appendix G.2.3** for a subset of the experiments. We made this more explicit in the main text in the revised version. Due to computational constraints, we couldn’t rerun all Pythia experiments for all seeds.
>
> > practical implications for practitioners could be articulated more clearly (what is the concrete guidance?)
>
> This is an important point. While our paper focuses on theoretical foundations, we agree it is valuable to begin identifying early takeaways for practice. To that end, we have added the following paragraph to the discussion section:
>
> **The Role of Generalisation.** We now highlight that Theorem 1 provides an existence proof for a perfect abstraction map (thus guaranteeing perfect IIA) between a DNN and an algorithm. This existence proof, however, leverages complex interactions between the intervened hidden states and the DNN’s structure, requiring perfect information about both and thus representing a form of extreme overfitting. Crucially, this theorem offers no guarantees regarding the learnability of the alignment map ϕ from limited data or its generalisation to unseen inputs. This gap between theoretical existence and practical learnability becomes evident in practice. For instance, in an additional experiment on the IOI task [added to the appendix in the revised version], we show that when training and test sets contain disjoint sets of names, the learned alignment map fails to generalise, resulting in low IIA on the test set. This suggests that generalisation should play a crucial role in causal abstraction analysis, as the ability to learn abstraction maps that transfer beyond training data seems fundamental to interpreting a model, distinguishing a genuine understanding of its inner workings from mere training pattern memorisation.
>
> We are further continuing to research this topic. In particular, we are evaluating the Carry One variable from the Mechanistic Interpretability Benchmark (MIB) to better understand variable encoding. So far, we have observed that even a linear (but non-orthogonal) alignment map achieves >0.96 IIA on this task, while appearing to *not* reflect any true mechanism used by the DNN.
> As this research is still preliminary, we refrain from making definitive recommendations and aim to offer clearer insights in future work.
>
> > throughout the paper, in figures and text, I'd advise to use different colors/patterns that are colorblind-friendly - you are using red-green contrast
>
> Thank you for the suggestion. We tested the plots with a colorblindness simulator and ensured the red is sufficiently dark relative to the green (in grayscale). That said, we're happy to revise the colors further if the reviewer has more insights.
>
> > math in Section 3 is quite dense and could benefit from more intuitive explanations
> > Add more intuitive explanations alongside formal definitions (especially Definitions 1-2, 6-7)
>
> We agree, and to address this we have added two new figures to our paper:
> * A visual illustration of the non-linear representation dilemma, visualising how causal abstraction works and demonstrating how increasing the complexity of the alignment map ϕ can improve interchange-intervention accuracy (IIA). This figure is in the main body of the paper.
> * A schematic overview of the causal abstraction definitions introduced in Section 3. This figure highlights the subset–superset relationships between different definitions, illustrating how constraints on the allowed abstraction map and interventions shape these relationships. This figure is in the appendix.
>
>
> > Can you provide a more systematic analysis of when the key assumptions (particularly input-injectivity and output-surjectivity) hold in practice? What are the concrete implications when these assumptions fail, and how common are such failures across different architectures and tasks?
>
> We touched on this in our reply to the first point with regard to the tasks used in this paper. We agree that it would be valuable for future work to systematically analyse when these assumptions hold. In practice, however, these assumptions seemed to be well-approximated in practice in our experiments, and we did not find either of them to affect our analyses.
>
> > What specific recommendations would you provide to practitioners currently using causal abstraction methods? How should alignment map complexity be chosen? What are the most promising directions for future research?
>
> We touched on this in our reply to the first point.
>
> > Do your findings extend to other interpretability methods beyond causal abstraction? Are there broader insights for the interpretability research community?
>
> Yes, we believe the non-linear representation dilemma is relevant to a range of interpretability methods. For instance, the **complexity–accuracy trade-off**—central to our dilemma—has also been observed in probing (e.g., Hewitt and Liang, 2019). This trade-off arises in any interpretability method that aims to identify features within hidden representations:
> * **Loosely constrained** methods may yield higher accuracy but risk capturing *spurious* correlations.
> * **Tightly constrained** methods may promote simplicity but fail to recover *important non-linear* structure.
>
> ---
>
> We also worked on clarifying the paper in response to the suggestions that were not explicitly addressed above. We hope this response addresses your questions and concerns. Please let us know if there are any points that remain unclear—we would be happy to elaborate further.
>
> ### References
>
> Pimentel et al. (2020b). Information-theoretic probing for linguistic structure
>
> Morris et al. (2023). Text embeddings reveal (almost) as much as text
>
> Papyan et al. (2020). Prevalence of Neural Collapse during the terminal phase of deep learning training
>
> Grivas et al. (2022). Low-rank softmax can have unargmaxable classes in theory but rarely in practice

---

> ### Author Response · Authors · 2025-08-05
>
> Dear Reviewer,
>
> Thank you again for your review and let us know if there are any remaining questions or concerns we can address.
>
> Best,
> The authors

---

### Official Review · Reviewer_XbMd · 2025-07-02

**Clarity:** 4
**Significance:** 3
**Originality:** 2
**Rating:** 5
**Confidence:** 4

**Summary:**

This paper studies the application of causal abstractions for mechanistic interpretability, where the idea is to explain black box ML models (like neural networks) with high-level algorithms that serve as causal hypotheses of the model’s behavior. The hypotheses are tested by learning an abstraction mapping between the ML model and the algorithm such that the properties of a causal abstraction hold. The main result of the paper is a theorem that proves that, without constraints on the abstraction map such as linearity, an abstraction map can be constructed between any ML model and any high-level hypothesis. The ensuing discussion emphasizes the implication that in the general case, this approach of explaining ML models can be vacuous. Experiments demonstrate that with nonlinear abstraction maps, interchange intervention accuracy remains consistently high.

**Questions:**

7. To what extent does Thm. 1 imply the problem is caused by the linearity assumption of the alignment rather than other factors? For example, perhaps IIA is an imperfect metric of measuring alignment since IIA does not consider semantic meaning of the algorithm nodes.

8. How does the topic of alignment relate to disentangled causal representation learning? As in, is the idea of learning the alignment map similar to the idea of learning a representation map that disentangles the inputs into latent causal factors?

**Ethical Concerns:**

["NO or VERY MINOR ethics concerns only"]

**Final Justification:**

I will maintain my score and recommend the paper for acceptance. The paper addresses an important problem and is well-written, providing a promising avenue of future research. The rebuttal has further reinforced this, and the promised changes will improve the paper even further.

**Limitations:**

Limitations are discussed.

**Paper Formatting Concerns:**

No issues.

**Quality:**

4

**Strengths And Weaknesses:**

**Strengths:**

1. Thm. 1 is an impactful result. Indeed, with growing literature in the space of causal abstractions for mechanistic interpretability, the non-linear representation dilemma really strikes at the heart of the issue of what assumptions are meaningful for determining causal hypotheses and what they imply in the behavior of neural networks.
2. Assumptions are clearly stated and justified in Sec. 4.
3. The paper is very well-written, with all notation clearly defined and well-labeled. The proof of Thm. 1 is carefully organized. The paper also neatly organizes all of the important background information, including the general process of using causal abstraction for mechanistic interpretability, the discussion of how information is encoded in deep neural networks, and some new definitions surrounding distributed abstractions that grounds prior work in the context of this paper.
4. The experimental results show some interesting analyses. It is notable that in linear alignments, IIA decreases in deeper layers, a problem that goes away with nonlinear alignments. The fact that IIA increases with simpler alignments following more training is also interesting.

**Weaknesses:**

5. The paper could benefit from having figures that describe the causal abstraction for mechanistic interpretability procedure and highlight the impact of Thm. 1.
6. The paper does not offer many new novel contributions and in some ways serves as a review or position paper. All content until Sec. 4 is background information (although Sec. 3.2 offers new definitions that portray existing work in new perspectives). The main contributions of the paper are Thm. 1 (with proof in the appendix) and the experimental results. The contributions are impactful, but they are mainly adding to discussion points surrounding the efficacy of this mechanistic interpretability approach rather than offering new solutions.

Overall the paper is quite strong, and I recommend it for acceptance. I do not believe that the weaknesses I have discussed warrant a lower score.

---

> ### Author Rebuttal · Authors · 2025-07-30
>
> Thank you very much for your thoughtful and constructive feedback. Below, we respond to your points and questions.
>
>
>
> > 5. The paper could benefit from having figures that describe the causal abstraction for mechanistic interpretability procedure and highlight the impact of Thm. 1.
>
> We have added two new figures to the revised version of the paper that directly address this point:
> * A visual illustration of the non-linear representation dilemma, visualising how causal abstraction works and demonstrating how increasing the complexity of the alignment map ϕ can improve interchange-intervention accuracy (IIA). This figure is in the main body of the paper.
> * A schematic overview of the causal abstraction definitions introduced in Section 3. This figure highlights the subset–superset relationships between different definitions, illustrating how constraints on the allowed abstraction map and interventions shape these relationships. This figure is in the appendix.
>
>
> > 6. The paper does not offer many new novel contributions and in some ways serves as a review or position paper. [...] The main contributions of the paper are Thm. 1 (with proof in the appendix) and the experimental results. The contributions are impactful, but they are mainly adding to discussion points surrounding the efficacy of this mechanistic interpretability approach rather than offering new solutions.
>
> Thank you for this perspective. We agree that a central focus of the paper is to mathematically and empirically demonstrate the non-linear representation dilemma. While this contribution is more diagnostic than prescriptive, we believe it fills a gap in the current literature. In our view, clarifying and naming this issue is a necessary step before more targeted solutions can be developed. Further, we explicitly discuss potential next steps, such as incorporating complexity constraints into causal abstraction, drawing parallels to similar concerns raised in the probing literature.
>
> Beyond this, we also added a new paragraph to emphasise the role of generalisation in causal abstraction:
>
> **The Role of Generalisation.** We now highlight that Theorem 1 provides an existence proof for a perfect abstraction map (thus guaranteeing perfect IIA) between a DNN and an algorithm. This existence proof, however, leverages complex interactions between the intervened hidden states and the DNN’s structure, requiring perfect information about both and thus representing a form of extreme overfitting. Crucially, this theorem offers no guarantees regarding the learnability of the alignment map ϕ from limited data or its generalisation to unseen inputs. This gap between theoretical existence and practical learnability becomes evident in practice. For instance, in an additional experiment on the IOI task [added to the appendix in the revised version], we show that when training and test sets contain disjoint sets of names, the learned alignment map fails to generalise, resulting in low IIA on the test set. This suggests that generalisation should play a crucial role in causal abstraction analysis, as the ability to learn abstraction maps that transfer beyond training data seems fundamental to interpreting a model, distinguishing a genuine understanding of its inner workings from mere training pattern memorisation.
>
> As a brief note—should it be of interest—we are currently building on the analysis in this paper to further explore how features are encoded in models. Specifically, we are evaluating the Carry One variable from the Mechanistic Interpretability Benchmark (MIB) to better understand this variable’s encoding. So far, we have observed that a linear (but non-orthogonal) alignment map achieves >0.96 IIA in that task—yet, this alignment map does not appear to reflect the underlying mechanism and seems to also “hack” the DNN. We hope this ongoing work will lead to deeper insights and, eventually, more practical guidance for causal abstraction.
>
>
>
>
>
> > 7. To what extent does Thm. 1 imply the problem is caused by the linearity assumption of the alignment rather than other factors? For example, perhaps IIA is an imperfect metric of measuring alignment since IIA does not consider semantic meaning of the algorithm nodes.
>
> We’re not entirely sure we fully understand this question. Could you give an example of how a metric that considers the semantic meaning of the algorithm nodes should look?
>
> As a minor clarification: the assumption of linearity is not the root cause of the non-linear representation dilemma; instead, the dilemma arises from the use of unrestricted non-linear alignment maps. However, relying exclusively on linear maps is also problematic, as noted in recent work (e.g., Kantamneni and Tegmark, 2025). This creates the tension: unrestricted non-linear maps are too flexible and can yield misleading alignment; linear maps may miss important aspects of the true mechanism. This tension reflects the non-linear representation dilemma.
>
>
>
>
> > 8. How does the topic of alignment relate to disentangled causal representation learning? As in, is the idea of learning the alignment map similar to the idea of learning a representation map that disentangles the inputs into latent causal factors?
>
> Yes, there is a meaningful connection. Causal abstraction—under distributed or constructive $\tau$—requires disentangling the representations of different variables. This is due to the inherent requirement for partitioning the latent variable space in these types of causal abstraction, which should disentangle inputs into such latent causal factors.
>
>
> ---
>
> Please don’t hesitate to reach out if we have misunderstood any part of your feedback or if additional clarification would be helpful. We’d be happy to continue the discussion and address any remaining questions.
>
> ### References
>
> Kantamneni and Tegmark. (2025). Language models use trigonometry to do addition

---

> > ### Comment · Reviewer_XbMd · 2025-08-07
> >
> > I thank the authors for their rebuttal and appreciate the new additions. The response clarifies many details for me, and the work looks very promising. I maintain my positive score.
> >
> > > We’re not entirely sure we fully understand this question. Could you give an example of how a metric that considers the semantic meaning of the algorithm nodes should look?
> >
> > I am largely thinking about whether there are limitations given that IIA only measures interventional quantities which are computed from the functional definitions of the algorithm model. It may be possible that individual nodes in the high-level algorithm model have English interpretations, but these interpretations could be completely changed without changing the DAG structure and then have no influence on IIA. Or, perhaps more seriously, it is possible that the internal functions of the algorithm may change in a way that does not change IIA.

---

> > > ### Author Response · Authors · 2025-08-08
> > >
> > > Thank you very much for your response. We are glad that our previous response helped clarify some of your questions and comments.
> > >
> > > > I am largely thinking about whether there are limitations given that IIA only measures interventional quantities which are computed from the functional definitions of the algorithm model. [...] Or, perhaps more seriously, it is possible that the internal functions of the algorithm may change in a way that does not change IIA.
> > >
> > > Yes, that is a good point. There is indeed an important distinction between what IIA and causal abstraction measure: while IIA only measures whether the outputs of a DNN and an algorithm match under interventions, the definitions of causal abstraction we explore in our paper further require the existence of a function $\tau$ which allows the value of each node in the algorithm to be predicted from the DNN’s hidden states. We hint at this point in lines 223 to 225:
> > >
> > > “Notably, DAS mostly ignores how function $\tau$ is constructed, relying solely on the assumed definition of $\tau_{\eta_y}$. Finding a low-loss alignment map $\phi$ is then assumed as sufficient evidence that $\mathtt{A}$ is an input-restricted distributed abstraction of $\mathtt{N}$.”
> > >
> > > As the reviewer points out, if two algorithms have the same structure and the same output, they may also lead to the same IIA—despite their nodes having potentially different semantics. In practice, IIA should thus be viewed as an approximation to causal abstraction (or an upper bound), rather than a full measurement of it. A complete causal abstraction assessment would require verifying whether a suitable $\tau_\eta$ exists for all nodes $\eta$.
> > >
> > > We acknowledge that this opens the door to edge cases where the behavior indicated by IIA does not necessarily reflect the model’s actual implementation. For example, assume a node in an algorithm distinguished between four possible values (e.g., $\{1, 2, 3, 4\}) of a variable but its downstream result only depends on simpler distinction (e.g., whether the node’s value is even or odds). In this case, even if the DNN only encodes the simpler distinction (i.e., even or odds), we may still obtain a strong IIA score.
> > > We will try to highlight this point more strongly in our manuscript.
> > >
> > > We sincerely appreciate your thoughtful review and the constructive dialogue it has fostered. Please let us know if we have misunderstood any part of your follow-up comment, or if there are any further points you think deserve clarification.

---

> ### Author Response · Authors · 2025-08-05
>
> Dear Reviewer,
>
> Thank you again for your review and let us know if there are any remaining questions or concerns we can address.
>
> Best,
> The authors

---

### Official Review · Reviewer_x7mN · 2025-07-22

**Clarity:** 3
**Significance:** 4
**Originality:** 4
**Rating:** 5
**Confidence:** 4

**Summary:**

The most popular formalization of interpretability so far is the theory of _Causal abstractions_. They define a neural network (NN) as a complex computational graph, and an interpretation (G, m) as a simple (abstract) computational graph G, plus a map m of the graph's nodes and edges map to the NN's. Then, the interpretation (G, m) is a good one if there exists a mapping \tau that maps values of the NN's nodes to their corresponding abstract graph nodes; and causally intervening in both NN and abstract graph A in the same way provides correct results.

It is well known that if \tau is trivial, then causal abstraction is pretty useless. For example, if \tau_n(x) = 0 for all nodes, all causal interventions produce the desired result and the graph is considered good.

The authors of this forcefully argue for the notion that, if we allow \tau to be arbitrarily powerful, causal abstractions are also vacuous. The theoretical argument centers on \tau partitioning NN activations into arbitrary groups (gerrymandering, if you will) that correspond to the algorithm's abstract values, which can be done regardless of whether the NN implements the algorithm A, so long as the NN has sufficiently many states. The practical argument centers on fitting a couple of well-known interpretability 'found' algorithms into completely unrelated or even randomly initialized networks, by using an up-to-8-layer NN as the map \tau.

The authors then conclude that restricting \tau (to be, for example, linear) works well.

**Questions:**

# Section 4, Theorem 1's assumptions

## Appendix F, proof for the injectivity assumption: Lemma 8

Because I was skeptical of transformers being input-injective, I dug into the proof, which line 237 claims exists in Appendix F.

The proof of this Theorem 2 uses Lemma 8 ("MLPs are injective almost surely") also does not assume linearities are injective.

My question is about the final line of the proof in section F.3. Each line seems to check out, except the last part in words in lines 1408-1411: "Now, eq. 87 holds because the right-hand side is a constant while the left-hand side is a random variable drawn from a continuous distribution". The left-hand-side is a row of the L layer's weights W_L, the right-hand-side is an expression involving the weights of other rows of W_L. **How is the RHS a constant?**

## Assumption 5: The DNN solves the task T

You state in 247-248 that "it would be impractical in practice to evaluate a NN that does not perform the task correctly. But that is exactly what you do in experiments! Fig 3, line 328-329 state that you're using random NNs.

**Why does the theorem need this assumption?**

## Assumption 3: output-surjectivity
You state that "in practice, all outputs seem to still be producible by language models" . This is directly contradicted the phenomenon of "glitch tokens" found by Rumbelow and Watkins (2023, https://www.lesswrong.com/posts/aPeJE8bSo6rAFoLqg/solidgoldmagikarp-plus-prompt-generation), where some token's unembedding has smaller norm than every other token in its direction, so there is no activation that will produce it as the maximum.

Again this shows that the theorem is false in practice, but it's *approximately* true in many circumstances and is a useful intuition pump. Please present it as such.

# Figure 1: right plot, no perfect IIA?
in the right-most plot in Figure 1, none of the alignment maps are able to get perfect IIA. Why do you think this is the case?

# Arguments and remarks from page 9

L379: claims that Burns et al. 2023 "leverag[es] unsupervised probes" in order to control the probe's complexity. In what way does unsupervised learning control the probe's complexity? Does that not come from the fact the probes are linear?

L388-389, where the authors state they cannot formalize why Fig. 3 (right) inclines them to support the linear probe. That's OK, but what about an informal justification? One could say that it's provided in the previous sentence ("\phi^lin accompanies the actual DNN performance much more closely"). If so, please just remove this sentence.

**Ethical Concerns:**

["NO or VERY MINOR ethics concerns only"]

**Final Justification:**

My concerns were minor to begin with, and the authors addressed all of them. I believe the paper provides significant insight into causal abstractions, its mathematics are correct and has a solid empirical evaluation section.

**Limitations:**

mostly. The authors addressed some of the limitationsI found in the final section. But I think they're problems with critical assumptions of the main theorem, so the assumptions should be addressed **when they're stated** and not at the end where readers might not find it.

**Quality:**

3

**Strengths And Weaknesses:**

# Strengths

The paper points out a critical problem with causal abstractions that many people that work with them do not think of, but that can pose problems. In practice basically all papers assume that the mapping \tau is simple in some way, so it does not come up much. However, it's still an assumption that practitioners are making without being aware of, which could go away soon; for example if causal abstraction is used as a metric for automatic interpretability. For this reason, I think this is a fundamentally significant and original paper and all of its weaknesses are about presentation or about mistakes in the argument that make it not quite go through; but which can fixed easily.

Here's my own experience reading the paper: I highlighted several chunks of the abstract, scribbling notes in disbelief, like "Surely it's impossible to map [arbitrary] models to [arbitrary] algo[rithm]s", implying that instead it's only possible to map arbitrary algorithms to NN models, because algorithms are simple whereas models are much complicated. I had also forgotten that causal abstraction only specifies that you can map high-level to low-level model, but not necessarily the opposite. If you were forced to map low-level to high-level model, then surely that map doesn't work for arbitrary low-level models? I think that _is_ true and made me believe this paper relatively quickly, but for the wrong reasons; I then thought that this paper spent too long on proofs for a relatively trivial statement. I read it again and realized the statement was not so trivial.

# Weaknesses

**My score is low because I want my comments to be addressed, but they should be easy to address; so I am hopeful that I will be able to recommend this paper to be accepted.** Here are the weaknesses, all fixable:

## Section 4, Theorem 1's assumption: input-injective NNs

I think the discussion in lines 234-237 of the "DNNs are input-injective" assumption is pretty misleading. First, this is clearly untrue: transformers use SiLUs or GeLUs which hit a negative peak at zero, and then go positive on both sides. ReLUs are also popular. Second: the citation of Morris et al. 2023 which is used to claim the assumption is "well-approximated in practice", only recovers 92% of 32-token inputs -- a far cry from 100%. Third, the authors themselves later write in lines 318-319 that this is not true in practice for their setup, and say this again in lines 407-410. Please be more upfront with this!

All of this needs to be said in Section 4 when discussing the assumptions. Input-injectivity is just not true. Which kind of invalidates the theorem. But in reality, NNs are ~close to input-injective, so they can be mapped to ~a large amount of algorithms that they don't implement. The theorem is still a good intuition builder.

## Clarity: give intuition for Theorem 2
It would be nicer to give an intuition for the theorem. That is, if the NN is injective (it has a different state for every input sequence) and the algorithm is allowed to be arbitrarily complicated, then we can just gerrymander the hidden states to map to the abstract model. The hidden states need to be countable so there exists a computable algorithm that maps them to an abstract algorithm. (I haven't checked if this is the case thorougly, but the theorem does make use of partitions).

---

> ### Author Rebuttal · Authors · 2025-07-30
>
> Thank you for reviewing our paper in such detail. Your engagement with our work is greatly appreciated! We agree that highlighting the limitations of our assumptions early on (when they are first stated) is important, and we implemented this change in our paper.
>
> The main changes we thus implemented, following suggestions, try to highlight that input-injectivity and output-surjectivity are likely not to hold exactly, but are close to true in practice, which is enough for many algorithms. These are the specific changes:
> * Input-injectivity: We changed “This assumption is also present in prior work (e.g., Pimentel et al., 2020b) and seems to be well approximated in practice (Morris et al., 2023). Further, we show in App. F that this is almost surely true for transformers at initialisation—assuming real-valued weights and activations (and thus ignoring floating point precision).” to “This assumption is also present in prior work (e.g., Pimentel et al., 2020b) and we show in App. F—assuming real-valued weights and activations—that this is almost surely true for transformers at initialisation. Due to floating point precision and neural collapse (Papyan et al., 2020), it is likely not to hold fully in practice; however, it still seems to be well-approximated in many empirical settings (e.g., see Morris et al. (2023) and App. H).” App. H (newly added) contains new experiments on input-injectivity for the hierarchical equality task (see details below).
> * Output-surjectivity: We changed “Notably, this assumption may not hold in theory, due to issues like the softmax-bottleneck (Yang et al., 2018). In practice, however, even with large vocabulary sizes, all outputs seem to still be producible by language models (Grivas et al., 2022).” to “[...] even with large vocabulary sizes, it seems that most outputs can still be produced by language models (Grivas et al., 2022) which is enough for these DNNs to be abstracted by many algorithms.”
>
> We hope these adjustments clarify the limitations sufficiently, and we'd be happy to further iterate based on the reviewer's feedback.
>
> > Input-injectivity is just not true. Which kind of invalidates the theorem. But in reality, NNs are ~close to input-injective, so they can be mapped to ~a large amount of algorithms that they don't implement. The theorem is still a good intuition builder.
>
> As suggested (and as mentioned above), we made this shortcoming of our assumptions clearer in the paper.
>
> We previously hypothesised (lines 318-319) that "perfect alignment remains elusive because the MLP is not actually injective or surjective at initialisation due to the ReLU activation function." However, after further investigation, we ruled out input-injectivity as the main cause of low IIA in the hierarchical equality task. We sampled 1.28M inputs, passed them through the MLP to get hidden states, and found no collisions across 10 random seeds. Thus, input-injectivity held for these inputs and is unlikely to have caused the observed issues. We added an appendix with this analysis.
>
> This reopens the question:  why didn’t we achieve perfect (100%) IIA in all our experiments, particularly for the "identity of first argument" intervention (Fig. 1, right), as also noted by the reviewer. Here are some thoughts. First,  output-surjectivity is satisfied here, since the model is able to perform the task (an “identity of first argument” intervention does not critically change the possible model outputs). High IIA for the “Both Equality Relations” and “Left Equality Relations” further suggest that other assumptions hold well enough. We believe the lower "identity of first argument" IIA results stem primarily from two factors:
>
> 1. **Generalisation Gap.**
> Theorem 1 presents an existence proof: there exists a function such that a DNN is abstracted by an algorithm. However, it does not discuss the learnability of this function and can be seen as a form of “strong overfitting”, providing no guarantees that a learned map would generalise to unseen data. In our experiments—where training and test sets are separate—the generalization gap becomes apparent.
>
> 2. **Model Capacity Limitations.**
> While Theorem 1 assumes arbitrarily complex models, a RevNet is relatively limited in capacity.  Our RevNet appears insufficiently expressive to fully overfit a large dataset (1.28M samples), never exceeding a 0.55 training IIA. But with just 1,000 samples, the RevNet achieved >99% training IIA.
>
> Given these results, we added an extra paragraph to our discussion about the role of generalisation in interpretability and causal abstraction analyses.
>
> > Clarity: give intuition for Theorem
>
> We agree with the reviewer, and we added the following proof sketch for our theorem:
>
> “We prove this Theorem by induction. We fix an arbitrary partitioning of the DNN $\mathtt{N}$'s neurons. Then, for each layer, we show that we can create a suitable alignment map that: (i) preserves the information of its and preceding nodes, and (ii) enforces that the DNN's output matches the output of the intervened algorithm $\mathtt{A}$ under any intervention up to that layer. These two conditions are sufficient to guarantee that an accompanying $\tau$ exists, and thus that $\mathtt{A}$ is an input-restricted distributed abstraction of $\mathtt{N}$. We enforce (i) by ensuring that $\phi_{\ell}$ is injective on each dimension—which is possible due to, in practice, its inputs only covering a countable set. We then leverage $\mathtt{N}$'s surjectivity (Assump. 3) to enforce (ii).”
>
> Additionally, we have added a new figure to our paper that offers a visual representation of how increasing alignment complexity can lead to higher or perfect IIA, which conveys intuition about the non-linear representation dilemma.
>
> > Appendix F, proof for the injectivity assumption: […] How is the RHS a constant?
>
> In Equations (82–83), we constrain the probability space by fixing the values of all weights of the MLP except one—specifically, the weight from node $i$ in the second last layer to node 1 in the final layer. This random variable is denoted $W^′$. The other parameters are treated as constants in the following equations, given their realisations in the conditions.
> Therefore, the RHS of Equation (87) does not depend on $W^′$, but only on fixed parameters and the input, and is thus constant. We tried to make this clearer in the paper.
>
>
> > Assumption 5: The DNN solves the task T [...] Why does the theorem need this assumption?
>
> Our theorem states that, under the given assumptions, any algorithm is an input-restricted distributed abstraction of any model. By definition, this abstraction only holds if the model’s behaviour exactly matches that of the algorithm—requiring IIA of 1.0. If the model does not solve the task, perfect IIA is impossible, as the *non*-intervened inputs yield incorrect outputs. Thus, assuming the model solves the task is required. In practice, however, even if the DNN does not perfectly solve the task, there still exists an alignment map which achieves correct outputs on all intervened inputs.
>
> We use random models in our experiments not only to provide empirical examples of the theorem, but to illustrate its relevance in a practical setting where not necessarily perfect IIA is required or reachable. A random neural network clearly does not implement the intended algorithm, yet the alignment methods can still report high IIA.
>
> > Assumption 3: output-surjectivity [...]  This is directly contradicted the phenomenon of "glitch tokens" found by Rumbelow and Watkins [...].
>
> We appreciate the comment about glitch tokens. While the blog post notes that “many of these tokens were among those closest to the centroid of the entire set of 50,257 tokens,” this alone does not confirm that these tokens are non-generatable given some hidden activation. Due to this unclear connection, we opted not to reference this explicitly. Still, based on blog comments, some glitch tokens may indeed be non-generatable.
>
> > in the right-most plot in Figure 1, none of the alignment maps are able to get perfect IIA. Why do you think this is the case?
>
> See answer to input-injectivity above.
>
>
> > L388-389, where the authors state they cannot formalize why Fig. 3 (right) inclines them to support the linear probe. That's OK, but what about an informal justification? One could say that it's provided in the previous sentence [...]. If so, please just remove this sentence.
>
> We removed this sentence.
>
>
> > L379: claims that Burns et al. 2023 "leverag[es] unsupervised probes" in order to control the probe's complexity. In what way does unsupervised learning control the probe's complexity? Does that not come from the fact the probes are linear?
>
> We added this example following the intuition that the complexity–accuracy trade-off of probing arises only in supervised settings, where increasing probing complexity allows increasingly more complex “features” to be extracted from the model’s representations. Unsupervised probing would, in this respect, be immune to such problems, since without supervision these “gerrymandered” maps would not be found. We acknowledge, however, that such reasoning was not made explicit in our paper, and we have now added this explanation to our paper.
>
> Again, we thank the reviewer for such a detailed reading of our paper. Please don’t hesitate to let us know if we’ve misunderstood any part of your feedback or if further clarification is needed. We’d be happy to address any additional questions.
>
> ### References
>
> Pimentel et al. (2020b). Information-theoretic probing for linguistic structure
>
> Morris et al. (2023). Text embeddings reveal (almost) as much as text
>
> Papyan et al. (2020). Prevalence of Neural Collapse during the terminal phase of deep learning training
>
> Grivas et al. (2022). Low-rank softmax can have unargmaxable classes in theory but rarely in practice

---

> ### Author Response · Authors · 2025-08-05
>
> Dear Reviewer,
>
> Thank you again for your review and let us know if there are any remaining questions or concerns we can address.
>
> Best,
> The authors

---

> > ### Comment · Reviewer_x7mN · 2025-08-05
> > **I'm satisfied with how you addressed concerns, but I still have some suggestions.**
> >
> > Thank you for the extra discussion, I think you've basically addressed everything.
> >
> > I kind of want to push back against describing too many things as limitations when they're not so bad. The input-injectivity and output-surjectivity assumptions are not perfectly achieved, which limits the set of arbitrary algorithms which can be shown to causally abstract a NN. But the set of algorithms is that can be causally abstracted is still much larger than what we have in practice! As I understand it, your theorem degrades very gradually as opposed to suddenly, and its assumptions are satisfied; so it is still definitely useful.
> >
> > ---
> >
> > I really appreciate the extra discussion about input injectivity! I had not realized that in the hierarchical equality task things were exactly input-injective already and the explanation is more likely to be probe capacity and learnability.
> >
> > ---
> >
> > Thank you for explaining why the RHS in the appendix proof is a constant too -- I checked and I understand. Perhaps just writing "is a constant because of the law of total probability box we opened a couple of equations above" + " p_{W \ W'}(q | ...) = 1".
> >
> > ---
> >
> > This discussion about the "model solves the task" assumption is great, and clarifies things for me. Consider finding a way to include this intuition in the paper as well!
> >
> > > Our theorem states that, under the given assumptions, any algorithm is an input-restricted distributed abstraction of any model. By definition, this abstraction only holds if the model’s behaviour exactly matches that of the algorithm—requiring IIA of 1.0. If the model does not solve the task, perfect IIA is impossible, as the non-intervened inputs yield incorrect outputs. Thus, assuming the model solves the task is required. In practice, however, even if the DNN does not perfectly solve the task, there still exists an alignment map which achieves correct outputs on all intervened inputs.
> >
> > > We use random models in our experiments not only to provide empirical examples of the theorem, but to illustrate its relevance in a practical setting where not necessarily perfect IIA is required or reachable. A random neural network clearly does not implement the intended algorithm, yet the alignment methods can still report high IIA.
> >
> > ---
> >
> > Also, regarding the informal evidence the supports the linear probe:
> >
> > > We removed this sentence.
> >
> > I think it would be a shame to not state the reasoning that I believe you had, if it's as succint as this:
> >
> > "phi^lin accompanies the actual DNN performance much more closely, [which is weak evidence in favor of the linear representation hypothesis]"
> >
> > If you prefer to just remove the sentence that's also OK.
> >
> > ---
> >
> > I'm going to go raise my score to 5: Accept now.

---

> ### Author Response · Authors · 2025-08-06
>
> Thank you very much for your response. We're glad we were able to address your questions and comments.
>
> We agree with your observation: while the theorem formally holds only under perfect alignment, in practice the observed IIA tends to degrade gradually as input-injectivity and output-surjectivity diminish, depending on the number of affected intervention samples. We will also aim to clarify the intuition around the model “solving the task,” as we agree this may be helpful for readers. Regarding the removed sentence, our intention was primarily to omit the phrase “We (the authors), however, cannot make this intuition formal to justify why we believe this is the case,” while preserving—and slightly rephrasing—the preceding sentence concerning the linear representation hypothesis, as suggested.
>
> Once again, thank you for your detailed feedback and for contributing to such a constructive and valuable review process. We sincerely appreciate it.

---

### Decision · Program_Chairs · 2025-09-17

**Decision:**

Accept (spotlight)

**Comment:**

This paper presents the non-linear representation dilemma: if we allow arbitrary alignment maps in causal abstraction, there is an apparent paradox wherein we can map neural networks to algorithms even when they are incapable of solving a task. This has shades of the no free lunch theorem and various impossibility results when trying to interpret models without making any assumptions. Although not surprising, causal abstractions have recently gained traction and this work suggests caution is needed when adopting them in practice.

During the discussion, several minor concerns were raised that were addressed by the authors. In the end there was unanimous agreement to accept this thought provoking paper.